# Graph Coarsening using Game Theoretic Approach

**Sonali Raj**  *sonaliraj.rs.cse23@iitbhu.ac.in*
*Department of Computer Science and Engineering*
*Indian Institute of Technology (BHU) Varanasi, India*

**Manoj Kumar**  *manojkr@iitism.ac.in*
*Department of Electronics Engineering*
*Indian Institute of Technology (Indian School of Mines) Dhanbad, India*

**Sumit Kumar**  *kumars@nitj.ac.in*
*Department of Computer Science and Engineering*
*Dr B R Ambedkar National Institute of Technology, Jalandhar, India*

**Ruchir Gupta**  *rgupta.cse@iitbhu.ac.in*
*Department of Computer Science and Engineering*
*Indian Institute of Technology (BHU) Varanasi, India*

**Amit Kumar Jaiswal**  *amit.chr@iitbhu.ac.in*
*Jay Chaudhry Software Innovation Centre*
*Indian Institute of Technology (BHU) Varanasi, India*

**Reviewed on OpenReview:** *https://openreview.net/forum?id=5vLBjQJCln*

## Abstract

Graph coarsening is a method for reducing the size of an original graph while preserving its structural and feature-related properties. In graph machine learning, it is often employed as a preprocessing step to improve efficiency and scalability when handling large graph datasets. In this study, we address the challenge of coarsening an original graph into a coarsened graph that retains these characteristics. We propose a Cooperative-Based Graph Coarsening (CGC) algorithm, which leverages cooperative game theory as a framework for combinatorial optimization, aiming to minimize the total Dirichlet energy of the graph through localized optimizations. We prove that the proposed coarsening game is a potential game that guarantees convergence to a stable coarsened graph. Tests on real-world datasets demonstrate that CGC algorithm surpasses prior state-of-the-art techniques in terms of coarsened graph accuracy and achieves reduced time complexity. These results highlight the potential of game-theoretic approaches in the advancement of graph coarsening techniques.

## 1 INTRODUCTION

Graphs are powerful data structures used to represent complex relationships and interactions among entities. Graphs offer a natural and flexible way to model real-world systems, which are widely employed across varied domains such as social networks, biological systems, transportation infrastructures, and communication networks (Fan et al., 2019; Li et al., 2021; Rahmani et al., 2023; Wu et al., 2022; Shi & Weninger, 2017). In these representations, entities are modeled as nodes, while the relationships or interactions among them are captured through edges. This framework enables systematic analysis of both local and global connectivity patterns, as well as the structural characteristics of complex systems (Neville & Jensen, 2007; Angles & Gutierrez, 2008).

With the growing need to extract knowledge from graph-structured data, Graph Neural Networks (GNNs) have emerged as a class of deep learning models specifically designed for learning on graphs. By leveraging

both the feature information of nodes and the structure of their neighborhoods, GNNs propagate information through the graph via message passing and aggregation mechanisms. This allows GNNs to capture rich relational patterns and contextual dependencies, making them highly effective for tasks such as node classification, link prediction, and graph classification (Corso et al., 2024).

However, as real-world data becomes increasingly voluminous and interconnected, large-scale graphs have become more common, driven by advances in scientific research, online platforms, and the proliferation of connected devices in the Internet of Things. The application of GNNs to such massive graphs introduces significant computational and memory bottlenecks. The recursive neighborhood expansion in message passing leads to rapidly growing computation, while deep GNN architectures suffer from over-smoothing, wherein node representations become indistinct. Additionally, training GNNs often requires extensive hyperparameter tuning, further increasing time and resource demands. These challenges highlight the need for efficient preprocessing strategies to reduce graph size without compromising structural and semantic integrity. There are various techniques for graph reduction, including graph sparsification (Fung et al., 2011; Bravo Hermsdorff & Gunderson, 2019; Li et al., 2022), graph coarsening (Dickens et al., 2024; Kumar et al., 2023; Loukas, 2019; Huang et al., 2021; Loukas & Vandergheynst, 2018; Wang et al., 2019), and graph condensation (Jin et al., 2021; Xiao et al., 2024; Gao et al., 2025).

Graph sparsification reduces the number of edges while preserving key structural properties such as cuts, pairwise distances, or spectral signatures. However, it retains the full set of nodes, limiting its potential for computational speedup in node-centric operations. Moreover, many sparsification algorithms depend on costly spectral computations (e.g., effective resistance or eigenvalue approximations) and do not always align with downstream task objectives, such as those in GNN-based learning (Li et al., 2022).

Graph condensation, in contrast, synthesizes a small, learnable graph from scratch such that a GNN trained on this synthetic graph approximates the performance of one trained on the original graph. While effective at reducing memory usage during GNN training, condensation methods rely on nested or bi-level optimization (e.g., gradient or distribution matching), which incurs substantial preprocessing cost and memory overhead. Furthermore, they often require label information and offer limited interpretability, which hampers generalization across models and datasets.

Graph coarsening offers a balanced alternative by reducing both nodes and edges through the aggregation of structurally or feature-similar nodes into supernodes, while maintaining the overall topology and semantics of the graph (Chen et al., 2022). This approach retains a clear mapping between the original and reduced graph, allowing interpretability and compatibility with a variety of downstream GNN tasks. As a result, coarsening enables more scalable and efficient graph learning without sacrificing representational fidelity.

Recent graph coarsening approaches fall into three main categories: heuristic-based, optimization-based, and deep learning-based. Heuristic coarsening methods (Loukas, 2019; Huang et al., 2021; Loukas & Vandergheynst, 2018) typically rely only on structural information such as adjacency patterns or spectral approximations, and ignore node features. As a result, coarsened graphs can be less informative and degrade downstream performance (e.g., node classification). Deep learning-based methods (Ying et al., 2018; Jin et al., 2021; Zhao et al., 2020; Zheng et al., 2023; Dickens et al., 2024) leverage graph neural networks (GNNs) to learn coarsening operations as part of graph representation learning. These methods are computationally intensive and do not suit generalized GNNs, as a GNN trained on a particular model does not perform well on other models. Optimization-based coarsening methods (Kumar et al., 2023; 2024) do incorporate structure and features but introduce many hyperparameters and require repeated tuning across datasets and coarsening ratios. This leads to significant computational overhead and undermines the basic purpose of coarsening, i.e., to reduce cost rather than add complexity. In these approaches, the search space becomes increasingly large and multidimensional as the graph size grows. Therefore, these approaches generally fail to produce the desired results when the problem size is large and suffer from scalability issues. Additionally, game-theoretic formulations have been explored mainly for graph clustering (Mandala et al., 2014; Bu et al., 2018; Zhao et al., 2024; Kumar & Gupta, 2021; Li et al., 2024). These game-theoretic clustering methods often rely solely on structural connectivity and ignore node attributes, or they depend on nontrivial hyperparameters that require clustering-specific expertise to tune.

These limitations demonstrate the need for a coarsening technique that is both computationally efficient and feature-aware, while preserving the structural properties of the original graph. Our work aims to develop a fast method without hyperparameter tuning that uses both graph topology and node attributes to produce high-quality coarsened graphs for the improvement of downstream task performance and hence, we introduce a cooperative game-theoretic framework for coarsening.

We utilize cooperative game theory for four key reasons. First, cooperative games allow us to incorporate both structural connectivity and feature similarity in a unified manner. The marginal contribution based cost reflects how much a node benefits the coalition in terms of both graph structure and feature smoothness. Second, the game theory allows us the construction of exact potential game, which guarantees that every time a node updates its coalition choice, the global Dirichlet energy decreases. This alignment ensures that local best responses naturally guide the system toward a stable coarsened representation without the need for tuning. Third, the potential game formulation ensures convergence to a pure Nash equilibrium, which corresponds to a stable coalition structure. Fourth, the existence of at least one Nash equilibrium ensures a stable and high-quality solution. This property makes the method more scalable than heuristic or conventional optimization techniques.

In our game-theoretic formulation, each node acts as a player that strategically forms coalitions with other nodes in the graph. This insight motivates our distributed formulation of the graph coarsening problem, where each player minimizes its cost. In our formulation, coalitions correspond to supernodes, and the cost of each player is defined through its marginal contribution to the Dirichlet energy, which measures feature-signal smoothness on the graph, so minimizing it preserves both connectivity and attribute consistency under coarsening. This creates a direct connection between local node-level decisions and the global objective of minimizing energy distortion in the coarsened graph. This gives a theoretical guarantee of stability and explains the robust, tuning-free performance of CGC across diverse datasets and coarsening ratios. This gives a theoretical guarantee of stability and explains the robust, tuning-free performance of CGC across diverse datasets and coarsening ratios.

The main contributions of this paper are summarized as follows:

- Introduces a novel graph coarsening framework using game theory.
- Introduces a cost function that utilizes marginal contributions to Dirichlet energy, ensuring structural preservation.
- Empirically validates the approach on real-world datasets, demonstrating its effectiveness in node classification tasks.
- Presentes a theoretical proof demonstrating that the proposed game satisfies the conditions of a potential game.

The remaining paper is structured as follows. Section 2 presents the necessary background on graphs and graph coarsening, followed by the proposed problem formulation and the game-theoretic framework. Section 3 introduces the algorithm along with its convergence guarantee. Section 4 provides experimental results on real-world datasets for node classification, structural property preservation, and comparisons to state-of-the-art methods.

## 2 BACKGROUND

This section presents the fundamental concepts of graphs and graph coarsening, including a toy example, followed by the formulation of the proposed problem and its game-theoretic formulation.

### 2.1 Graph

A simple undirected graph with features and labels, denoted as $G = (V, E, W, X, Y)$, is defined by a node set $V = \{V_1, V_2, \ldots, V_p\}$ and an edge set $E \subseteq [V]^2$. The topology of the graph is represented using a weighted adjacency matrix $W \in \mathbb{R}^{p \times p}$, where each entry $w_{ij} \geq 0$ indicates the strength of the connection between the nodes $V_i$ and $V_j$, and $p = |V|$ indicates the total count of nodes. The matrix $X \in \mathbb{R}^{p \times n}$ encodes node features, with each row $x_i$ representing an $n$-dimensional attribute vector corresponding to $V_i$ node, and the label is denoted as $Y \in \mathbb{R}^{p \times l}$, assigns each node a one-hot encoded categorical label, representing its

membership in a predefined set of $l$ classes. To analyze structural properties such as connectivity or spectral characteristics of the graph, we employ the combinatorial Laplacian matrix $\Theta = D - W$, where $D$ is the degree matrix. The Laplacian is symmetric, positive semidefinite, and has zero row sums, making it useful for computational tasks such as graph compression, embedding, manifold learning and spectral preservation. It satisfies $\Theta_{ij} = -W_{ij}$ for $i \neq j$, while diagonal elements represent node degrees. The set of all feasible combinatorial graph Laplacians $\mathcal{L}$ is defined as:

$$\mathcal{L} = \{\Theta \in \mathbb{R}^{p \times p} \mid \Theta = \Theta^T, \Theta_{ij} \leq 0 \text{ for } i \neq j, \Theta_{ii} = -\sum_{j \neq i} \Theta_{ij}\} \tag{1}$$

However, directly analyzing large-scale graphs is computationally expensive, which requires reducing their size while retaining structural properties. In the next subsection, we discuss graph coarsening in detail.

## 2.2 Graph Coarsening

Graph coarsening aims to construct a smaller and tractable graph $G_c = (V_c, E_c, W_c, X_c, Y_c)$ with k nodes from a larger graph $G = (V, E, W, X, Y)$ with p nodes, while preserving structural and featural properties of the original graph, where $k \ll p$. Each node in $G_c$, referred to as a supernode, represents a merging of nodes from $G$ grouped based on shared attributes or structural similarities. The coarsening process aims to construct a mapping matrix $C \in \mathbb{R}^{p \times k}$, where $C_{ij} > 0$ signifies that the $i$-th node of the original graph $G$ is mapped to the $j$-th supernode of the coarsened graph $G_c$. This mapping adheres to a many-to-one relationship, as each node in $G$ is assigned to exactly one supernode in $G_c$, while preserving an orthogonal structure among the supernodes. The matrix $C$ lies in the following set $S_c$:

$$S_c = \left\{C \in \mathbb{R}_+^{p \times k} \;\middle|\; \langle C_j, C_{j'} \rangle = 0 \; \forall j \neq j'; \; \langle C_j, C_j \rangle = d_j, \; \|C_j\|_0 \geq 1, \; C_j \in \mathbb{R}^p; \; \|[C^\top]_i\|_0 = 1, \; [C^\top]_i \in \mathbb{R}^k\right\} \tag{2}$$

Here, $\langle C_j, C_{j'} \rangle$ denotes the standard inner product of the vectors $C_j$ and $C_{j'}$, $\|C_j\|_0$ denotes the number of non-zero entries of a vector (the $\ell_0$-norm), $C_j$ and $C_{j'}$ denote the $j$-th and $j'$-th columns, and $[C^\top]_j$ represents the $j$-th row of $C$ of the loading matrix $C$. The orthogonality condition $\langle C_j, C_{j'} \rangle = 0$ ensures that the columns of $C$ are mutually orthogonal. This condition leads to two constraints: first, the identity $\langle C_j, C_j \rangle = d_j$ captures the aggregated contribution of the nodes assigned to the $j$-th supernode; second, the constraint $\|[C^\top]_i\|_0 = 1$ enforces that each row of $C$ contains exactly one non-zero entry, meaning that each node in the original graph is associated with a single supernode in the coarsened graph, thereby implementing a hard assignment.

Let $P \in \mathbb{R}^{k \times p}$ denote the coarsening matrix, which serves as the (pseudo-)inverse of the loading matrix $C$. That is, we define $P = C^\dagger$, where $C^\dagger$ is the Moore-Penrose pseudoinverse of $C$. This matrix enables the reconstruction (or projection) of coarse-level representations back to the original graph domain. The Laplacian matrix of the original graph $\Theta$, the Laplacian matrix of the coarsened graph $\Theta_c$, the feature matrix of the original graph $X$ and the feature matrix of the coarsened graph $X_c$, together satisfy the following relations such as:

$$\Theta_c = C^\top \Theta C, \quad X_c = PX, \quad C = P^\dagger, \quad Y_c = \arg\max(PY) \tag{3}$$

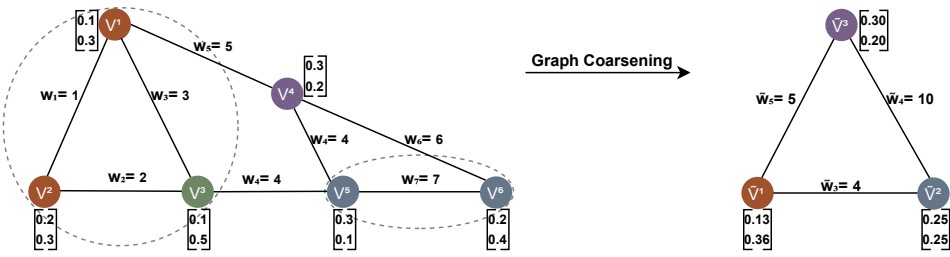

Figure 1: Illustration of the graph coarsening process on a small example.

We demonstrates how node grouping of a simple graph $G$ with six nodes $V = \{V_1, V_2, V_3, V_4, V_5, V_6\}$ through the assignment matrix $\Theta \in \mathbb{R}^{3\times6}$ leads to reduced-dimensional graph $G_c$ with nodes $V_c = \{\tilde{V}_1, \tilde{V}_2, \tilde{V}_3\}$ and where each node is assigned a two-dimensional feature vector. The coarsening matrix $P$ is defined below and its pseudoinverse is calculated using $C = P^{\dagger}$. The feature matrix $X \in \mathbb{R}^{6\times2}$ and its coarsened feature matrix $X_c \in \mathbb{R}^{3\times2}$ is calculated using $X_c = PX$, as shown in equation (1). Now the original and coarsened Laplacian matrices $\Theta \in \mathbb{R}^{6\times6}$ and $\Theta_c \in \mathbb{R}^{3\times3}$, computed as $\Theta_c = C^{\top}\Theta C$:

$$P = \begin{bmatrix} \frac{1}{3} & \frac{1}{3} & \frac{1}{3} & 0 & 0 & 0 \\ 0 & 0 & 0 & 0 & \frac{1}{2} & \frac{1}{2} \\ 0 & 0 & 0 & 1 & 0 & 0 \end{bmatrix} \quad , \quad \Theta = \begin{bmatrix} 9 & -1 & -3 & -5 & 0 & 0 \\ -1 & 3 & -2 & 0 & 0 & 0 \\ -3 & -2 & 9 & 0 & -4 & 0 \\ -5 & 0 & 0 & 15 & -4 & -6 \\ 0 & 0 & -4 & -4 & 15 & -7 \\ 0 & 0 & 0 & -6 & -7 & 13 \end{bmatrix} \quad \text{and} \quad \Theta_c = C^{\top}\Theta C = \begin{bmatrix} 9 & -4 & -5 \\ -4 & 14 & -10 \\ -5 & -10 & 15 \end{bmatrix} \tag{4}$$

To ensure meaningful coarsening, modern methods optimize the mapping matrix $C$ under constraints such as sparsity, orthogonality, and balance among supernodes. These techniques incorporate the characteristic matrix $X$, and in some cases, both $X$ and the label information $Y$, together with the adjacency matrix, to generate coarser graphs more informative. The resulting graph facilitates downstream tasks while maintaining the structural and spectral fidelity of the original graph.

## 2.3 Proposed Problem Formulation

Given a weighted adjacency matrix $W \in \mathbb{R}^{p\times p}$ and a feature matrix $X \in \mathbb{R}^{p\times n}$ corresponding to the node set $V = \{V_1, V_2, \ldots, V_p\}$ of the original graph, we model the graph coarsening problem as a cooperative game. In this formulation, graph nodes are treated as players, and the edges associated with each node define the possible actions available to that player. The game aims to study how groups of players (nodes) form coalitions (supernodes) to achieve a common goal and fairly distribute outcomes among the players. Throughout this paper, we use the terms player and node interchangeably. The goal of each coalition in this cooperative game is to minimize the cost of each player. The objective is to determine a set of coalitions $\mathcal{S} = \{S_1, S_2, \ldots, S_k\}$, where each coalition $S_i \subseteq V$, that collectively minimizes the overall Dirichlet energy of the coarsened graph and is formulated as

$$\min_{\mathcal{S}\in\mathcal{S}_T} \sum_{\substack{S_i,S_j\in\mathcal{S} \\ i<j}} \tilde{w}_{ij}\|\tilde{x}_i - \tilde{x}_j\|^2 \quad \text{s.t.} \quad \mathcal{S}_T = \left\{\mathcal{S} \mid \bigcup_{i=1}^{k} S_i = V, \; S_i \cap S_j = \emptyset \; \forall i \neq j \right\} \tag{5a}$$

$$\tilde{w}_{ij} = \sum_{u\in S_i, v\in S_j} w_{uv}, \quad \tilde{x}_i = \frac{1}{|S_i|}\sum_{v\in S_i} x_v, \quad \forall i \neq j, \; \forall i, \; \forall u, v \in S_i, \; \text{path}(u,v) \leq 2 \tag{5b}$$

where each $S_i \in \mathcal{S}$ denotes a subset of nodes mapped to a corresponding supernode $\tilde{V}_i$ and here $i$ represents the $i$th supernode. The term $w_{ij}$ refers to the $(i,j)$-th entry of the original graph weight matrix $W$, while $\tilde{w}_{ij}$ denotes the $(i,j)$-th entry of the coarsened weight matrix $W_c$, representing the weight of the edge between supernodes $\tilde{V}_i$ and $\tilde{V}_j$. Similarly, $x_i$ denotes the $i$-th row of the original feature matrix $X$, and $\tilde{x}_i$ denotes the $i$-th row of the coarsened feature matrix $X_c$, representing the feature vector of the supernode $\tilde{V}_i$. The term $\|\tilde{x}_i - \tilde{x}_j\|^2$ quantifies the local variation in the feature signal between the supernodes, and the summation aggregates this variation between all pairs of connected supernodes. Here, the summation over $i < j$ ensures that each coarsened edge is counted exactly once in the global dirichlet energy. The constraint $S_T$ defines the coalition structure, ensuring a complete and disjoint partition of the node set $V$ into $k$ subsets. $|S_i|$ is the cardinality (size) of the nodes in set $S_i$. The $\text{path}(u,v)$ defines the length of the shortest path between two nodes $u$ and $v$ within the same subset $S_i$ and constraint by most two in the original graph. This constraint ensures that each coalition forms a connected subgraph with mutual reachability. By minimizing the objective function defined in equation (5a), the resulting supernodes exhibit smooth variations and preserve the structure of the original graph while ensuring energy-efficient coarsening. Minimizing this problem is NP-hard, as it generalizes diameter-constrained graph partitioning as described in (Zhang et al.). We formulate the graph coarsening task as a cooperative game such that a set of coalitions $\mathcal{S} = \{S_1, S_2, \ldots, S_k\}$ by incorporating structural constraints. This new interpretation enables game-theoretic reasoning over graph structures. The formal definition of this game is presented in the next subsection.

## 2.4 Game setup

In this work, we apply cooperative game theory to address the problem of graph coarsening. Cooperative game theory, as defined by (v Neumann & Morgenstern, 1953), is a mathematical framework in which a group of players interacts to achieve a common goal by forming coalitions based on binding agreements. We adopt this framework to model graph coarsening as a game, defined as $\text{CGame} = (N, v)$, where $N = \{1, 2, \ldots, p\}$ denotes the set of players (graph nodes), and $v$ is the characteristic function that assigns a transferable total cost value to each feasible coalition. Each player $i \in N$ has a feature vector $x_i$ and a neighborhood set $A_i$, consisting of all nodes connected to $i$. The members of the set $A_i$ will be the potential collaborators of player $i$ for coalition formation, from which a feasible subset can be selected based on cost and structural constraints. To model coalition dynamics and cost distribution in the CGame framework, we define the mapping profile, characteristic function, and player cost. The mapping profile is defined as a tuple $m = (m_1, m_2, \ldots, m_p)$, where each $m_i$ indicates the coalition $S_j \in \mathcal{S}$ to which node $i$ is assigned based on the merging decision. This mapping induces a set of supernodes $V_c = \{\tilde{V}_1, \tilde{V}_2, \ldots, \tilde{V}_k\}$, where $k$ is the number of distinct coalitions formed. The profile $m$ thus specifies the complete node-to-supernode assignment. For a particular node $i$, the mapping can be expressed as $(m_i, m_{-i})$, where $m_{-i}$ represents the mapping decisions of all nodes except $i$. The set of nodes assigned to coalition $S_i$ under mapping $m$ is denoted by $R_{S_i}^m$, and the coalition $S_i$ excluding node $j$ under mapping $m$ is represented by $R_{S_i}^m \setminus \{j\}$. The characteristic function $v$ of $R_{S_i}^m$ is defined as:

$$v(R_{S_i}^m) = \begin{cases} \text{DE}(R_{S_i}^m), & \text{if } |R_{S_i}^m| = 1 \\ \sum_{j \in R_{S_i}^m} c_{S_i}^m(j), & \text{if } |R_{S_i}^m| > 1 \end{cases} \tag{6}$$

$$\text{where, } \text{DE}(R_{S_i}^m) = \frac{1}{2} \sum_{(u,v) \in E_{S_i}} \tilde{w}_{uv} \|\tilde{x}_u - \tilde{x}_v\|^2, \quad c_{S_i}^m(j) = \text{DE}(R_{S_i}^m) - \text{DE}(R_{S_i}^m \setminus \{j\}) \tag{7}$$

Where, $DE(R_{S_i}^m)$ represents the contribution of Dirichlet energy of the $i$ th supernode or coalition $S_i$, formed under the mapping profile $m$ and $c_{S_i}^m(j)$ denotes the cost of node $j \in R_{S_i}^m$, defined as the marginal increase in Dirichlet energy due to its presence in coaliton $S_i$. $E_{S_i}$ denotes the set of edges connected to the $i$th supernode corresponding to coalition $S_i$ on the coarsened graph. Since each edge is incident to exactly two supernodes, the factor $\frac{1}{2}$ guarantees that each edge contributes exactly once to the global energy.

Given the characteristic function $v$ and the individual player cost $c(j)$ in terms of cost, we now analyze the strategic behavior of players to reach the convergence state of the game, or a stable coalition structure. A coalition structure is considered stable if it satisfies two conditions: internal stability and external stability.

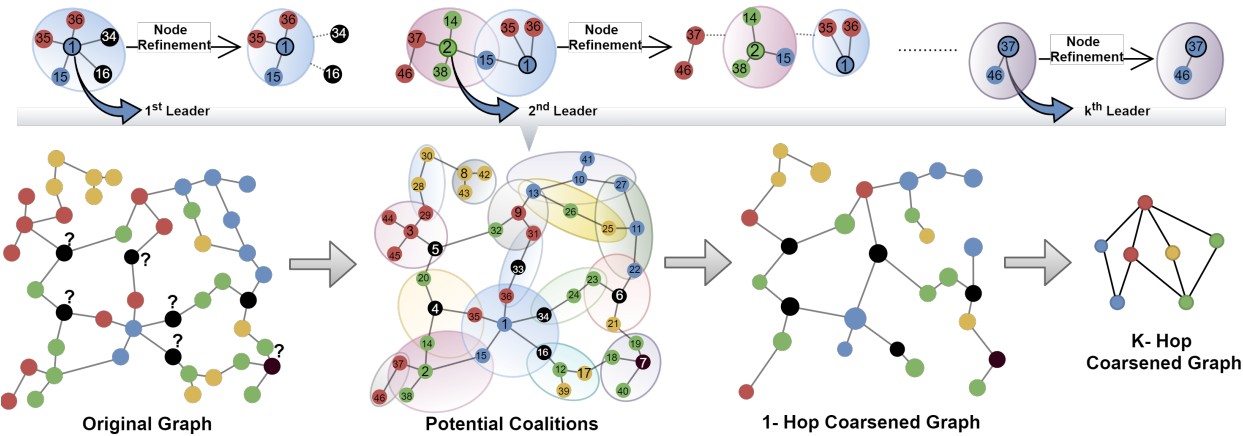

Figure 2: The figure illustrates the step-by-step process of cooperative game-based graph coarsening. It depicts coalition formation based on degree (as leaders), followed by refinement using the cost of each node within a coalition to achieve stability. The input to the CGC algorithm is the original graph $G(V, E, X, Y)$, where some nodes may not have label information.

A coalition is internally stable if no subset of its members can decrease their total allocated cost by deviating to form a new coalition, as expressed in equation (8).

$$\sum_{i \in T} c_i \leq v(T), \quad \forall T \subseteq S \tag{8}$$

A coalition is externally stable if no player has an incentive to deviate from their current coalition to join another feasible coalition. Let $\Gamma_i$ represent the set of feasible coalitions available to the player $i$, that is, coalitions for which the player's inclusion does not increase their cost or violate coalition constraints. Then, the optimal strategy for the player $i$, denoted by $m_i^*$, is defined as the best response that minimizes their individual cost, as shown in equation (9).

$$m_i^* = \arg \min_{m_i \in \Gamma_i} c_i(m_i, m_{-i}) \tag{9}$$

where $c_i$ denotes the cost allocated to player $i$, which can be expressed as a function $c_i(m_i, m_{-i})$ representing the cost when player $i$ selects the mapping $m_i$ while others choose a fixed profile $m_{-i}$. Here, $v(T)$ is the value of subset $T$ given by the characteristic function. In the next subsection, we present the CGC algorithm that ensures convergence to this equilibrium state.

## 3 ALGORITHM DEVELOPMENT

In CGame, each player joins a coalition to minimize individual cost with respect to the current coalition structure. The contribution of each player within a coalition is then computed. Players follow best-response decisions, leading the system to converge toward a stable coalition structure where no player has an incentive to deviate. To implement this process, we introduce the cooperative game-based graph coarsening (CGC) algorithm.

Given a graph $G = (V, E, X)$ with a weighted adjacency matrix $W \in \mathbb{R}^{|V| \times |V|}$ and node features $X \in \mathbb{R}^{|V| \times n}$, CGC algorithm employs an iterative procedure to minimize the cost of each player and converge to a stable coalition structure $S$ (i.e., a partition of the node set $V$ into disjoint coalitions) as described in algorithm 1. The algorithm begins by initializing each node as a singleton coalition, where its Dirichlet-energy cost $DE$ is computed according to equation (7) (line 2). The set S_coalition is initialized as empty to track nodes already assigned to multi-node coalitions, preventing them from being reconsidered as leaders. The set leaderSet stores the nodes selected as coalition leaders during the first iteration and is subsequently reused in later iterations. The array c_cost stores the initial $DE$ values for each singleton node, while c_update holds the updated player costs during the evaluation of candidate coalitions $S_i$.

In the coalition formation step, we begin by forming a local coalition with a node and its neighbors. To check the stability of this coalition, we evaluate whether any individual node or group of nodes within the coalition has an incentive to deviate and form a separate coalition that would reduce their cost as defined in equation (7).

**Algorithm 1** Cooperative game-based graph coarsening (CGC) using one-hop neighbor exploration

1: **Input:** Graph $G(V, E, X)$
2: Initialize: $c_{\text{cost}} \leftarrow \{ i : DE[i], \ \forall i \in V \}$, $c_{\text{update}} \leftarrow c_{\text{cost}}$, $S \leftarrow \{\{i\}, \forall i \in V\}$, $S_{\text{coalition}} \leftarrow \emptyset$, leaderSet $\leftarrow \emptyset$
3: Sort $V$ by degree in descending order
4: **repeat**
5:    $S_{\text{prev}} \leftarrow S$
6:    **if** $leaderSet \neq \emptyset$ **then** $V \leftarrow leaderSet$
7:    **for** each node $i \in V$ **do**
8:      **if** $i \notin S_{\text{coalition}}$ **and** $i$ has edges **then**
9:       $S_i \leftarrow \{i\} \cup$ neighbors$(i)$
10:       Compute each node cost $c_{S_i}(j)$ for all $j \in S_i$ (equation (7)) and
11:       set coalition cost $v(S_i) = \sum_{j \in S_i} c_{S_i}(j)$ (equation (6)).
12:       **if** $v(S_i) < \sum_{j \in S_i} c_{\text{cost}}(j)$ **then**
13:         **for** each node $j \in S_i$ **do**
14:           **if** $c_{S_i}(j) > c_{\text{update}}(j)$ **then** Remove $j$ from $S_i$
15:         **end for**
16:       **else** $S_i \leftarrow \emptyset$
17:      **end if**
18:      **if** $|S_i| > 1$ **then** $leaderSet \leftarrow leaderSet \cup \{i\}$, $S_{\text{coalition}} \leftarrow S_{coalition} \cup S_i$, update $S$, $c_{\text{update}}$
19:      **end if**
20:    **end for**
21: **until** $S = S_{\text{prev}}$
22: **Output:** Loading matrix C (constructed based on S)

This requires checking whether any subset of the current coalition can deviate as a best response to obtain a lower cost. However, exhaustively evaluating all subsets is computationally expensive for large coalitions.

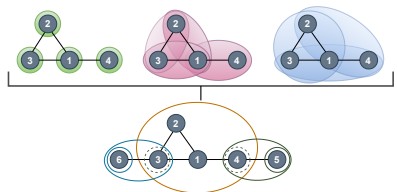 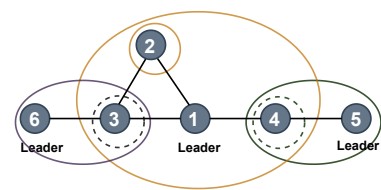

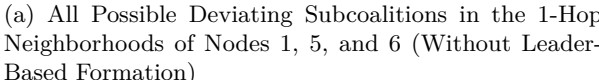

(a) All Possible Deviating Subcoalitions in the 1-Hop Neighborhoods of Nodes 1, 5, and 6 (Without Leader-Based Formation)

(b) All Possible Deviating Subcoalitions in the 1-Hop Neighborhoods of Nodes 1, 5, and 6 (Leader-Based Formation)

Figure 3: Comparison of possible internal deviations in coalitions formed using leaderless and leader-based methods. Circles indicate all possible subgroups that can break the coalition, while dotted circles highlight nodes that may deviate into alternate coalitions.

To improve convergence speed and emphasize structurally important nodes, we incorporate a leader-based coalition formation strategy. In this strategy, a coalition is only valid if it includes a designated leader; otherwise, it is discarded. The nodes with the highest degree of centrality are prioritized as leaders, ensuring that highly connected nodes initiate the formation of coalitions.

The formation process commences by sorting the nodes in descending order of degree (line 3), since high-degree nodes are more influential in the network, and their early consideration ensures that major connectivity hubs are prioritized during coalition formation. Then each node is sequentially evaluated as a potential leader. A potential leader attempts to form a preliminary coalition by selecting its immediate one-hop neighbors as $S_i$. The feasibility of a potential coalition $S_i$ is evaluated by checking whether $v(S_i) \leq \sum_{i \in S_i} v(\{i\})$, i.e., its total cost is less than or equal to the sum of the costs of its members when acting individually. If $S_i$ is not feasible, the potential leader is retained as a singleton coalition, and the algorithm proceeds to the next leader. Otherwise, if $S_i$ is feasible, the tentative coalition undergoes a refinement phase (lines 17–21), where each node's individual cost is compared against its previous configuration, whether as a singleton or as part of another coalition. Only nodes whose cost is reduced by joining the coalition are retained. If the refined coalition contains more than one member, it is accepted, and the initiating node is confirmed as the coalition leader (lines 17). The next leader is selected from the remaining nodes in the sorted list, skipping those already assigned to an existing multinode coalition (line 8).

In non-leader-based coalition refinement, the problem involves exploring all possible subsets of a coalition to check for deviations. We consider the Stable Coalition problem, which asks whether there exists a coalition satisfying the internal stability condition, as defined in equation (8). We prove in Theorem 1 that this problem is NP-complete. To illustrate the efficiency of our approach, consider the example shown in Figure 3a. In the leader-based coalition refinement, we adopt a heuristic approach to assess the Stable Coalition condition. Checking the stability of a single coalition requires $O(\Delta)$ comparisons (one for each neighboring node). Checking all $k$ coalitions requires $O(k\Delta)$ total comparisons, where $\Delta$ is the average degree and $k$ is the number of coalitions. This improvement is illustrated in Figure 3b, with further details provided in Appendix A.2. The performance in leader-based coalition formation will not be affected. Once the coalition cost is distributed among players, each player can unilaterally deviate under best-response dynamics whenever it is beneficial. Moreover, each such deviation decreases the overall system cost, as demonstrated in the next subsection, where we prove that this game is a potential game.

**Theorem 1.** *The* Stable Coalition *problem is NP-complete.*

**Proof.** The proof of Theorem 1 has been given in Appendix A.1

The cost allocated to an individual node $j$ within a candidate coalition $S_i$, denoted by $c_{S_i}(j)$, is computed using the Dirichlet energy based marginal contribution defined in equation (7), which measures the incremental impact of node $j$ on the total value of the coalition (line 10). After computing the costs, each node finds its best mapping decision according to equation (8). If its cost within the coalition exceeds the cost from its previous configuration, it is removed from the coalition. This ensures that participation is beneficial and rational for all members of the final coalition structure. Finally, during the update of the coalition structure

$S$, all nodes in $S_i$ are removed from the existing coalition set $S$, and the refined coalition $S_i$ is inserted back into $S$ as a new coalition and `c_update` stores the updated player costs inside the candidate coalition $S_i$. As per algorithm 1, the entire process is repeated iteratively until the coalition structure $S$ reaches equilibrium, where no further beneficial changes in the mapping decision are possible, with $S_{\text{prev}}$ denoting the coalition structure from the previous iteration used to check convergence. At this point, the configuration is considered stable as no node has an incentive to switch coalitions, leave, or form new coalitions.

The proposed method dynamically determines the final size of the coarsened graph based on the number of coalitions formed at the stable state. To enable deeper reductions while preserving structural locality, we extend our approach via a recursive multi-level strategy. In this setting, the coarsened graph from one iteration of the CGC algorithm 1 becomes the input for the next. In this setting, the coarsened graph from one iteration of the CGC algorithm 1 becomes the input for the next. The features of the coarsened graph at each level is calculated using $X_c = PX$. Although each level performs only one-hop neighborhood exploration, the overall effect of $k$ such iterations achieves an effective $k$-hop coarsening of the original graph. The algorithm for multi-level coarsening is provided in Appendix A.4.

## 4 ANALYSIS OF THE ALGORITHM

### 4.1 Theoretical Analysis

This section analyzes the convergence of the CGC algorithm to an equilibrium state (where no coalition can reduce the cost by unilateral deviation) within CGame. To assess the existence of an equilibrium, it is essential to examine the convergence behavior of the CGC algorithm. This can be achieved by demonstrating that the CGC algorithm of CGame constitutes a potential game. Before presenting the proof, we first introduce the concept of a potential game and define the potential function used in our analysis.

Table 1: Performance comparison across different GNN models (GCN, GAT, APPNP) on datasets including Cora, DBLP, CoCS, Pubmed, Co-Physics, Genius, and OGBN-Arxiv, under various coarsening ratios using the proposed CGC algorithm. Each value represents the mean accuracy $\pm$ standard deviation over 10 runs.

| Dataset | r=k/p | GCN | GAT | APPNP |
|---|---|---|---|---|
| Cora | 0.21 | 87.96 ± 0.05 | 86.74 ± 0.02 | 89.07 ± 0.03 |
| | 0.05 | 80.57 ± 0.07 | 80.95 ± 0.04 | 82.00 ± 0.02 |
| DBLP | 0.23 | 84.75 ± 0.12 | 84.00 ± 0.06 | 84.97 ± 0.08 |
| | 0.05 | 80.74 ± 0.06 | 83.29 ± 0.05 | 83.43 ± 0.04 |
| | 0.01 | 76.77 ± 0.08 | 78.45 ± 0.06 | 80.24 ± 0.05 |
| CoCS | 0.14 | 92.53 ± 0.09 | 89.36 ± 0.07 | 93.47 ± 0.05 |
| | 0.03 | 91.13 ± 0.03 | 88.34 ± 0.04 | 91.15 ± 0.02 |
| | 0.01 | 85.66 ± 0.05 | 83.82 ± 0.06 | 88.27 ± 0.03 |
| Pubmed | 0.235 | 86.84 ± 0.04 | 78.63 ± 0.05 | 85.22 ± 0.06 |
| | 0.07 | 86.02 ± 0.10 | 76.20 ± 0.08 | 85.74 ± 0.07 |
| | 0.02 | 83.41 ± 0.09 | 78.31 ± 0.07 | 83.99 ± 0.05 |
| Co-Physics | 0.09 | 94.79 ± 0.11 | 92.11 ± 0.10 | 95.85 ± 0.08 |
| | 0.01 | 94.15 ± 0.02 | 93.51 ± 0.03 | 95.14 ± 0.01 |
| | 0.005 | 92.43 ± 0.07 | 92.03 ± 0.05 | 93.94 ± 0.04 |
| Genius | 0.06 | 80.00 ± 0.03 | 79.84 ± 0.34 | 79.78 ± 0.51 |
| | 0.02 | 79.97 ± 0.08 | 79.98 ± 0.04 | 79.96 ± 0.09 |
| | 0.009 | 79.91 ± 0.14 | 80.00 ± 0.00 | 79.15 ± 1.34 |
| | 0.007 | 78.63 ± 0.03 | 80.00 ± 0.00 | 79.60 ± 0.73 |
| OGBN-Arxiv | 0.15 | 53.83 ± 0.23 | 51.14 ± 0.13 | 51.07 ± 0.20 |
| | 0.04 | 52.81 ± 0.33 | 49.87 ± 0.25 | 49.92 ± 0.28 |
| | 0.01 | 50.91 ± 0.42 | 48.06 ± 0.28 | 47.93 ± 0.58 |
| | 0.004 | 46.86 ± 0.50 | 45.21 ± 0.28 | 45.13 ± 0.58 |
| | 0.001 | 42.01 ± 0.56 | 40.44 ± 0.58 | 39.25 ± 1.29 |

**Definition 1. Potential Game** *A cooperative game can be regarded as a potential game if there exists a global scalar function $\Phi$, called the potential function, that captures the alignment of individual cost changes with a collective objective. Formally, for any player $i \in N$, and any two mapping decisions $m_i, m_i' \in A_i$, with a fixed mapping configuration of other players $m_{-i} \in \prod_{j \neq i} A_j$, the following implication holds:*

$$c_i(m_i', m_{-i}) - c_i(m_i, m_{-i}) \leq 0 \quad \Rightarrow \quad \Phi(m_i', m_{-i}) - \Phi(m_i, m_{-i}) \leq 0 \tag{10}$$

*where, $c_i$ represents the cost incurred by node $i$. This condition ensures that a reduction in an individual node's cost corresponds to a non-increasing global potential, thereby aligning local choices with a global optimization goal.*

The total potential of the CGame can be evaluated by aggregating the individual potentials of each coalition. When a node is mapped to a coalition, it incurs a cost $c$, which contributes to the coalition's potential. Accordingly, we compute the potential of each coalition. The potential of coalition $S_i$ under mapping profile $m$, denoted by $\phi(R_{S_i}^m)$, is defined as follows:

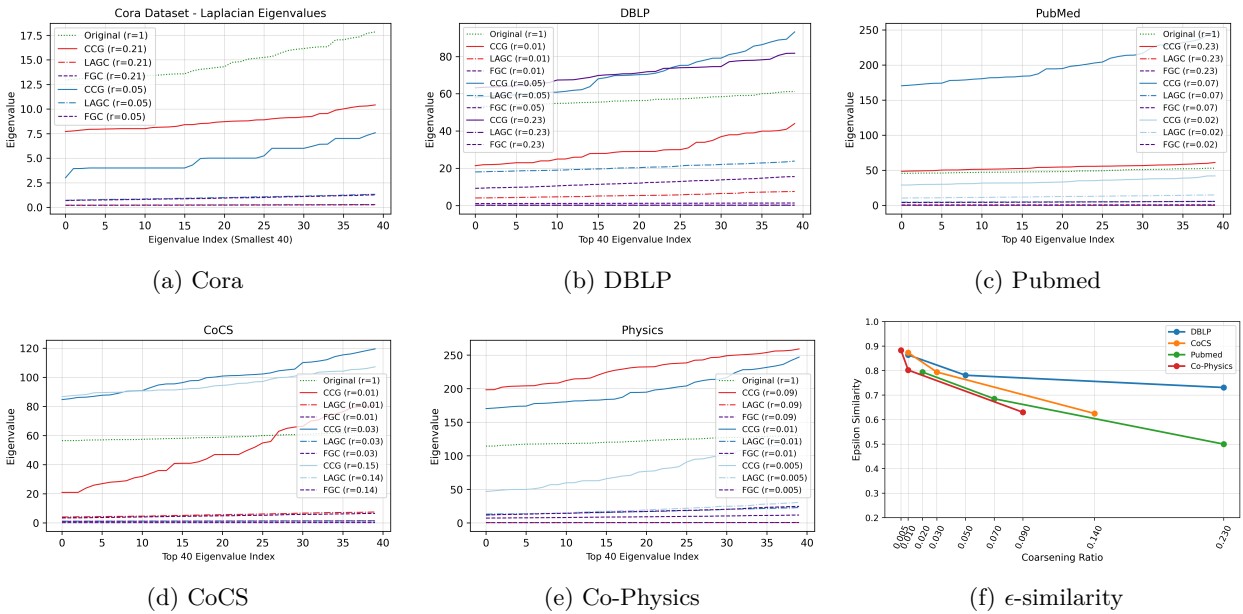

(a) Cora

(b) DBLP

(c) Pubmed

(d) CoCS

(e) Co-Physics

(f) $\epsilon$-similarity

Figure 4: This figure plots the top 40 eigenvalues of the Laplacian matrix for both the original and coarsened graphs across various datasets: (a) Cora, (b) DBLP, (c) PubMed, (d) CoCS, and (e) Co-Physics, under different coarsening ratios. Figure (f) shows the corresponding $\epsilon$-similarity values for the same datasets. The values of $\epsilon$, which range from 0 to 1, shows the similarity between the original graph $G$ and the coarse graph $G_c$ generated by the proposed CGC algorithm, with lower values implying a higher structural similarity.

**Calculation of Coalition-Level Potential:** Given a coalition $S_i$ under a specific mapping profile $m$, its potential contribution $\phi(R_{S_i}^m)$ is defined as:

$$\phi(R_{S_i}^m) = \text{DE}(R_{S_i}^m) = \frac{1}{2} \sum_{(u,v) \in E_{S_i}} \tilde{w}_{uv} \|\tilde{x}_u - \tilde{x}_v\|^2, \tag{11}$$

where $E_{S_i}$ denotes the set of edges connected to the $i$th supernode (coalition $S_i$) on the coarsened graph, $\tilde{w}_{uv}$ is the weight on edge $(u,v)$, and $\tilde{x}_u, \tilde{x}_v$ are feature vector of supernodes $u$ and $v$ under $m$.

**Global Potential $\Phi(m)$:** Let $\mathcal{S} = \{S_1, S_2, \ldots, S_k\}$ represent the set of all coalitions. This global potential aggregates the contributions of all coalitions, providing a holistic measure of the graph reduction configuration. It coincides with the Dirichlet energy objective given in equation (5). Since each coarsened edge is incident to exactly two supernodes, the factor $\frac{1}{2}$ ensures correct edge accounting when coalition-level potentials are aggregated. The total potential of CGame under a mapping profile $m$ is given by:

$$\Phi(m) = \sum_{S_i \in \mathcal{S}} \phi(R_{S_i}^m) = \sum_{S_i \in \mathcal{S}} \left( \frac{1}{2} \sum_{(u,v) \in E_{S_i}} \tilde{w}_{uv} \|\tilde{x}_u - \tilde{x}_v\|^2 \right) = \sum_{\substack{S_i, S_j \in \mathcal{S} \\ i < j}} \tilde{w}_{ij} \|\tilde{x}_i - \tilde{x}_j\|^2. \tag{12}$$

where, $\phi(R_{S_i}^m)$ is the Coalition-Level Potential as defined in equation (11)

**Theorem 2.** *The function $\Phi(m)$ satisfies the conditions of a potential function. Therefore, CGame constitutes a potential game.*

**Proof.** The proof of Theorem 2 has been given in Appendix A.7

**Theorem 3.** *At least one Pure Nash Equilibrium (PNE) is ensured in the proposed CGame framework when operating under the CGC algorithm.*

**Proof.** The proof of Theorem 3 has been given in Appendix A.7

Table 2: Node classification accuracy on various real-world datasets using the proposed CGC algorithm, compared against state-of-the-art methods SCAL, FGC, and LAGC. Experiments are conducted under different coarsening ratios ($r = k/p$). Each value represents the mean accuracy $\pm$ standard deviation computed over 10 runs. The results demonstrate that CGC consistently achieves superior performance across a wide range of datasets and coarsening levels. OOM indicates cases where execution exceeded memory.

| Dataset | $r = k/p$ | SCAL | FGC | LAGC | CGC | Whole Data |
|---|---|---|---|---|---|---|
| CORA | 0.21 | 80.08 ± 1.02 | 83.33 ± 0.02 | 84.01 ± 0.13 | 87.96 ± 0.32 | 89.50 ± 1.20 |
| | 0.05 | 54.07 ± 0.94 | 67.98 ± 1.77 | 71.73 ± 0.13 | 80.57 ± 0.21 | |
| DBLP | 0.23 | 75.09 ± 1.59 | 81.47 ± 0.13 | 81.31 ± 0.75 | 84.75 ± 0.54 | 85.35 ± 0.86 |
| | 0.05 | 76.38 ± 1.83 | 74.75 ± 0.10 | 75.53 ± 0.65 | 80.74 ± 0.43 | |
| | 0.01 | 68.70 ± 4.50 | 69.28 ± 0.06 | 69.90 ± 0.69 | 76.77 ± 0.12 | |
| CO-CS | 0.14 | 81.41 ± 3.15 | 91.37 ± 0.00 | 90.01 ± 0.82 | 92.53 ± 0.67 | 93.32 ± 0.62 |
| | 0.03 | 50.46 ± 5.43 | 78.74 ± 0.04 | 84.71 ± 0.25 | 91.13 ± 0.31 | |
| | 0.01 | 25.20 ± 7.32 | 83.13 ± 0.01 | 77.29 ± 0.13 | 85.66 ± 0.45 | |
| PUBMED | 0.235 | 78.02 ± 0.48 | 81.71 ± 0.09 | 83.21 ± 0.71 | 86.84 ± 0.11 | 88.89 ± 0.57 |
| | 0.07 | 73.73 ± 0.90 | 69.37 ± 0.06 | 77.24 ± 0.81 | 86.02 ± 0.28 | |
| | 0.02 | 64.71 ± 4.68 | 71.10 ± 1.93 | 76.37 ± 0.05 | 83.41 ± 0.72 | |
| CO-PHYSICS | 0.09 | 90.68 ± 1.03 | 87.25 ± 0.40 | 87.11 ± 0.03 | 94.79 ± 0.19 | 96.22 ± 0.74 |
| | 0.01 | 31.08 ± 10.02 | 91.38 ± 0.12 | 91.11 ± 0.19 | 94.15 ± 0.37 | |
| | 0.005 | 24.92 ± 2.97 | 88.01 ± 0.29 | 89.26 ± 0.03 | 92.43 ± 0.58 | |
| Genius | 0.06 | 79.99 ± 0.11 | OOM | OOM | 80.00 ± 0.03 | 86.71 ± 0.12 |
| | 0.02 | 80.01 ± 0.09 | OOM | OOM | 79.97 ± 0.08 | |
| | 0.009 | 79.98 ± 0.09 | OOM | OOM | 79.91 ± 0.14 | |
| | 0.007 | 79.99 ± 0.11 | OOM | OOM | 78.63 ± 0.03 | |
| OGBN-Arxiv | 0.15 | 27.50 ± 1.41 | OOM | OOM | 53.83 ± 0.23 | 56.24 ± 0.22 |
| | 0.04 | 7.55 ± 0.25 | 16.05 ± 0.06 | 29.79 ± 0.09 | 52.81 ± 0.33 | |
| | 0.01 | 5.86 ± 0.00 | 31.00 ± 0.20 | 32.37 ± 0.22 | 50.91 ± 0.42 | |
| | 0.004 | 5.86 ± 0.00 | 22.19 ± 0.29 | 30.45 ± 0.88 | 46.86 ± 0.50 | |
| | 0.001 | 5.86 ± 0.00 | 19.34 ± 0.01 | 25.71 ± 0.68 | 42.01 ± 0.56 | |

## 4.2 Numerical Analysis

In this section, we first describe the experimental setup, a detailed overview of the datasets, followed by analyzing the proposed algorithm's experimental results on real graph data sets which evaluate its performance across various graph neural network models, and then compare it with state-of-the-art methods to showcase the effectiveness of the CGC algorithm.

**Experimental Setup:** Experiments were conducted on a system running `Ubuntu 18.04.6 LTS (x86_64)` with dual `Intel Xeon E5-2630 v4` CPUs (40 logical cores, $2.20\,$GHz) and $247\,$GB RAM.

**Dataset:** We have evaluated our method on real-world datasets, including Cora, DBLP, CoCS, PubMed, Co-Physics, OGBN-Arxiv, and Genius. Detailed data set statistics are provided in Table A.5 in the appendix.

**Baseline models**: We evaluate the performance of the CGC algorithm (proposed) by its output as a coarsened graph as the input for the GNN model for node classification task. To ensure consistency, we adopt GNN architectures similar to those in the baseline study, including GCN (Kipf & Welling, 2016), GAT (Veličković et al., 2017), and APPNP (Gasteiger et al., 2018). We compare the performance of the proposed CGC algorithm with state-of-the-art methods, including SCAL (Huang et al., 2021), FGC (Kumar et al., 2023), and LAGC (Kumar et al., 2024), in terms of node classification accuracy. Experiments conducted on real-world datasets demonstrate that our model surpasses existing state-of-the-art methods in both node classification accuracy and computational efficiency.

**Node Classification:** For the node classification task, we first apply the proposed CGC algorithm to the original graph $G(\Theta, X)$ to obtain a coarsened graph $G_c(\Theta_c, X_c)$ without using any node labels. After obtaining $G_c$, we split the node labels on the original graph into 80% training labels($Y_{\text{train}}$) and 20% testing labels($Y_{\text{test}}$). The test labels $Y_{\text{test}}$ are completely masked and excluded during label projection. Concretely, we construct $Y_{\text{train}}$ by zeroing out the one-hot label rows of all test nodes, so $Y_{\text{train}}$ contains labels only for training nodes. We infer coarse labels via $Y_c = \arg\max(PY_{\text{train}})$ (Huang et al., 2021; Kumar et al., 2024; 2023), where $P = C^+$, ensuring that $Y_c$ depends exclusively on training labels with zero contribution from $Y_{\text{test}}$. We then train the Graph Neural Network (GNN) on $G_c(\Theta_c, X_c, Y_c)$ and evaluate performance on the 20% test nodes in the original graph, whose labels $Y_{\text{test}}$ were completely excluded during coarsening and are used only for final evaluation. Furthermore, we perform node classification on the original graph and compare the results with those obtained from the coarsened graph. To ensure an equitable comparison, we use the same data split, allocating 80% of the node labels for training and the remaining 20% for testing.

It is evident from Table 1 that the node classification performance remains consistent across different GNN models (GCN, GAT, APPNP) on datasets including Cora, DBLP, CoCS, PubMed, Co-Physics, Genius, and OGBN-Arxiv under various coarsening ratios, demonstrating the robustness and effectiveness of the proposed CGC algorithm. Furthermore, we compare the node classification performance of the GCN model using our proposed CGC method against state-of-the-art approaches including SCAL, FGC, and LAGC. The results, summarized in Table 2, demonstrate that CGC consistently outperforms these baseline methods across various datasets and coarsening ratios. We present robustness studies that complement the results reported in the main paper. These include additional baseline comparison, sensitivity analysis to leader selection, evaluation on heterophilous graphs, comparison with a random node-masking baseline, experiments under reduced label supervision, ablation studies on leader variants, and results on the ogbn-arxiv dataset using the official OGB splits. Overall, CGC shows stable performance across all settings with comprehensive results are in Appendix A.6.

**Structural Properties**: To assess the structural preservation of the coarsened graph, we evaluate spectral and smoothness properties using the Relative Eigenvalue Error (REE) and $\epsilon$-similarity metrics. **REE** is defined as $\frac{1}{q} \sum_{i=1}^{q} \frac{|\tilde{\lambda}_i - \lambda_i|}{\lambda_i}$, measures the spectral deviation between the original and coarsened Laplacian matrices $\Theta$ and $\Theta_c$ over the smallest $q$ eigenvalues, where $\lambda_i$ and $\tilde{\lambda}_i$ denote the $i$-th eigenvalues of $\Theta$ and $\Theta_c$, respectively. Lower REE indicates better spectral preservation. It is evident from Figure 4 that the plots of the top 40 eigenvalues of the Laplacian matrix for both the original and coarsened graphs across various datasets (a) Cora, (b) DBLP, (c) PubMed, (d) CoCS, and (e) Co-Physics under different coarsening ratios, closely approximate the original spectra and demonstrate improved alignment compared to state-of-the-art algorithms. For smoothness, we adopt a feature-dependent $\epsilon$-**similarity** notion: given the node-feature matrix $X$, we require $(1-\epsilon)\operatorname{tr}(X^\top \Theta X) \le \operatorname{tr}(\tilde{X}^\top \Theta_c \tilde{X}) \le (1+\epsilon)\operatorname{tr}(X^\top \Theta X)$, where $\operatorname{tr}(X^\top \Theta X)$ and $\operatorname{tr}(\tilde{X}^\top \Theta_c \tilde{X})$ denote the Dirichlet energy of

Table 3: Runtime comparison of the proposed CGC algorithm with baseline algorithms SCAL, FGC, and LAGC across various real-world datasets and coarsening ratios ($r = k/p$). Time(in seconds) includes only the graph coarsening step. The results demonstrate that CGC is computationally efficient and scales well across different datasets.

| Dataset | r | SCAL | FGC | LAGC | CGC |
|---|---|---|---|---|---|
| Cora | 0.21 | 2.28 | 18.90 | 9 | 0.35 |
| | 0.05 | 6.58 | 7.50 | 9 | 0.35 |
| DBLP | 0.23 | 85.02 | 2818.14 | 1980 | 14 |
| | 0.05 | 65.96 | 581.31 | 309 | 15 |
| | 0.01 | 64.27 | 169.28 | 84 | 15 |
| CoCS | 0.14 | 23.13 | 3398.42 | 1758 | 13 |
| | 0.03 | 27.38 | 704.16 | 357 | 15 |
| | 0.01 | 45.33 | 261.51 | 126 | 15 |
| Pubmed | 0.235 | 53.37 | 2609.87 | 4380 | 23 |
| | 0.07 | 59.75 | 640.52 | 569 | 24 |
| | 0.02 | 61.34 | 167.48 | 202 | 24 |
| Co-Physics | 0.09 | 58.98 | 6647.98 | 10078 | 58 |
| | 0.01 | 110.38 | 914.43 | 585 | 67 |
| | 0.005 | 155.63 | 634.21 | 422 | 67 |
| Genius | 0.06 | 17707 | - | - | 7019 |
| | 0.02 | 22732 | - | - | 9011 |
| | 0.009 | 24251 | - | - | 9613 |
| | 0.007 | 24521 | - | - | 9720 |
| OGBN-Arxiv | 0.15 | 3792 | - | - | 1503 |
| | 0.04 | 6506 | 84388 | 61262 | 2579 |
| | 0.01 | 7258 | 49139 | 58341 | 2877 |
| | 0.004 | 7946 | 15284 | 14172 | 2912 |
| | 0.001 | 7376 | 9677 | 8957 | 2924 |

this specific feature signal on the original and coarsened graphs, respectively. This definition measures how well the coarsened graph preserves the energy of the task-relevant feature matrix $X$, consistent with feature-dependent smoothness evaluation used in recent graph coarsening research Kumar et al. (2023). A smaller $\epsilon$ therefore indicates better preservation of the smoothness of $X$ under coarsening. It is evident from Figure 4(f) that the $\epsilon$-similarity values for the DBLP, CoCS, PubMed, and Co-Physics datasets across different coarsening ratios $r$ remain within the valid range $[0, 1]$, indicating reliable smoothness preservation.

**Time complexity:** Given a graph with $p$ nodes and an average degree $\Delta$, the time complexity of the proposed coarsening algorithm is $\mathcal{O}(p \cdot \Delta \cdot k)$, where $k$ denotes the number of nodes of the coarsened graph. The per-iteration cost and the convergence bound are provided in Appendix A.3. This computational efficiency arises from the fact that each node interacts only with its neighbors and that the coarsening process significantly reduces the size of the problem. As demonstrated in Table 3, the proposed CGC algorithm achieves faster execution times (in seconds) compared to the baseline methods such as SCAL,

FGC, and LAGC in most coarsening ratios. Entries marked "–" indicate baseline runs that failed due to out-of-memory (OOM).

## 5 CONCLUSION

This paper introduced the CGC algorithm, a cooperative game-theoretic approach to graph coarsening that utilizes node features and the weighted adjacency matrix. We defined the coalition cost using the Dirichlet energy. We measured each player's cost as its marginal contribution, which is the change in energy that results from including or excluding the player from the coalition. We formally proved that the CGC algorithm operates within the CGame framework as a potential game, guaranteeing convergence to a stable solution. Since the potential function is defined as the sum of Dirichlet energies of all coalitions, the final stable state corresponds to a minimum total energy configuration. Empirical results on node classification tasks show that our method achieves superior accuracy and significantly lower coarsening time than state-of-the-art baselines. Overall, the CGC algorithm provides a computationally efficient and effective solution for graph coarsening, demonstrating both theoretical soundness and practical advantages.

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

# A   Appendix

## A.1   NP-Completeness of Coalition Instability Detection in Non-Leader-Based Refinement

Recall that internal stability (Eq. equation 8) requires that for a given coalition $S$

$$\sum_{i \in T} c_i \ \leq \ v(T), \quad \forall T \subseteq S, \tag{13}$$

where $c_i$ denotes the cost allocated to player $i$ in the current coalition $S$, and $v(T)$ is the cost that subset $T$ would incur if it deviates and forms a standalone coalition.

If there exists some non-empty subset $T \subseteq S$ such that

$$\sum_{i \in T} c_i \ > \ v(T),$$

then the members of $T$ are collectively overpaying relative to their stand-alone worth and have an incentive to deviate; in this case, the coalition $S$ is *internally unstable*.

**Coalition Instability Detection (decision version).** *Given a finite set of players $S$ with allocated costs $\{c_i\}_{i \in S}$ and a characteristic function $v(\cdot)$, does there exist a non-empty subset $T \subseteq S$ such that the internal-stability inequality equation 13 is violated, i.e.,*

$$\sum_{i \in T} c_i \; > \; v(T) \, ?$$

*If such a subset exists, the members of $T$ have incentive to deviate and form their own coalition, making the original coalition $S$ unstable.*

We now show that this decision problem is NP-complete.

***Proof of Theorem 1.*** To prove that the Coalition Instability Detection decision problem $Q'$ is NP-complete, we must show two things: (1) $Q' \in$ NP and (2) a known NP-complete problem $Q$ can be reduced to $Q'$ in polynomial time.

**Step 1: Coalition Instability Detection is in NP.** A decision problem is in NP if, for every instance of the problem, a proposed solution (certificate) can be verified in polynomial time. For the Coalition Instability Detection decision problem, a certificate is a non-empty subset $T \subseteq S$ that is claimed to violate internal stability. To verify this certificate, we need to compute (i) the total cost $\sum_{i \in T} c_i$ and (ii) the sub-coalition cost $v(T)$, and then check whether the inequality

$$\sum_{i \in T} c_i \; > \; v(T)$$

holds.

In our CGC formulation, the per-player costs and the characteristic function are defined via the Dirichlet energy $DE()$ of a coalition on the coarsened graph ( equation 6 and equation 7). Let $G = (V, E)$ be the underlying fine graph with node features $\{x_u\}_{u \in V}$ and edge weights $\{w_{uv}\}_{(u,v) \in E}$. Each coalition $S \subseteq V$ induces a supernode on the coarsened graph, and we denote by $E_x(S)$ the set of edges *incident* to this supernode $x$ on the coarsened graph.

The Dirichlet energy contribution associated with coalition $S$ is then

$$\mathrm{DE}(S) = \frac{1}{2} \sum_{(u,v) \in E_x(S)} \tilde{w}_{uv} \, \|\tilde{x}_u - \tilde{x}_v\|^2.$$

Each term $\|\tilde{x}_u - \tilde{x}_v\|^2$ can be computed in $O(n)$ time, where $n$ is the feature dimension, and each edge in $E_x(S)$ is processed once. Hence, $\mathrm{DE}(S)$ can be computed in $O(|E_x(S)| \cdot n)$ time.

The cost allocated to player $j$ in a coalition $S$ is given by

$$c_S(j) \; = \; \mathrm{DE}(S) \; - \; \mathrm{DE}\big(S \setminus \{j\}\big),$$

where $\mathrm{DE}(S)$ and $\mathrm{DE}(S \setminus \{j\})$ are both defined using the corresponding incident-edge sets $E_x(S)$ and $E_x(S \setminus \{j\})$ on the coarsened graph. Thus, computing $c_S(j)$ requires two Dirichlet-energy evaluations and is therefore $O(|E_x(S)| \cdot n)$ for each player's cost. Given a fixed coalition $S$, the total cost of any subset $T \subseteq S$ is

$$\sum_{i \in T} c_S(i),$$

which can be obtained in $O(|T|)$ time once the costs $\{c_S(i)\}_{i \in S}$ are available.

The characteristic function $v(\cdot)$ used in our CGC game is defined in Eqs. equation 6 and equation 7; specializing those definitions to an arbitrary subset $T$ (corresponding to some coalition / supernode), we have

$$v(T) \;=\; \begin{cases} \mathrm{DE}(T), & \text{if } |T| = 1, \\ \sum_{j \in T} c_T(j), & \text{if } |T| > 1, \end{cases}$$

where
$$c_T(j) \ = \ \mathrm{DE}(T) \ - \ \mathrm{DE}\big(T \setminus \{j\}\big),$$

and each $\mathrm{DE}(\cdot)$ is evaluated using the corresponding incident-edge set $E_x(T)$ on the coarsened graph. As above, each Dirichlet energy term can be computed in $O(|E_x(T)| \cdot n)$ time, so all $c_T(j)$ for $j \in T$ can be obtained in $O(|T| \cdot |E_x(T)| \cdot n)$ time, and hence $v(T)$ is computable in polynomial time.

Therefore, for any proposed subset $T \subseteq S$, both $\sum_{i \in T} c_i$ (with $c_i = c_S(i)$) and $v(T)$ (as defined above) can be evaluated in time polynomial in the size of the input. The final comparison $\sum_{i \in T} c_i > v(T)$ is constant time. Hence, the Coalition Instability Detection decision problem $Q'$ admits a polynomial-time verification procedure and belongs to the class NP.

**Step 2: NP-hardness via reduction from a known NP-complete problem.** Let $Q$ denote the Max-Cut decision problem and let $Q'$ denote the Coalition Instability Detection decision problem defined above. We reduce from $Q$, which is NP-complete (Ben-Ameur et al., 2014), and give a polynomial-time many-one reduction $Q \leq_p Q'$.

**Max-Cut (decision version).** Given an undirected graph $G = (V, E)$ with nonnegative integer edge weights $\{w_{uv}\}_{(u,v) \in E}$ and a threshold $K \in \mathbb{N}$, the Max-Cut decision problem asks whether there exists a subset $U \subseteq V$ such that
$$\mathrm{cut}_G(U, V \setminus U) \ := \ \sum_{\substack{(u,v) \in E \\ u \in U, \, v \notin U}} w_{uv} \ \geq \ K.$$

**Reduction ($Q \leq_p Q'$).** Given an instance $(G, K)$ of $Q$, we construct an instance of $Q'$ as follows. Let $W = \sum_{(u,v) \in E} w_{uv}$ denote the total edge weight.

1. **Player set.** Define $S = V \cup \{p\}$, where $p$ is a dummy player.

2. **Characteristic function.** For any coalition $T \subseteq S$, let $U = T \cap V$ and define
$$v(T) \ := \ W - \mathrm{cut}_G(U, V \setminus U).$$
Since the Dirichlet energy of the indicator function $f_U$ satisfies $\mathcal{E}(f_U) = \mathrm{cut}_G(U, V \setminus U)$, this can be written as $v(T) = W - \mathcal{E}(f_U)$, i.e., an affine transformation of the cut-based Dirichlet energy.

3. **Costs.** For all $i \in V$, set $c_i = 0$, and for the dummy player set
$$c_p = W - K + \tfrac{1}{2}.$$

This construction is clearly computable in polynomial time in $|V| + |E|$.

**Equivalence.** We now show that the Max-Cut instance $(G, K)$ is a "yes" instance if and only if the constructed Coalition Instability Detection instance admits a violating coalition.

($\Rightarrow$) Suppose there exists $U^* \subseteq V$ with $\mathrm{cut}_G(U^*, V \setminus U^*) \geq K$. Consider the coalition $T = U^* \cup \{p\}$. Then
$$\sum_{i \in T} c_i = c_p = W - K + \tfrac{1}{2},$$
$$v(T) = W - \mathrm{cut}_G(U^*, V \setminus U^*) \ \leq \ W - K.$$

Hence $\sum_{i \in T} c_i \ = \ W - K + \tfrac{1}{2} \ > \ W - K \ \geq \ v(T)$, so $T$ violates internal stability.

($\Leftarrow$) Suppose there exists a coalition $T \subseteq S$ such that $\sum_{i \in T} c_i > v(T)$. If $p \notin T$, then $\sum_{i \in T} c_i = 0$, while $v(T) = W - \mathrm{cut}_G(T, V \setminus T) \geq 0$, so the inequality $\sum_{i \in T} c_i > v(T)$ cannot hold. Thus any violating coalition must contain $p$.

Let $T = U \cup \{p\}$ for some $U \subseteq V$. Then $\sum_{i \in T} c_i = c_p = W - K + \frac{1}{2}, \quad v(T) = W - \mathrm{cut}_G(U, V \setminus U)$. The violation condition becomes $W - K + \frac{1}{2} > W - \mathrm{cut}_G(U, V \setminus U)$, which simplifies to $\mathrm{cut}_G(U, V \setminus U) > K - \frac{1}{2}$. Since cut values are integers, this is equivalent to $\mathrm{cut}_G(U, V \setminus U) \geq K$, so $U$ is a feasible solution to the original MAX-CUT instance.

**Conclusion.** Since the COALITION INSTABILITY DETECTION problem $Q'$ is in NP (Step 1), and known NP-complete problem MAX-CUT $Q$ reduces to it in polynomial time (Step 2), it follows that $Q'$ is NP-complete.

$\square$

## A.2 Illustrative Example of Coalition Stability

As illustrated in Figure 3, consider a graph in which three coalitions $\mathcal{C}_1, \mathcal{C}_2, \mathcal{C}_3$ are formed based on the 1-hop neighborhoods of nodes 1, 5, and 6, respectively. Specifically, for node 1, its local neighborhood includes nodes 2, 3, and 4, resulting in the coalition $\mathcal{C}_1 = \{1, 2, 3, 4\}$. To analyze the stability of these coalitions, we consider two approaches: first, the case without leader-based refinement, and then the case with leader-based refinement.

In Non-Leader-Based Coalition Refinement, the stability of a coalition is evaluated by checking whether any individual node or group of nodes has an incentive to deviate. Evaluating stability in this way requires examining all possible non-empty subsets of the coalition. For $\mathcal{C}_1$, which contains four nodes, the maximum number of non-empty subsets to consider is $2^4 - 1 = 15$. However, as shown in Figure 3a, only 11 subsets are observed to deviate since $\mathcal{C}_1$ is not a fully connected clique. When the nodes in a coalition are fully connected, the deviation-checking process corresponds to the Stable Coalition problem, which is NP-complete, as established in Theorem 1.

In the leader-based coalition refinement, nodes $1, 5, 6$ are selected as leaders, as shown in Figure 3b, and form coalitions with all their neighbors. In the refinement step, each neighbor $a_i$ is individually checked to decide whether it should (i) remain in the leader's coalition, (ii) leave to join another coalition led by a different leader, or (iii) remain unassigned if no better option is available. This requires only 6 checks, one per neighbor, resulting in linear time complexity. This localized refinement procedure generalizes efficiently across the entire graph. Let $k$ denote the number of leaders (at most $k$), and $d_i$ the number of neighbors of leader $i$. The total number of comparisons to check stability across all leader-based coalitions is $O(\sum_{i=1}^{m} d_i)$. Since $\sum_{i=1}^{m} d_i \leq k \cdot \Delta$, where $\Delta$ is the average degree of the graph, the overall comparison cost is $O(k \cdot \Delta)$.

## A.3 Complexity and Convergence Analysis

**Per-iteration complexity.** One iteration corresponds to a full sweep over all leaders, that is, at most $k$ leader updates. Each leader considers at most $\Delta$ one-hop neighbors. Evaluating all marginal costs inside a candidate coalition requires $O(\Delta^2)$ computation, and checking membership or updating assignments adds $O(p\Delta)$. The per-iteration cost is therefore $O\big(k(\Delta^2 + p\Delta)\big)$. Since $p > \Delta$, the dominant term is $O(k\,p\,\Delta)$.

**Convergence bound.** CGC induces a finite exact potential game: every accepted coalition update produces the same decrease in the global potential as in the individual player's cost, and the potential is a non-negative Dirichlet-energy quantity. Hence the potential cannot decrease indefinitely, and only finitely

many improving deviations are possible, so the repeat–until loop converges to a pure Nash equilibrium (PNE) in finitely many steps.

More concretely, after the first full sweep over the $p$ nodes, the leader set stabilizes and contains exactly $k$ leaders. In the subsequent dynamics, only the remaining $p - k$ follower nodes may change their coalition, and each such move strictly decreases the Dirichlet-energy potential. Since each follower can choose among at most $k$ leaders, a follower can switch coalitions at most $(k-1)$ times. Hence, the total number of accepted coalition updates is bounded by $(p-k)(k-1) = O(pk)$. Under our update schedule, each iteration processes at most $k$ leaders, so the number of iterations until convergence is at most $O(pk/k) = O(p)$, while the total number of node-level updates is $O(pk)$, which in the worst case ($k \leq p$) is at most quadratic, $O(p^2)$, in the number of nodes.

## A.4 Multi-Level CGC Algorithm

---
**Algorithm 2** Multi-Level CGC Algortihm based Graph Coarsening
---
**Require:** Original graph $G = (V, E, X)$, number of coarsening levels $k$
**Ensure:** Coarsened graph $G^{(k)}$
 1: $G^{(0)} \leftarrow G$
 2: **for** $i = 1$ to $k$ **do**
 3:     Apply CGC algorithm 1 on $G^{(i-1)}$
 4:     Obtain coarsened graph $G^{(i)}$
 5: **end for**
 6: **return** $G^{(k)}$
---

## A.5 Additional Dataset Details

| Dataset | number of nodes | features | levels |
|---|---|---|---|
| Cora | 2,704 | 1,433 | 7 |
| DBLP | 17,716 | 1,639 | 4 |
| CoCS | 18,333 | 2,000 | 5 |
| Pubmed | 19,717 | 500 | 3 |
| Co-Physics | 34,493 | 8,415 | 6 |
| OGBN-Arxiv | 169343 | 128 | 40 |
| Genius | 421961 | 12 | 2 |

Table 4: Details of the real datasets, including the number of nodes $p$, features $n$, and levels $l$.

## A.6 Experiments

This section presents supplementary robustness studies that complement the results reported in the main paper. These include additional baseline comparison, sensitivity analysis to leader selection, evaluation on heterophilous graphs, comparison with a random node-masking baseline, experiments under reduced label supervision, ablation studies on leader variants, and results on the ogbn-arxiv dataset using the official OGB splits.

### A.6.1 Additional Baseline Comparison

To ensure fairness and completeness in our empirical evaluation, we additionally compare CGC against several recent and widely used graph coarsening baselines. Specifically, we include the recently proposed structure-guided SGBGC method (Xia et al., 2025), along with the scalable non-learning approaches HEM (Karypis &

Table 5: Node classification accuracy of CGC compared with the recent supervised coarsening method SGBGC and scalable non-learning baselines HEM, Graclus, and METIS across multiple real-world datasets and coarsening ratios ($r = k/p$). Each value reports the mean accuracy $\pm$ standard deviation over 10 runs.

| Dataset | $r = k/p$ | SGBGC | HEM | Graclus | METIS | CGC |
|---|---|---|---|---|---|---|
| Cora | 0.21 | 84.62 $\pm$ 0.34 | 79.88 $\pm$ 1.42 | 81.32 $\pm$ 0.31 | 83.31 $\pm$ 0.30 | 87.96 $\pm$ 0.32 |
| | 0.05 | 82.67 $\pm$ 0.44 | 30.21 $\pm$ 0.00 | 63.32 $\pm$ 2.21 | 30.21 $\pm$ 0.00 | 80.57 $\pm$ 0.21 |
| DBLP | 0.23 | 84.21 $\pm$ 0.50 | 75.29 $\pm$ 2.10 | 82.30 $\pm$ 0.16 | 83.31 $\pm$ 0.24 | 84.75 $\pm$ 0.54 |
| | 0.05 | 78.72 $\pm$ 0.26 | 66.77 $\pm$ 2.57 | 64.65 $\pm$ 0.41 | 53.18 $\pm$ 4.75 | 80.74 $\pm$ 0.43 |
| | 0.01 | 73.86 $\pm$ 0.25 | 44.71 $\pm$ 0.00 | 44.71 $\pm$ 0.00 | 44.71 $\pm$ 0.00 | 76.77 $\pm$ 0.12 |
| CoCS | 0.14 | 92.11 $\pm$ 0.44 | 85.76 $\pm$ 0.90 | 90.52 $\pm$ 0.64 | 90.94 $\pm$ 0.48 | 92.53 $\pm$ 0.67 |
| | 0.03 | 91.38 $\pm$ 0.28 | 22.56 $\pm$ 0.00 | 60.75 $\pm$ 2.61 | 23.77 $\pm$ 3.30 | 91.13 $\pm$ 0.31 |
| | 0.01 | 91.25 $\pm$ 0.17 | 22.56 $\pm$ 0.00 | 22.56 $\pm$ 0.00 | 22.56 $\pm$ 0.00 | 85.66 $\pm$ 0.45 |
| PubMed | 0.235 | 85.46 $\pm$ 0.20 | 82.04 $\pm$ 0.46 | 84.37 $\pm$ 0.42 | 82.97 $\pm$ 0.27 | 86.84 $\pm$ 0.11 |
| | 0.07 | 83.15 $\pm$ 0.15 | 57.50 $\pm$ 0.13 | 81.29 $\pm$ 0.22 | 57.30 $\pm$ 0.38 | 86.02 $\pm$ 0.28 |
| | 0.02 | 82.45 $\pm$ 0.31 | 39.27 $\pm$ 0.01 | 39.27 $\pm$ 0.01 | 39.17 $\pm$ 0.05 | 83.41 $\pm$ 0.72 |
| Co-Physics | 0.09 | 94.78 $\pm$ 0.43 | 81.57 $\pm$ 1.37 | 16.67 $\pm$ 0.00 | 16.67 $\pm$ 0.00 | 94.79 $\pm$ 0.19 |
| | 0.01 | 92.56 $\pm$ 0.14 | 50.52 $\pm$ 0.00 | 50.52 $\pm$ 0.00 | 50.52 $\pm$ 0.00 | 94.15 $\pm$ 0.37 |
| | 0.005 | 89.54 $\pm$ 0.22 | 50.52 $\pm$ 0.00 | 50.52 $\pm$ 0.00 | 50.52 $\pm$ 0.00 | 92.43 $\pm$ 0.58 |

Kumar, 1998a), Graclus (Dhillon et al., 2007), and METIS (Karypis & Kumar, 1998b). All baseline models are executed using their publicly available implementations with recommended hyperparameter configurations. Table 5 reports node-classification accuracy under varying coarsening ratios across multiple benchmark datasets. As shown in Table 5, CGC consistently outperforms all compared methods across datasets and coarsening levels, with particularly notable gains under stronger compression (smaller $r = k/p$).

### A.6.2 Sensitivity Analysis of CGC to Leader Selection

We evaluate the sensitivity of CGC to random leader selection on four benchmark datasets (Cora, DBLP, CoCS, and Pubmed) by running CGC multiple times with independently sampled leader sets, while keeping all other settings fixed. For each dataset, we perform multiple independent runs (150 for Cora, 50 for CoCS, and 35 for DBLP and Pubmed) and record the resulting node-classification accuracies. As summarized by the histograms in Figure 5a, 5b, 5c, and 5d, CGC exhibits strong stability under random leader selection across all four datasets. On Cora, 150 runs yield accuracies ranging from 85.19% to 87.53%, with a narrow spread of 2.34 percentage points. On DBLP, 35 runs fall within an even tighter band of 84.36%–85.05% (range: 0.69%). CoCS shows similar robustness, with 50 runs distributed between 91.80% and 92.77% (range: 0.97%), while Pubmed displays the highest stability, with 35 runs concentrated in the interval 87.13%–87.50% (range: 0.37%). The empirical distributions in Figure 5 reveal that the vast majority of runs cluster tightly around the mean accuracy, with very few outliers. This narrow concentration indicates that CGC's performance is highly robust and not strongly dependent on the specific random choice of initial leaders. Although the theoretical convergence guarantees apply only to local equilibria, these empirical results demonstrate that different leader selections consistently lead to similar high-quality solutions, suggesting that the local equilibria reached by CGC are of comparable quality across different starting configurations.

### A.6.3 Evaluation on Heterophilous Graphs

CGC is primarily designed for homophilic graph settings where neighboring nodes tend to share similar features and labels, consistent with the Dirichlet-energy objective that encourages merging structurally and semantically aligned nodes. Since the datasets used in our main experiments (e.g., Cora, Citeseer, PubMed, DBLP, Co-CS, Co-Physics, OGBN-Arxiv, Genius) are predominantly homophilic, we additionally evaluate CGC on two heterophilous benchmarks (Cornell and Texas) to assess robustness beyond this regime. We compare CGC against the classical coarsening baselines HEM, Graclus, and METIS under multiple coarsening ratios ($r = k/p$), reporting mean accuracy $\pm$ standard deviation over 10 runs. As shown in Table 6, CGC substantially outperforms all baselines across all coarsening levels, with the performance gap widening under stronger compression (smaller $r$), where traditional methods degrade to near-random accuracy ($\approx$

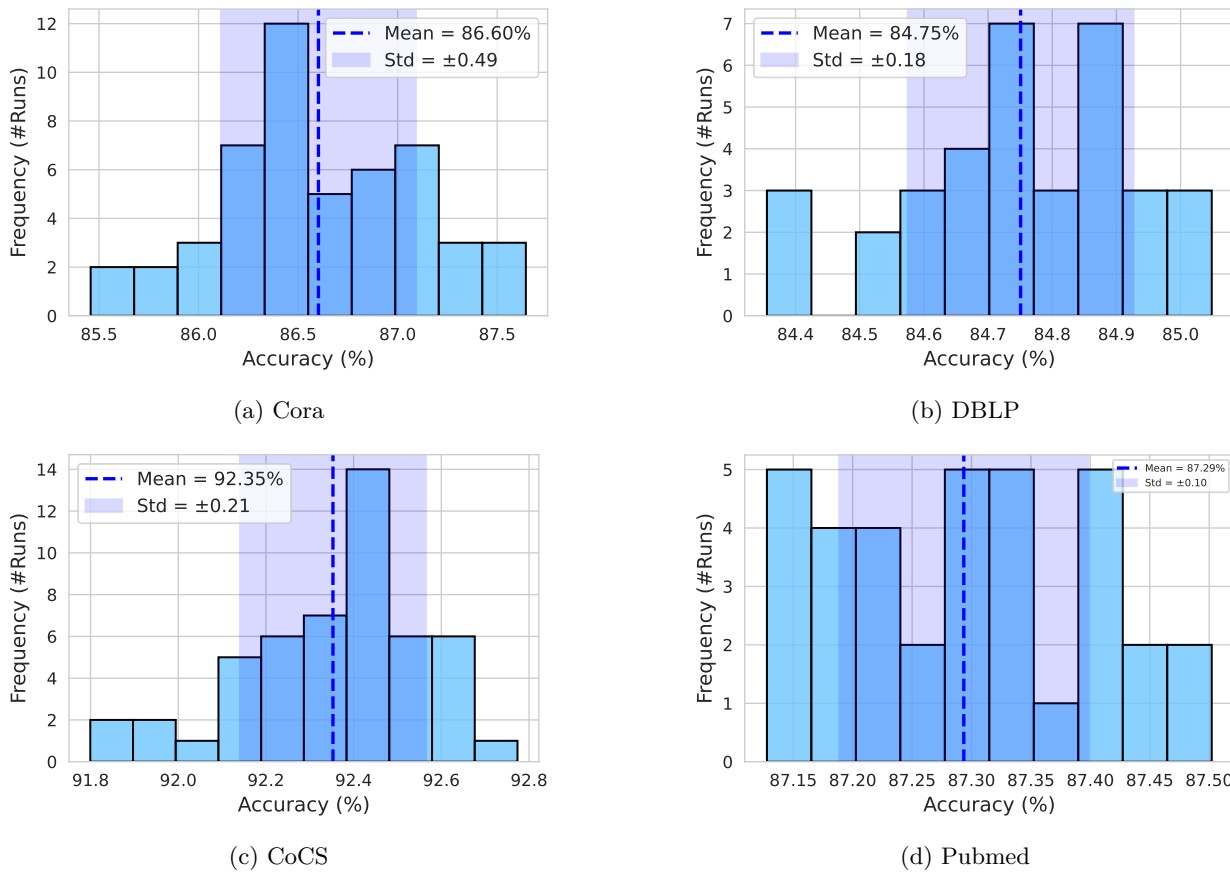

Figure 5: Distributions of node-classification accuracy across multiple runs of CGC under random leader selection, based on 150 runs for Cora, 50 for CoCS, and 35 for DBLP and Pubmed. Each histogram reports the sample mean ($\mu$) and standard deviation ($\sigma$), highlighting the tight concentration of accuracy values and the robustness of CGC to leader selection.

44–56%) while CGC maintains significantly higher accuracy (71–84%). These results demonstrate that CGC remains effective in heterophilous settings and preserves discriminative structural information even under aggressive graph reduction.

Table 6: Node classification accuracy on heterophilous graph datasets (Cornell and Texas), comparing the proposed CGC algorithm against classical graph coarsening baselines HEM, Graclus, and METIS under different coarsening ratios ($r = k/p$). Each value reports the mean accuracy $\pm$ standard deviation over 10 runs.

| **Dataset** | $r = k/p$ | **HEM** | **Graclus** | **METIS** | **CGC** | **Original** |
|---|---|---|---|---|---|---|
| Cornell | 0.23 | $54.10 \pm 4.25$ | $52.46 \pm 3.67$ | $53.88 \pm 2.38$ | $84.53 \pm 0.91$ | $86.90 \pm 0.56$ |
| | 0.06 | $44.81 \pm 0.00$ | $43.61 \pm 4.25$ | $44.81 \pm 0.00$ | $79.21 \pm 1.20$ | |
| | 0.02 | $44.81 \pm 0.00$ | $44.81 \pm 0.00$ | $44.81 \pm 0.00$ | $71.25 \pm 2.07$ | |
| Texas | 0.23 | $56.39 \pm 5.32$ | $61.31 \pm 2.96$ | $57.05 \pm 1.27$ | $84.53 \pm 0.91$ | $87.31 \pm 1.38$ |
| | 0.06 | $55.19 \pm 0.00$ | $40.11 \pm 12.08$ | $55.19 \pm 0.00$ | $79.21 \pm 1.20$ | |
| | 0.02 | $55.19 \pm 0.00$ | $47.76 \pm 14.86$ | $55.19 \pm 0.00$ | $71.25 \pm 2.07$ | |

### A.6.4 Random Node-Masking Baseline

To determine whether the performance gains of CGC arise solely from graph size reduction or from the quality of the coarsening mechanism, we implement a random node-masking baseline that removes nodes uniformly at random to match the coarsened graph size produced by CGC. We run this process 10 times with different random seeds and report mean accuracy $\pm$ standard deviation. As shown in Table 7, CGC consistently outperforms random masking across all coarsening ratios, confirming the value of the coalition-based merging strategy over naive downsampling.

Table 7: Node classification comparison of CGC with a random node-masking baseline under different coarsening ratios ($r = k/p$). Each value reports the mean accuracy $\pm$ standard deviation over 10 runs.

| Dataset | $r = k/p$ | Node Masking Baseline Accuracy | CGC |
|---|---|---|---|
| Cora | 0.21 | $44.63 \pm 6.10$ | $\mathbf{87.96 \pm 0.32}$ |
| | 0.05 | $22.84 \pm 9.67$ | $\mathbf{80.57 \pm 0.21}$ |
| DBLP | 0.23 | $77.92 \pm 1.33$ | $\mathbf{84.75 \pm 0.54}$ |
| | 0.05 | $70.16 \pm 3.61$ | $\mathbf{80.74 \pm 0.43}$ |
| | 0.01 | $55.10 \pm 16.99$ | $\mathbf{76.77 \pm 0.12}$ |
| Co-CS | 0.14 | $89.08 \pm 1.65$ | $\mathbf{92.53 \pm 0.67}$ |
| | 0.03 | $83.84 \pm 3.05$ | $\mathbf{91.13 \pm 0.31}$ |
| | 0.01 | $81.42 \pm 9.92$ | $\mathbf{85.66 \pm 0.45}$ |
| Pubmed | 0.235 | $56.01 \pm 7.60$ | $\mathbf{86.84 \pm 0.11}$ |
| | 0.07 | $36.92 \pm 14.14$ | $\mathbf{86.02 \pm 0.28}$ |
| | 0.02 | $31.12 \pm 17.03$ | $\mathbf{83.41 \pm 0.72}$ |
| Co-Physics | 0.09 | $92.01 \pm 1.72$ | $\mathbf{94.79 \pm 0.19}$ |
| | 0.01 | $90.36 \pm 3.44$ | $\mathbf{94.15 \pm 0.37}$ |
| | 0.005 | $79.13 \pm 7.25$ | $\mathbf{92.43 \pm 0.58}$ |

### A.6.5 Robustness to Limited Labeled Data

To study the robustness of CGC under reduced label availability, we perform additional experiments using smaller training subsets (30%, 40%, 60%, and 70% of labeled nodes), following the same settings as the main evaluation. The results, presented in Table 8, show that CGC maintains strong performance across reduced supervision levels, demonstrating that CGC remains effective even with significantly fewer labeled nodes.

### A.6.6 Ablation Studies on Leader Variants

To assess the impact of key design choices in CGC, we perform ablation studies on two variants: (i) Random-leader initialization and (ii) Random-neighbor merge under restricted hop sizes. In the Random-leader variant, coalition leaders are selected uniformly at random across 10 independent runs, and we report the mean node-classification accuracy with the corresponding standard deviation. The value in parentheses indicates the average coarsening ratio $r = k/p$ achieved across the 10 runs. In the Random-neighbor merge ablation, each node selects a random neighbor within a limited hop neighborhood for merging, instead of using Dirichlet-energy–based marginal contribution, thereby isolating the effect of informed coalition formation.

As shown in Table 9, both variants result in substantial accuracy degradation and increased variance relative to the proposed CGC method, demonstrating the critical role of energy-guided coalition formation and structured leader selection.

### A.6.7 Official OGB Splits

We additionally report the performance of CGC on the `ogbn-arxiv` benchmark using the official data splits provided by the Open Graph Benchmark (OGB). Table 10 summarizes the node-classification accuracy of CGC across a range of coarsening ratios $r = k/p$. The results indicate that CGC retains competitive performance even under aggressive coarsening, with accuracy decreasing gracefully as the ratio $r$ is reduced.

Table 8: Node classification accuracy of CGC under different training label ratios (70%, 60%, 40%, and 30%). Each value represents the mean accuracy ± standard deviation over 10 runs.

| Dataset | $r = k/p$ | 70% labels | 60% labels | 40% labels | 30% labels |
|---|---|---|---|---|---|
| Cora | 1 | 85.16 ± 0.51 | 85.59 ± 0.48 | 86.53 ± 0.58 | 86.99 ± 0.64 |
| | 0.2338 | 82.15 ± 0.90 | 78.67 ± 1.50 | 67.25 ± 3.12 | 56.76 ± 3.17 |
| | 0.0406 | 77.49 ± 1.48 | 75.76 ± 2.58 | 73.70 ± 2.49 | 65.65 ± 3.03 |
| DBLP | 1 | 85.73 ± 0.34 | 85.31 ± 0.28 | 84.91 ± 0.13 | 84.39 ± 0.19 |
| | 0.23 | 82.49 ± 0.29 | 80.06 ± 0.40 | 72.98 ± 0.53 | 66.47 ± 0.93 |
| | 0.05 | 79.54 ± 0.50 | 78.39 ± 0.74 | 76.65 ± 0.81 | 72.97 ± 0.98 |
| | 0.01 | 76.44 ± 0.89 | 76.13 ± 1.64 | 74.90 ± 1.46 | 72.60 ± 2.29 |
| CoCS | 1 | 93.97 ± 0.09 | 93.73 ± 0.24 | 91.56 ± 0.20 | 88.32 ± 0.20 |
| | 0.14 | 90.46 ± 0.47 | 89.26 ± 0.68 | 82.07 ± 2.83 | 76.87 ± 2.88 |
| | 0.03 | 89.77 ± 0.42 | 89.46 ± 0.70 | 87.29 ± 0.65 | 85.63 ± 1.74 |
| | 0.01 | 84.63 ± 0.98 | 84.20 ± 1.90 | 83.78 ± 1.60 | 83.06 ± 1.60 |
| PubMed | 1 | 87.87 ± 0.39 | 87.76 ± 0.27 | 86.48 ± 0.10 | 85.34 ± 0.16 |
| | 0.235 | 83.20 ± 0.62 | 77.16 ± 1.44 | 57.09 ± 1.84 | 30.27 ± 0.97 |
| | 0.07 | 81.47 ± 0.96 | 78.09 ± 1.43 | 62.50 ± 2.32 | 49.08 ± 2.62 |
| | 0.02 | 79.53 ± 1.41 | 77.17 ± 1.85 | 70.15 ± 1.94 | 65.48 ± 2.38 |
| Co-Physics | 1 | 96.56 ± 0.05 | 96.44 ± 0.08 | 96.30 ± 0.08 | 96.11 ± 0.07 |
| | 0.09 | 93.47 ± 0.40 | 92.21 ± 0.60 | 87.01 ± 2.32 | 82.01 ± 2.41 |
| | 0.01 | 93.50 ± 0.37 | 92.88 ± 0.62 | 90.32 ± 1.74 | 88.00 ± 2.46 |
| | 0.005 | 91.24 ± 1.84 | 91.36 ± 1.55 | 90.80 ± 1.98 | 90.48 ± 1.78 |
| Genius | 1 | 79.97 ± 1.05 | 80.00 ± 0.01 | 78.97 ± 3.05 | 72.74 ± 4.20 |
| | 0.06 | 78.82 ± 12.26 | 78.29 ± 4.81 | 77.26 ± 5.12 | 67.15 ± 23.16 |
| | 0.02 | 59.39 ± 21.09 | 48.73 ± 26.56 | 48.73 ± 26.56 | 45.86 ± 26.97 |
| OGBN-Arxiv | 1 | 55.30 ± 0.70 | 53.30 ± 0.17 | 53.03 ± 1.17 | 45.31 ± 0.42 |
| | 0.15 | 51.31 ± 0.41 | 48.43 ± 0.56 | 40.89 ± 0.73 | 22.66 ± 1.31 |
| Cornell | 1 | 86.99 ± 0.64 | 86.53 ± 0.58 | 85.59 ± 0.48 | 85.16 ± 0.51 |
| | 0.23 | 82.76 ± 1.27 | 82.76 ± 1.27 | 67.35 ± 2.20 | 57.49 ± 2.96 |
| | 0.06 | 78.99 ± 1.21 | 78.55 ± 1.54 | 74.75 ± 2.33 | 68.46 ± 2.05 |
| | 0.02 | 71.31 ± 1.96 | 70.32 ± 2.82 | 69.28 ± 3.60 | 67.11 ± 2.60 |
| Texas | 1 | 86.91 ± 0.90 | 86.72 ± 0.71 | 85.75 ± 0.59 | 85.22 ± 0.57 |
| | 0.23 | 82.76 ± 1.27 | 79.21 ± 1.26 | 67.35 ± 2.20 | 57.49 ± 2.96 |
| | 0.06 | 78.99 ± 1.21 | 78.55 ± 1.54 | 74.75 ± 2.33 | 68.46 ± 2.05 |
| | 0.02 | 71.31 ± 1.96 | 70.32 ± 2.82 | 69.28 ± 3.60 | 67.11 ± 2.60 |
| Squirrel | 1 | 40.25 ± 3.51 | 31.18 ± 2.50 | 23.95 ± 0.83 | 21.35 ± 0.44 |
| | 0.16 | 29.33 ± 0.71 | 28.42 ± 0.92 | 27.71 ± 0.90 | 26.03 ± 0.85 |
| | 0.01 | 30.17 ± 1.72 | 28.81 ± 2.13 | 28.24 ± 1.63 | 29.33 ± 1.39 |
| | 0.004 | 22.95 ± 2.13 | 22.07 ± 1.42 | 22.34 ± 1.32 | 23.08 ± 2.35 |

Table 9: Ablation study of random-leader and random-neighbor variants. Values show mean accuracy and ± standard deviation over 10 runs; parentheses denote the average coarsening ratio $r = k/p$.

| Dataset | $r = k/p$ | Random-leader (r) | Random-neighbor merge | CGC (Desc) | Whole Data |
|---|---|---|---|---|---|
| Cora | 0.21 | 86.90 ± 0.92 (0.27) | 83.03 ± 0.12 | 87.96 ± 0.32 | 89.50 ± 1.20 |
| | 0.05 | 79.09 ± 1.33 (0.09) | 62.35 ± 3.23 | 80.57 ± 0.21 | |
| DBLP | 0.23 | 83.45 ± 0.33 (0.26) | 82.45 ± 0.16 | 84.75 ± 0.54 | 85.35 ± 0.86 |
| | 0.05 | 80.24 ± 0.16 (0.06) | 78.44 ± 2.71 | 80.74 ± 0.43 | |
| | 0.01 | 78.87 ± 0.75 (0.02) | 68.65 ± 0.36 | 76.77 ± 0.12 | |
| CoCS | 0.14 | 92.11 ± 0.73 (0.14) | 89.68 ± 0.38 | 92.53 ± 0.67 | 93.32 ± 0.62 |
| | 0.03 | 90.94 ± 0.44 (0.03) | 63.49 ± 3.57 | 91.13 ± 0.31 | |
| | 0.01 | 88.25 ± 0.83 (0.01) | 35.37 ± 1.56 | 85.66 ± 0.45 | |
| PubMed | 0.235 | 86.53 ± 0.18 (0.16) | 82.40 ± 0.32 | 86.84 ± 0.11 | 88.89 ± 0.57 |
| | 0.07 | 85.77 ± 0.39 (0.04) | 41.11 ± 0.31 | 86.02 ± 0.28 | |
| | 0.02 | 82.93 ± 0.67 (0.01) | 81.13 ± 0.22 | 83.41 ± 0.72 | |
| Co-Physics | 0.09 | 94.42 ± 0.24 (0.09) | 88.98 ± 0.23 | 94.79 ± 0.19 | 96.22 ± 0.74 |
| | 0.01 | 93.88 ± 0.49 (0.02) | 67.79 ± 1.37 | 94.15 ± 0.37 | |
| | 0.005 | 92.12 ± 0.63 (0.008) | 67.20 ± 0.30 | 92.43 ± 0.58 | |

Table 10: Node-classification accuracy on `ogbn-arxiv` using official OGB splits

| Dataset | $r = k/p$ | CGC Accuracy |
|---|---|---|
| ogbn-arxiv | 1.0 | $53.44 \pm 0.34$ |
| ogbn-arxiv | 0.15 | $47.23 \pm 0.35$ |
| ogbn-arxiv | 0.04 | $46.28 \pm 0.36$ |
| ogbn-arxiv | 0.01 | $44.17 \pm 0.46$ |
| ogbn-arxiv | 0.004 | $41.85 \pm 0.59$ |
| ogbn-arxiv | 0.001 | $38.75 \pm 0.59$ |

## A.7 Theoretical Convergence Guarantee

***Proof of Theorem 2.*** Recall that for each coalition $S_i$, the coalition-level potential $\phi(R^m_{S_i})$ is defined in equation (11). The global potential is then given by $\Phi(m) = \sum_{S_i \in \mathcal{S}} \phi(R^m_{S_i})$.

If the potential function $\Phi$ meets the cost-alignment condition defined in equation 10, then CGame confirms the properties of a potential game, where local cost improvements support the minimization of global potential. Under CGC algorithm, node $i$ updates its mapping decision from a coalition $S_i$ to $S'_i$ at a mapping profile $m = (m_i, m_{-i})$, only if its cost decreases. Suppose $R^m_{S_i}$ and $R^m_{S'_i}$ are the nodes in coalitions $S_i$ and $S'_i$, respectively, before changing the mapping decision. The change in node $i$'s cost due to the transition from coalition $S_i$ to $S'_i$ is calculated as:

$$\Delta c_i = c_i(m'_i, m_{-i}) - c_i(m_i, m_{-i}) = DE(R^m_{S'_i} \cup \{i\}) - DE(R^m_{S'_i}) - \left(DE(R^m_{S_i}) - DE(R^m_{S_i} \setminus \{i\})\right) \quad (14)$$

where $\Delta c_i < 0$. This inequality holds because a node deviates to $S'_i$ only when such a move strictly reduces its cost. Formally,

$$c_i(m'_i, m_{-i}) < c_i(m_i, m_{-i}) \quad \Rightarrow \quad \Delta c_i = c_i(m'_i, m_{-i}) - c_i(m_i, m_{-i}) < 0. \quad (15)$$

Therefore, the update rule ensures that a node's cost always decreases after deviation.

Now, when a node updates its mapping, the value of the potential function changes as follows:

$$\Delta\phi = \phi_{\text{inc}} - \phi_{\text{dec}}, \quad (16)$$

where $\phi_{\text{inc}}$ represents the increase in $\phi$, and $\phi_{\text{dec}}$ represents the decrease in $\phi$.

Specifically, when node $i$ is remapped from coalition $S_i$ to $S'_i$, the potential value $\phi$ of coalition $S'_i$ increases, while the potential value $\phi$ of coalition $S_i$ decreases. These changes in $\phi$ are calculated as follows:

The increase in $\phi$ for the new coalition $S'_i$ is:

$$\begin{aligned}
\phi_{\text{inc}} &= \phi(R^m_{S'_i} \cup \{i\}) - \phi(R^m_{S'_i}) \\
&= DE(R^m_{S'_i} \cup \{i\}) - DE(R^m_{S'_i}) &&\text{(by equation (11))} \\
&= c_i(m'_i, m_{-i}) &&\text{(by equation (7)).} \quad (17)
\end{aligned}$$

The decrease in $\phi$ for the original coalition $S_i$ is:

$$\begin{aligned}
\phi_{\text{dec}} &= \phi(R^m_{S_i}) - \phi(R^m_{S_i} \setminus \{i\}) \\
&= DE(R^m_{S_i}) - DE(R^m_{S_i} \setminus \{i\}) &&\text{(by equation (11))} \\
&= c_i(m_i, m_{-i}) &&\text{(by equation (7)).} \quad (18)
\end{aligned}$$

If the condition $\Delta c_i < 0$ is satisfied, then from equations (17) and (18), the change in the global potential $\Delta \phi$ is equal to the change in the individual cost $\Delta c_i$. Consequently, $\Delta \phi < 0$, which shows that every locally improving move strictly decreases the global potential. This proves that CGame is an exact potential game. Moreover, this equality holds because a unilateral move of node $i$ affects only the coalition it leaves and the coalition it joins, and the global potential is defined as the sum of the corresponding coalition-level Dirichlet energies. As a result, the value of $\Delta \phi$ decreases monotonically as nodes update their mapping decisions according to the CGC algorithm.

$\square$

***Proof of Theorem 3.*** The presence of a PNE[1] is justified using the arguments outlined below.

1. In the proposed `CGame`, the potential function $\phi(m)$ is defined over the set of all possible mapping profiles, where each node selects a coalition from its neighbors. Since both the number of nodes and their neighbor sets are finite, the number of mapping profiles and thus the range of $\phi(m)$ is finite.

2. The game progresses through updates to these profiles, with nodes updating their choices based on the CGC algorithm. As shown in Theorem 2 and above point that this game is a finite-potential game where each update results in a strictly decreasing value of $\phi(m)$, preventing the game from revisiting previous states and ensuring convergence. Once the potential function reaches its minimum value, any unilateral deviation by a player would increase their own cost and thus increase $\phi(m)$. Therefore, a player will not unilaterally deviate, confirming that the converged state corresponds to a Pure Nash Equilibrium.[2]

$\square$

---

[1]In CGame, a PNE represents a mapping profile where no player can further reduce its cost by unilaterally changing its mapping choice.

[2]Every finite potential game admits a pure-strategy equilibrium, as proved by (Monderer & Shapley, 1996).

