# OpenReview forum: "Graph Coarsening using Game Theoretic Approach"
_TMLR — Accepted by TMLR_

### Review · Reviewer_c8vT · 2025-11-04

**Summary Of Contributions:**

This paper addresses the lack of a unified and interpretable principle for node merging in existing graph coarsening methods, which mostly rely on heuristic or topology-based rules. To overcome these issues, the authors propose CGC, a cooperative-game-based coarsening framework that models node merging as a game where each node’s marginal contribution to the Dirichlet energy determines coalition formation, ensuring interpretable and convergent coarsening with better preservation of structural and feature information. Experimental results show that, compared with existing graph coarsening methods, CGC achieves higher accuracy and faster runtime in most cases.

Strengths:

1) Modeling graph coarsening as a cooperative game introduces a principled and interpretable decision mechanism for node merging. This formulation not only explains why specific nodes are merged but also links the coarsening process to a clear optimization principle, providing stronger interpretability and consistency than heuristic merging rules.
2) The formulation as a potential game ensures convergence to a pure-strategy Nash equilibrium, which offers stronger theoretical justification than many heuristic coarsening methods.

Weaknesses:

1) The proposed method does not demonstrate clear novelty. The overall approach is conceptually similar to previous game-theoretic graph clustering works and fails to introduce new techniques to address existing challenges.
2) The convergence guarantee of potential games applies only to local equilibria. The method may be sensitive to initialization or leader selection, but this is not analyzed.
3) The paper does not clarify whether the pseudo-inverse label projection uses only training labels; if test labels are included, this could cause data leakage and artificially inflate classification performance.
4) The experimental evaluation is incomplete and potentially unfair: many baselines are reported as OOM without explanation or mitigation, and the paper does not compare against the latest graph coarsening methods.

**Audience:**

Yes

**Audience Explanation:**

The paper presents a potential-function framework linking graph coarsening with cooperative game theory, which holds some significance for researchers studying graph compression, spectral approximation, and GNN stability.

**Claims And Evidence:**

No

**Claims Explanation:**

The theoretical proposition of the paper, that the game is a potential game and there exists a PNE, has been rigorously proven to be credible. The convergence conclusion has clear logic and sufficient mathematical derivation. However, the empirical claims are currently not fully substantiated: Several baselines in the tables report “OOM” or missing values, preventing a fair comparison; The numerical stability of pseudoinverse label inference and the separation of training/testing data are not clarified, potentially affecting result reliability; No ablation studies are provided to verify that the performance gains stem from the game-theoretic mechanism rather than specific hyperparameters or feature choices.

**Requested Changes:**

1) Could the authors clarify whether only the training set labels are used when computing the projected labels for the coarsened nodes? If test set labels are included, might this lead to data leakage and overestimated accuracy?
2) Could the authors clarify or provide mitigation strategies for cases where baseline methods report ‘OOM’ or missing results, to ensure experimental completeness and fairness?
3) Could the authors analyze the sensitivity of CGC to initialization and leader selection, given that the convergence guarantees only apply to local equilibria?
4) Could the authors include more recent and relevant baseline methods for a fairer comparison?
5) Inconsistency in notation and dimensions: On page 3 (Background section), you first define 𝑝 as the number of nodes, but later describe the Laplacian matrix as having dimensions 𝑚×𝑚. Please unify the notation and correct all matrix dimension annotations accordingly.

---

> ### Author Response · Authors · 2025-12-01
> **Response to Reviewer c8vT (part 1/5)**
>
> We thank the reviewer for their thoughtful feedback and constructive suggestions. In response to the identified weaknesses (W) and requested changes (C), we provide detailed clarifications and answers (A), additional experiments, and revisions to strengthen the paper. All corresponding updates have been incorporated into the revised manuscript and highlighted in blue for clarity.
>
> ## W1. The proposed method does not demonstrate clear novelty. The overall approach is conceptually similar to previous game-theoretic graph clustering works and fails to introduce new techniques to address existing challenges.
>
> **A1** We thank the reviewer for this comment. Our method is related to earlier game-theoretic graph clustering work in the sense that nodes form coalitions and the algorithm converges to a stable partition. However, the game we define, including its characteristic function, player-cost definition, and potential function, is fundamentally different from both non-cooperative clustering games[1,2] and cooperative clustering games[3,4,5,6,7].
>
> Existing game-theoretic clustering methods define the worth of a coalition or the utility of a player using application-specific similarity or quality scores. Examples include neighbor-overlap or modularity gains in non-cooperative community games[1,2], hyperedge compatibility in hypergraph matching [3], communication and idle-listening energy in WSN clustering [4], internal versus boundary edge weights in strong-community games [5], and node-cluster distance, cluster tightness, and homogeneity in overlapping clustering games on attributed graphs [6,7]. These formulations do not utilize Dirichlet energy to define coalition worth or player cost, nor do they construct a game whose potential aligns with a spectral graph coarsening objective.
>
> Our contributions introduce clear novelty:
>
> 1. __Dirichlet-energy-based characteristic function and exact potential function__.
>  We define coalition worth and player cost using marginal Dirichlet-energy contributions of merged supernodes. We prove that the global objective aggregated across coalitions forms an exact potential function, making this the first game-theoretic framework whose dynamics directly minimize a spectral (Dirichlet-energy) coarsening objective. Therefore, it maintains smooth graph signals and approximates the original Laplacian’s eigenstructure. Such spectral preservation is crucial for retaining the utility of the graph in downstream tasks. For instance, prior work has shown that graph coarsening methods that minimize spectral distortion (differences in Laplacian eigenvalues or Dirichlet energy) achieve minimal loss in graph classification accuracy compared to original graphs [15].
>
>   2. __Alignment with graph coarsening, spectral preservation, and empirical performance__.
>   In CGC, the characteristic function and potential function are constructed from edge-weighted squared feature differences(Dirichlet energy) on the coarsened graph. As a result, every accepted update is guaranteed to decrease a well-defined spectral objective that is meaningful for graph reduction. We further restrict feasible coalitions to locally connected subgraphs with bounded path length, so each supernode represents a small, structurally coherent region where the graph signal is smooth, and it also reduces the search space, making the method faster and more scalable. In contrast, previous game-theoretic clustering methods optimize hyperedge similarity, energy budgets, cut-based community strength, or node-cluster distances. Simply merging their clusters into supernodes does not guarantee that the Dirichlet energy or other eigenvalue-based characteristics of the original graph, will be preserved. Hence, those approaches may compromise spectral fidelity and degrade performance if the coarsened graph is used for spectral algorithms or GNN-based learning.
> 3. __Empirical support for the novelty and impact of the formulation__.
>  Our experiments show that CGC preserves downstream GNN accuracy better than widely used graph coarsening baselines (FGC, LAGC, SCAL, HEM, Graclus, METIS). To directly examine the value of our game formulation, we also implement a GLEAM-style clustering baseline and treat its clusters as supernodes under identical coarsening ratios and iteration budgets. Across Cora and PubMed, CGC consistently achieves higher node-classification accuracy than this GLEAM-based baseline (see Table 1}).  Each value reports the mean accuracy ± standard deviation over 10 multiple runs. This clearly demonstrates that aligning the coalition game with Dirichlet-energy minimization provides tangible performance benefits.

---

> ### Author Response · Authors · 2025-12-01
> **Response to Reviewer c8vT (part 2/5)**
>
> In summary, our method introduces a new cooperative-game formulation tailored specifically for graph coarsening, a Dirichlet-energy-based characteristic and potential function, and a coalition-formation mechanism designed to preserve spectral properties. These elements are fundamentally different from and not addressed by previous game-theoretic graph clustering approaches, thereby establishing clear novelty in the proposed method. We have added Table~1 and its explanation to the experimental section of the revised manuscript, comparing CGC with the GLEAM baseline across different coarsening ratios on Cora and Pubmed.
>
> Table 1: Comparison with GLEAM baseline
> | Dataset | r = k/p | CGC (mean ± std) | GLEAM (mean ± std) |
> | :-----: | :-----: | :--------------: | :----------------: |
> |   Cora  |   0.21  |   87.96 ± 0.32   |    72.34 ± 0.84    |
> |   Cora  |   0.05  |   80.57 ± 0.21   |    42.87 ± 1.47    |
> |  Pubmed |  0.235  |   86.84 ± 0.11   |    74.69 ± 1.69    |
> |  Pubmed |   0.02  |   83.41 ± 0.72   |    39.59 ± 0.45    |
>
> ## W2,C3. The convergence guarantee of potential games applies only to local equilibria. The method may be sensitive to initialization or leader selection, but this is not analyzed.
> **A2**  We thank the reviewer for this insightful question. The sensitivity of our CGC formulation is primarily influenced by the initialization stage, specifically how the initial leaders are selected to begin the coalition formation process. To evaluate this effect, we employ random leader initialization and examine the variability across repeated executions of CGC.
>
> Experimental Setup: We conducted extensive sensitivity analysis on four benchmark datasets (Cora, DBLP, CoCS, and Pubmed) by running CGC multiple times with different random initializations. For each dataset, we performed multiple independent runs (150 for Cora, 50 for CoCS, and 35 for DBLP and Pubmed) and recorded the resulting node classification accuracies.
>
> Results: As summarized in Table 2, Table 3, Table 4, and Table 5, CGC demonstrates remarkable stability under random leader initialization across all four datasets. On Cora, 150 independent runs yield accuracies ranging from 85.19% to 87.53%, with a narrow spread of only 2.34 percentage points. On DBLP, 35 runs fall within an even tighter band of 84.36%–85.05% (range: 0.69%). CoCS exhibits similar robustness, with 50 runs distributed between 91.80% and 92.77% (range: 0.97%). Pubmed shows the highest stability, with 35 runs concentrated in the interval 87.13%–87.50% (range: 0.37%).
> The frequency distributions in these tables reveal that the vast majority of runs cluster around the mean accuracy, with very few outliers. This tight concentration indicates that CGC's performance is highly robust and not strongly dependent on the specific random choice of initial leaders. Despite theoretical convergence guarantees applying only to local equilibria, our empirical results demonstrate that different initializations consistently lead to similar high-quality solutions, suggesting that the local equilibria reached by CGC are of comparable quality across different starting configurations. We have updated this full sensitivity analysis to the experimental section of the revised manuscript.
>
> Table 2. Accuracy distribution across 150 random initializations on Cora
> | Accuracy Range (%) | Frequency |
> | ------------------ | --------- |
> | 85.19–85.42        | 2         |
> | 85.42–85.66        | 2         |
> | 85.66–85.89        | 13        |
> | 85.89–86.12        | 12        |
> | 86.12–86.36        | 30        |
> | 86.36–86.59        | 30        |
> | 86.59–86.82        | 31        |
> | 86.82–87.06        | 20        |
> | 87.06–87.29        | 7         |
> | 87.29–87.53        | 3         |
>
> Table 3. Accuracy distribution across 35 random initializations on DBLP
>
> | Accuracy Range (%) | Frequency |
> | ------------------ | --------- |
> | 84.36–84.42        | 3         |
> | 84.42–84.49        | 0         |
> | 84.49–84.56        | 2         |
> | 84.56–84.63        | 3         |
> | 84.63–84.70        | 4         |
> | 84.70–84.77        | 7         |
> | 84.77–84.84        | 3         |
> | 84.84–84.91        | 7         |
> | 84.91–84.98        | 3         |
> | 84.98–85.05        | 3         |
>
>
> Table 4. Accuracy distribution across 50 random initializations on CoCS
> | Accuracy Range (%) | Frequency |
> | ------------------ | --------- |
> | 91.80–91.90        | 2         |
> | 91.90–91.99        | 2         |
> | 91.99–92.09        | 1         |
> | 92.09–92.19        | 5         |
> | 92.19–92.29        | 6         |
> | 92.29–92.38        | 7         |
> | 92.38–92.48        | 14        |
> | 92.48–92.58        | 6         |
> | 92.58–92.68        | 6         |
> | 92.68–92.77        | 1         |

---

> ### Author Response · Authors · 2025-12-01
> **Response to Reviewer c8vT (part 3/5)**
>
> Table 5. Accuracy distribution across 35 random initializations on Pubmed
> | Accuracy Range (%) | Frequency |
> | ------------------ | --------- |
> | 87.13–87.17        | 5         |
> | 87.17–87.20        | 4         |
> | 87.20–87.24        | 4         |
> | 87.24–87.28        | 2         |
> | 87.28–87.32        | 5         |
> | 87.32–87.35        | 5         |
> | 87.35–87.39        | 1         |
> | 87.39–87.43        | 5         |
> | 87.43–87.46        | 2         |
> | 87.46–87.50        | 2         |
>
>
> ## W3,C1. The paper does not clarify whether the pseudo-inverse label projection uses only training labels; if test labels are included, this could cause data leakage and artificially inflate classification performance.
>
> **A3**  We sincerely thank the reviewer for this question and the opportunity to clarify this aspect. We use only training-set labels $Y_{\text{train}}$ when computing the projected labels $Y_c$ for coarsened nodes via $Y_c = \arg\max(PY_{\text{train}})$[8,9,10], where $P = C^{+}$. This follows the standard evaluation pipeline adopted in prior graph coarsening works. Test labels $Y_{\text{test}}$ are never used either in constructing $Y_c$ or in training the GNN on the coarsened graph, ensuring that no label leakage occurs.
> In the revised manuscript, we clarify this procedure in the experimental section as follows:
> For the node classification task, we first apply the proposed CGC algorithm to the original graph $G(\Theta, X)$ to obtain a coarsened graph $G_c(\Theta_c, X_c)$ without using any node labels. After obtaining $G_c$, we split the node labels on the original graph into 80\% training ($Y_{\text{train}}$) and 20\% testing ($Y_{\text{test}}$). Critically, we completely mask $Y_{\text{test}}$ during the label projection phase. We infer coarsened labels via $Y_c = \arg\max(PY_{\text{train}})$[8,9,10], where \( P = C^{+} \)
> , ensuring that $Y_c$ depends exclusively on training labels with zero contribution from $Y_{\text{test}}$. We then train the Graph Neural Network (GNN) on $G_c(\Theta_c, X_c, Y_c)$ and evaluate performance on the 20\% test nodes in the original graph, whose labels $Y_{\text{test}}$ were completely excluded during coarsening and are used only for final evaluation. This protocol ensures no label leakage and maintains strict separation between training and testing sets.
>
>
> ## W4,C2,C4. The experimental evaluation is incomplete and potentially unfair: many baselines are reported as OOM without explanation or mitigation, and the paper does not compare against the latest graph coarsening methods.
>
> **A4-(a) Explanation and Mitigation of OOM Baseline Failures**
>
> We thank the reviewer for this question. For large datasets such as Genius and OGBN-Arxiv, several baseline methods (FGC, LAGC, SCAL) initially encountered out-of-memory (OOM) errors. To ensure a fair comparison, We reimplemented their computations using sparse matrices and memory-optimized code. This enabled all baselines to run on the OGBN-Arxiv dataset without OOM issues, and the reported accuracies were recomputed under these optimized settings. However, LAGC and FGC still experienced OOM failures (requiring over 200\~GB vRAM) for some larger coarsening ratios.
> We have updated Table~2 in the experimental section of the revised manuscript.
>
>
> Table 2. Node classification accuracy comparison across different coarsening ratios ($r = k/p$) on large-scale datasets
> | **Dataset** | **$r = k/p$** |   **SCAL**   |    **FGC**   |   **LAGC**   |    **CGC**   | **Whole Data** |
> | :---------: | :-----------: | :----------: | :----------: | :----------: | :----------: | :------------: |
> |    Genius   |      0.06     | 79.99 ± 0.11 |      OOM     |      OOM     | 80.00 ± 0.03 |  86.71 ± 0.12  |
> |    Genius   |      0.02     | 80.01 ± 0.09 |      OOM     |      OOM     | 79.97 ± 0.08 |       --       |
> |    Genius   |     0.009     | 79.98 ± 0.09 |      OOM     |      OOM     | 79.91 ± 0.14 |       --       |
> |    Genius   |     0.007     | 79.99 ± 0.11 |      OOM     |      OOM     | 78.63 ± 0.03 |       --       |
> |  OGBN-Arxiv |      0.15     | 27.50 ± 1.41 |      OOM     |      OOM     | 53.83 ± 0.23 |  56.24 ± 0.22  |
> |  OGBN-Arxiv |      0.04     |  7.55 ± 0.25 | 16.05 ± 0.06 | 29.79 ± 0.09 | 52.81 ± 0.33 |       --       |
> |  OGBN-Arxiv |      0.01     |  5.86 ± 0.00 | 31.00 ± 0.20 | 32.37 ± 0.22 | 50.91 ± 0.42 |       --       |
> |  OGBN-Arxiv |     0.004     |  5.86 ± 0.00 | 22.19 ± 0.29 | 30.45 ± 0.88 | 46.86 ± 0.50 |       --       |
> |  OGBN-Arxiv |     0.001     |  5.86 ± 0.00 | 19.34 ± 0.01 | 25.71 ± 0.68 | 42.01 ± 0.56 |       --       |

---

> ### Author Response · Authors · 2025-12-01
> **Response to Reviewer c8vT (part 4/5)**
>
> **A4-(b) Inclusion of Recent and Relevant Baselines for Fair Comparison**
>
> We thank the reviewer for this valuable suggestion to include additional recent and relevant baseline methods. We have now included additional recent and relevant baseline methods to ensure a fairer comparison. In particular, we compare CGC against the recently proposed graph coarsening method SGBGC[11], along with scalable non-learning baselines HEM[12], Graclus[13], and METIS[14]. The results, reported in Table 3, demonstrate that CGC consistently outperforms these methods across multiple datasets and coarsening ratios.
> We have added additional recent and relevant baseline comparisons in Table~3, along with an explanatory discussion, to the experimental section of the revised manuscript for a fair and comprehensive evaluation.
>
> Table 3. Comparison with recent SGBGC and scalable non-learning baselines
> | **Dataset** | **r** |    **SGBGC** |      **HEM** |  **Graclus** |    **METIS** |          **CGC** |
> | ----------- | :---: | -----------: | -----------: | -----------: | -----------: | ---------------: |
> | CORA        |  0.21 | 84.62 ± 0.34 | 79.88 ± 1.42 | 81.32 ± 0.31 | 83.31 ± 0.30 | **87.96 ± 0.32** |
> | CORA        |  0.05 | 82.67 ± 0.44 | 30.21 ± 0.00 | 63.32 ± 2.21 | 30.21 ± 0.00 | **80.57 ± 0.21** |
> | DBLP        |  0.23 | 84.21 ± 0.50 | 75.29 ± 2.10 | 82.30 ± 0.16 | 83.31 ± 0.24 | **84.75 ± 0.54** |
> | DBLP        |  0.05 | 78.72 ± 0.26 | 66.77 ± 2.57 | 64.65 ± 0.41 | 53.18 ± 4.75 | **80.74 ± 0.43** |
> | DBLP        |  0.01 | 73.86 ± 0.25 | 44.71 ± 0.00 | 44.71 ± 0.00 | 44.71 ± 0.00 | **76.77 ± 0.12** |
> | CO-CS       |  0.14 | 92.11 ± 0.44 | 85.76 ± 0.90 | 90.52 ± 0.64 | 90.94 ± 0.48 | **92.53 ± 0.67** |
> | CO-CS       |  0.03 | 91.38 ± 0.28 | 22.56 ± 0.00 | 60.75 ± 2.61 | 23.77 ± 3.30 | **91.13 ± 0.31** |
> | CO-CS       |  0.01 | 91.25 ± 0.17 | 22.56 ± 0.00 | 22.56 ± 0.00 | 22.56 ± 0.00 | **85.66 ± 0.45** |
> | PUBMED      | 0.235 | 85.46 ± 0.20 | 82.04 ± 0.46 | 84.37 ± 0.42 | 82.97 ± 0.27 | **86.84 ± 0.11** |
> | PUBMED      |  0.07 | 83.15 ± 0.15 | 57.50 ± 0.13 | 81.29 ± 0.22 | 57.30 ± 0.38 | **86.02 ± 0.28** |
> | PUBMED      |  0.02 | 82.45 ± 0.31 | 39.27 ± 0.01 | 39.27 ± 0.01 | 39.17 ± 0.05 | **83.41 ± 0.72** |
> | CO-PHYSICS  |  0.09 | 94.78 ± 0.43 | 81.57 ± 1.37 | 16.67 ± 0.00 | 16.67 ± 0.00 | **94.79 ± 0.19** |
> | CO-PHYSICS  |  0.01 | 92.56 ± 0.14 | 50.52 ± 0.00 | 50.52 ± 0.00 | 50.52 ± 0.00 | **94.15 ± 0.37** |
> | CO-PHYSICS  | 0.005 | 89.54 ± 0.22 | 50.52 ± 0.00 | 50.52 ± 0.00 | 50.52 ± 0.00 | **92.43 ± 0.58** |
>
> ## C5. Inconsistency in notation and dimensions: On page 3 (Background section), you first define $p$ as the number of nodes, but later describe the Laplacian matrix as having dimensions m×m. Please unify the notation and correct all matrix dimension annotations accordingly.
>
> **A5** Thank you for pointing this out. This was a notation mistake, and I have corrected it. The Laplacian matrix is now consistently defined as a \(p \times p\) matrix throughout the revised manuscript.
>
> ---
>
> **References**
>
> [1] Supreet Mandala, Soundar Kumara, and Kalyan Chatterjee. 2014. A game-theoretic approach to graph clustering. INFORMS Journal on Computing, 26(3):629–643.
>
> [2] Zhan Bu, Jie Cao, Hui-Jia Li, Guangliang Gao, and Haicheng Tao. 2018. GLEAM: A graph clustering framework based on potential game optimization for large-scale social networks. Knowledge and Information Systems, 55(3):741–770.
>
> [3] Jian Hou, Marcello Pelillo, and Huaqiang Yuan. 2022. Hypergraph matching via game-theoretic hypergraph clustering. Pattern Recognition, 125:108526.
>
> [4] Xiao Yan, Cheng Huang, Jianyuan Gan, and Xiaobei Wu. 2022. Game theory-based energy-efficient clustering algorithm for wireless sensor networks. Sensors, 22(2):478.
>
> [5] Chao Zhao, Ali Al-Bashabsheh, and Chung Chan. 2024. Game Theoretic Clustering for Finding Strong Communities. Entropy, 26(3):268.
>
> [6] Hui-Jia Li, Yuhao Feng, Chengyi Xia, and Jie Cao. 2024. Overlapping graph clustering in attributed networks via generalized clus
>
> [7] Mayank Kumar and Ruchir Gupta. 2021. Overlapping attributed graph clustering using mixed strategy games. Applied Intelligence, 51(8):5299–5313.
>
> [8] Zengfeng Huang, Shengzhong Zhang, Chong Xi, Tang Liu, and Min Zhou. 2021. Scaling up graph neural networks via graph coarsening. Proceedings of the 27th ACM SIGKDD Conference on Knowledge Discovery & Data Mining (KDD), pages 675–684. ACM.
>
> [9] Manoj Kumar, Anurag Sharma, Shashwat Saxena, and Sandeep Kumar. 2023. Featured graph coarsening with similarity guarantees. International Conference on Machine Learning (ICML), pages 17953–17975. PMLR.
>
> [10] Manoj Kumar, Subhanu Halder, Archit Kane, Ruchir Gupta, and Sandeep Kumar. 2024. Optimization framework for semi-supervised attributed graph coarsening. Proceedings of the 40th Conference on Uncertainty in Artificial Intelligence (UAI), 2024.

---

> ### Author Response · Authors · 2025-12-01
> **Response to Reviewer c8vT (part 5/5)**
>
> [11] Shuyin Xia, Xinjun Ma, Zhiyuan Liu, Cheng Liu, Sen Zhao, and Guoyin Wang. 2024. Graph Coarsening via Supervised Granular-Ball for Scalable Graph Neural Network Training. arXiv:2412.13842.
>
> [12] George Karypis and Vipin Kumar. 1998. A Fast and High Quality Multilevel Scheme for Partitioning Irregular Graphs. SIAM Journal on Scientific Computing, 20(1):359–392.
>
> [13] Inderjit S. Dhillon, Yuqiang Guan, and Brian Kulis. 2007. Weighted Graph Cuts without Eigenvectors: A Multilevel Approach. IEEE Transactions on Pattern Analysis and Machine Intelligence (TPAMI), 29(11):1944–1957.
>
> [14] George Karypis and Vipin Kumar. 1998. METIS: A Software Package for Partitioning Unstructured Graphs, Partitioning Meshes, and Computing Fill-Reducing Orderings of Sparse Matrices. Version 4.0, Technical Report, University of Minnesota.
>
> [15] Yu Jin, Andreas Loukas, and Joseph JaJa. 2020. Graph coarsening with preserved spectral properties. International Conference on Artificial Intelligence and Statistics (AISTATS), pages 4452–4462. PMLR.

---

> > ### Comment · Reviewer_c8vT · 2025-12-02
> > **Review of Paper6222**
> >
> > The authors’ responses to Weaknesses 2–4 and the corresponding requested changes are detailed, and they have supplemented the experiments and addressed the issues in the manuscript. However, my concern regarding Weakness 1, the level of innovation, remains unresolved. The authors state that the core contribution lies in constructing feature and potential functions based on the Dirichlet energy. However, energy minimization and spectral preservation are already established objectives in graph coarsening. The paper does not provide theoretical or empirical evidence showing that an energy-driven latent game offers unique advantages. As a result, the proposed method appears to be a combination of two existing techniques rather than introducing a genuinely new approach that addresses the current challenges in graph coarsening.

---

> > > ### Author Response · Authors · 2025-12-08
> > > **Response to Reviewer c8vT**
> > >
> > > ## The proposed method does not demonstrate clear novelty. The overall approach is conceptually similar to previous game-theoretic graph clustering works and fails to introduce new techniques to address existing challenges.
> > >
> > > **A.**
> > > We thank the reviewer for this comment and appreciate the opportunity to clarify the novelty of our approach. Our method differs fundamentally from existing game-theoretic clustering frameworks, which typically define coalition value using application-specific heuristics—such as neighbor overlap, modularity gains, hyperedge weights, energy budgets, or cluster compactness. These prior formulations were designed for graph clustering, aiming to detect communities in the original graph. In contrast, CGC is a coarsening framework: it explicitly produces a reduced graph optimized for preserving spectral properties and enabling downstream learning.
> > >
> > > To further distinguish CGC, we now contrast it with two recent and relevant game-theoretic clustering methods: the strong-community game of ~\cite{zhao2024game}[1] and the overlapping attributed clustering model of ~\cite{li2024overlapping}[2]. While both utilize game-theoretic principles, they focus solely on structural features and clustering tasks, and rely on nontrivial game-specific hyperparameters. By contrast, CGC integrates both structural connectivity and node features, and directly targets the graph coarsening problem under a principled spectral objective without introducing any additional game-specific hyperparameters.
> > >
> > >
> > > Importantly, to the best of our knowledge, no prior work has introduced a potential game formulation for graph coarsening. We respectfully indicate that CGC is the first graph coarsening framework formulated as a cooperative potential game with a Dirichlet energy-based utility. Theorem 2 explicitly proves that CGame is a potential game, providing formal novelty and theoretical grounding. This principled formulation is unique to our work. As a potential game, CGC enjoys guaranteed convergence: each update strictly decreases the global Dirichlet potential, leading to a stable pure Nash equilibrium. Theorem 3 further ensures that at least one pure NE exists, meaning the algorithm always reaches a stable coarsening (no node can unilaterally improve its cost). Indeed, since our potential is the sum of coalition Dirichlet energies, each step strictly decreases total energy, so the game converges to the minimal-energy coarsened graph. These properties imply efficient, robust convergence in practice: notably, CGC maintains consistent performance across diverse datasets and coarsening ratios without manual tuning.
> > >
> > > Theoretical insights are borne out in experiments.  CGC outperforms competing methods: across diverse datasets and GNN models, it achieves consistently higher classification accuracy and better Laplacian spectral preservation.  For example, Table 2 and Fig. 4 show CGC yields higher accuracy with tighter eigenvalue alignment.  CGC also significantly reduces coarsening time, demonstrating improved efficiency and scalability.  These consistent gains across benchmarks underline the practical benefits of our approach.
> > >
> > >
> > > ---
> > >
> > > References:
> > >
> > > [1] Chao Zhao, Ali Al-Bashabsheh, and Chung Chan. 2024. Game Theoretic Clustering for Finding Strong Communities. Entropy, 26(3):268.
> > >
> > > [2] Hui-Jia Li, Yuhao Feng, Chengyi Xia, and Jie Cao. 2024. Overlapping graph clustering in attributed networks via generalized cluster game.

---

> > > > ### Comment · Reviewer_c8vT · 2025-12-11
> > > > **Review of Paper6222**
> > > >
> > > > The authors have addressed most of my questions and added additional experiments that demonstrate the advantages of combining the two techniques, but the motivation remains insufficiently clear.

---

> > > > > ### Author Response · Authors · 2025-12-13
> > > > > **Response to Reviewer c8vT**
> > > > >
> > > > > ## The authors have addressed most of my questions and added additional experiments that demonstrate the advantages of combining the two techniques, but the motivation remains insufficiently clear.
> > > > >
> > > > > **A.** Existing graph coarsening methods fall broadly into two categories, which are heuristic approaches and optimization-based frameworks, but both exhibit fundamental limitations that motivate our work. Heuristic coarsening methods typically rely only on structural information such as adjacency patterns or spectral approximations, and ignore node features. Since modern graph learning depends strongly on feature smoothness and attribute consistency, neglecting feature information often produces coarsened graphs that are less informative and lead to degraded performance on downstream tasks such as node classification. In contrast, optimization-based coarsening methods do incorporate structure and features but introduce many hyperparameters and require repeated tuning across datasets and coarsening ratios. This creates significant computational overhead and undermines the basic purpose of coarsening, which is expected to reduce cost rather than add additional complexity. Furthermore, in these approaches, the search space becomes increasingly large and multidimensional as the graph size grows. Therefore, these approaches generally fail to produce the desired results if the problem size is large and suffer from a scalability issue. These limitations demonstrate the need for a coarsening technique that is both computationally efficient and feature aware while also preserving the structural properties of the original graph. Our work is motivated by this gap and aims to develop a fast method with no hyperparameter tuning that uses both graph topology and node attributes to produce high-quality coarsened graphs that improve downstream task performance.
> > > > >  Building on this motivation, we introduce a framework that uses cooperative game theory to address the limitations of existing approaches. Game theory provides a natural way to model graph coarsening because each node can be viewed as a player that seeks to reduce its individual cost while forming coalitions with other nodes in the graph. This insight motivates our distributed formulation of the graph coarsening problem, where each player minimizes its cost. In our formulation, coalitions correspond to supernodes, and the cost of each player is defined through its marginal contribution to the Dirichlet energy. This creates a direct connection between local node-level decisions and the global objective of minimizing energy distortion in the coarsened graph.
> > > > >
> > > > > We utilize game theory for four key reasons that are grounded in the design of our method. First, cooperative games allow us to incorporate both structural connectivity and feature similarity in a unified manner. The marginal contribution based cost reflects how much a node benefits the coalition in terms of both graph structure and feature smoothness. Second, the game we construct is an exact potential game, which guarantees that every time a node updates its coalition choice, the global Dirichlet energy decreases. This alignment ensures that local best responses naturally guide the system toward a stable coarsened representation without the need for tuning. Third, the potential game formulation ensures convergence to a pure Nash equilibrium, which corresponds to a stable coalition structure where no node has an incentive to deviate. This gives a theoretical guarantee of stability and explains the consistent empirical performance of the method across many datasets. Fourth, because our proposed game-theoretic approach guarantees the existence of at least one Nash equilibrium, it can produce a stable and high-quality solution even when the problem size is very large. This property makes the method more scalable than heuristic or conventional optimization techniques.
> > > > >
> > > > > In summary, we employ cooperative game theory because it offers a principled way to combine structural and feature information, it eliminates the need for costly hyperparameter tuning, and it provides strong theoretical guarantees through the potential game property. This results in a fast and effective coarsening technique that produces stable and informative supernodes which enhance downstream learning performance.

---

### Review · Reviewer_Edju · 2025-11-11

**Summary Of Contributions:**

The paper introduces a Cooperative Game-based Graph Coarsening (CGC) method that models graph coarsening as a cooperative game among nodes. Each node acts as a player forming coalitions (supernodes) to minimize its Dirichlet energy cost, leading to a stable, energy-efficient coarsened graph. The authors prove that the game is a potential game, guaranteeing convergence to equilibrium, and demonstrate that CGC achieves better accuracy and lower runtime than prior coarsening methods across multiple datasets.

Strengths

S1) The application of game theory to the graph coarsening problem is interesting. Treating each node as an individual agent and viewing the node-merging process as coalition formation is both novel and intuitive.

S2) The paper is clearly written and well-organized. The main ideas are presented logically, supported by sound theoretical analysis and solid mathematical grounding.

S3) The proposed approach exhibits strong empirical performance, demonstrating both effectiveness (in preserving downstream accuracy) and efficiency (in computational cost) across diverse datasets.

Weaknesses / Questions

W1) The evaluation protocol for node classification requires clarification. If the primary goal of coarsening is to accelerate training while preserving performance, then inference should ideally be conducted on the original graph, not the coarsened one, for a fair comparison with existing methods. The paper should explicitly describe this procedure.

W2) It would be helpful to discuss whether CGC is applicable to heterophilous graphs, where connected nodes often have dissimilar labels. Such settings can pose challenges for methods that assume homophily.

W3) A simple node-masking baseline—where a subset of nodes is randomly masked to match the coarsened graph’s size—could serve as a valuable reference point, highlighting how much of the performance gain comes from the coarsening mechanism itself.

W4) The choice of using 80% of nodes for training seems unusually high. Many graph learning benchmarks use smaller labeled subsets (e.g., 30% or less). Including experiments with more limited supervision would provide deeper insights into the robustness of CGC.

W5) The paper should also clarify how labels for merged (super)nodes are assigned—whether via majority voting, weighted aggregation, or soft-labeling strategies. This detail is essential for reproducibility and understanding how supervision propagates through the coarsened graph.

**Additional Comments:**

N/A

**Audience:**

Yes

**Audience Explanation:**

Graph coarsening is a highly practical problem with potentially broad applications across various domains.

**Claims And Evidence:**

Yes

**Claims Explanation:**

Overall, the claims made in this paper are well supported by both theoretical analysis and empirical evidence.

**Requested Changes:**

Please address or clarify the above-mentioned weaknesses and questions in the revision.

---

> ### Author Response · Authors · 2025-12-02
> **Response to Reviewer Edju (part 1/3)**
>
> We thank the reviewer for their thoughtful feedback and constructive suggestions. In response to the identified weaknesses / Questions (W), we provide detailed clarifications and answers (A), additional experiments, and revisions to strengthen the paper. All corresponding updates have been incorporated into the revised manuscript and highlighted in blue for clarity.
>
> ## W1. The evaluation protocol for node classification requires clarification. If the primary goal of coarsening is to accelerate training while preserving performance, then inference should ideally be conducted on the original graph, not the coarsened one, for a fair comparison with existing methods. The paper should explicitly describe this procedure.
>
> **A1.** We thank the reviewer for this question. We train the model on the coarsened graph using only the projected training labels. During inference, all predictions are made on the original graph. Thus, training is accelerated through coarsening, while evaluation remains fully comparable to existing methods.
>
> ## W2. It would be helpful to discuss whether CGC is applicable to heterophilous graphs, where connected nodes often have dissimilar labels. Such settings can pose challenges for methods that assume homophily.
>
> **A2.** We thank the reviewer for raising this important point about heterophilous graphs.
> Our method is designed for homophilic graphs, where neighboring nodes exhibit similar feature and label patterns. Since the Dirichlet-energy objective encourages merging structurally and semantically similar nodes, CGC naturally aligns with homophilous settings. All datasets used in our main experiments (e.g., Cora, Citeseer, PubMed, DBLP, Co-CS, Co-Physics, OGBN-Arxiv, Genius) fall within this category. To assess robustness beyond homophily, we additionally tested CGC on two heterophilic benchmarks (Cornell and Texas). We evaluate node-classification accuracy , comparing CGC against classical coarsening baselines HEM [1], Graclus [2], and METIS [3]  under different coarsening ratios (r = k/p). Each value reports the mean accuracy ± standard deviation over 10 runs. As shown in Table 1, CGC substantially outperforms all baseline methods across all coarsening levels on both datasets. Notably, the performance gap widens under stronger coarsening (smaller r), where classical coarseners degrade to near-random accuracy (≈44–56%), whereas CGC maintains significantly higher accuracy (71–84%). This demonstrates that CGC is robust in heterophilic regimes and retains discriminative structural information even when aggressively reducing the graph. We have added Table~1 and its explanation to the experimental section of the revised manuscript.
>
> Table 1. Node classification accuracy (%) on heterophilic datasets (Cornell & Texas)
> |   Dataset   | r = k/p |      HEM     |    Graclus    |     METIS    |      CGC     |   Original   |
> | :---------: | :-----: | :----------: | :-----------: | :----------: | :----------: | :----------: |
> | **Cornell** |   0.23  | 54.10 ± 4.25 |  52.46 ± 3.67 | 53.88 ± 2.38 | 84.53 ± 0.91 | 86.90 ± 0.56 |
> |             |   0.06  | 44.81 ± 0.00 |  43.61 ± 4.25 | 44.81 ± 0.00 | 79.21 ± 1.20 |      --      |
> |             |   0.02  | 44.81 ± 0.00 |  44.81 ± 0.00 | 44.81 ± 0.00 | 71.25 ± 2.07 |      --      |
> |  **Texas**  |   0.23  | 56.39 ± 5.32 |  61.31 ± 2.96 | 57.05 ± 1.27 | 84.53 ± 0.91 | 87.31 ± 1.38 |
> |             |   0.06  | 55.19 ± 0.00 | 40.11 ± 12.08 | 55.19 ± 0.00 | 79.21 ± 1.20 |      --      |
> |             |   0.02  | 55.19 ± 0.00 | 47.76 ± 14.86 | 55.19 ± 0.00 | 71.25 ± 2.07 |      --      |
>
>
> ## W3. A simple node-masking baseline—where a subset of nodes is randomly masked to match the coarsened graph’s size—could serve as a valuable reference point, highlighting how much of the performance gain comes from the coarsening mechanism itself.
>
> **A3.**  We thank the reviewer for this useful suggestion regarding a simple node-masking baseline. To address this suggestion, we implemented a random node-masking baseline, where a subset of nodes is randomly removed to match the size of the coarsened graph produced by CGC. For a fair comparison, we run the node-masking procedure 10 times with different random seeds and report the mean node-classification accuracy ± standard deviation across these 10 random masking trials. As shown in Table 2, which reports node classification accuracy under different coarsening ratios ($r = k/p$) for both CGC and the random node-masking baseline, CGC consistently achieves higher accuracy across all evaluated coarsening levels. This demonstrates that the performance gains of CGC do not simply result from reducing graph size, but rather from its principled coalition-based merging mechanism that preserves structural and label information. Therefore, the coarsening strategy itself plays a critical role in achieving higher accuracy compared to naive downsampling.

---

> ### Author Response · Authors · 2025-12-02
> **Response to Reviewer Edju (part 2/3)**
>
> We have added Table 2 and its explanation to the experimental section of the revised manuscript.
>
> Table 2. Node classification comparison of CGC with a random node-masking baseline
> |   **Dataset**  | **r = k/p** | **Node-Masking Baseline** |      **CGC**     |
> | :------------: | :---------: | :-----------------------: | :--------------: |
> |    **Cora**    |     0.21    |        44.63 ± 6.10       | **87.96 ± 0.32** |
> |                |     0.05    |        22.84 ± 9.67       | **80.57 ± 0.21** |
> |    **DBLP**    |     0.23    |        77.92 ± 1.33       | **84.75 ± 0.54** |
> |                |     0.05    |        70.16 ± 3.61       | **80.74 ± 0.43** |
> |                |     0.01    |       55.10 ± 16.99       | **76.77 ± 0.12** |
> |    **Co-CS**   |     0.14    |        89.08 ± 1.65       | **92.53 ± 0.67** |
> |                |     0.03    |        83.84 ± 3.05       | **91.13 ± 0.31** |
> |                |     0.01    |        81.42 ± 9.92       | **85.66 ± 0.45** |
> |   **Pubmed**   |    0.235    |        56.01 ± 7.60       | **86.84 ± 0.11** |
> |                |     0.07    |       36.92 ± 14.14       | **86.02 ± 0.28** |
> |                |     0.02    |       31.12 ± 17.03       | **83.41 ± 0.72** |
> | **Co-Physics** |     0.09    |        92.01 ± 1.72       | **94.79 ± 0.19** |
> |                |     0.01    |        90.36 ± 3.44       | **94.15 ± 0.37** |
> |                |    0.005    |        79.13 ± 7.25       | **92.43 ± 0.58** |
>
>
>
> ## W4. The choice of using 80% of nodes for training seems unusually high. Many graph learning benchmarks use smaller labeled subsets (e.g., 30% or less). Including experiments with more limited supervision would provide deeper insights into the robustness of CGC.
>
> **A4.**
> We thank the reviewer for this question. We used $80\%$ of nodes for training to maintain consistency with prior coarsening baselines that follow the same protocol. However, we agree that evaluating CGC under more limited supervision is important for understanding its robustness. We have therefore added additional experiments using smaller labeled subsets ($30\%$, $40\%$, $60\%$, and $70\%$ of training nodes) on all benchmark datasets reported in the paper (Cora, DBLP, CoCS, Pubmed, Co-Physics, Genius, OGBN-Arxiv, Cornell, Texas, and Squirrel) and report the results in Table 3. These results demonstrate that CGC maintains competitive and comparable accuracy even when the amount of supervision is substantially reduced, particularly for higher coarsening ratios. This confirms that CGC remains effective and robust across different amounts of training label data.  We have added Table 3 and its explanation to the experimental section of the revised manuscript.
>
> **Table 3.** Node classification accuracy of CGC under different training label ratios (70%, 60%, 40%, and 30%).  Each value represents the mean accuracy ± standard deviation over 10 runs.
> | Dataset | r = k/p | 70% labels | 60% labels | 40% labels | 30% labels |
> |:-------:|:-------:|:-----------:|:-----------:|:-----------:|:-----------:|
> | Cora | 1 | 85.16 ± 0.51 | 85.59 ± 0.48 | 86.53 ± 0.58 | 86.99 ± 0.64 |
> | Cora | 0.2338 | 82.15 ± 0.90 | 78.67 ± 1.50 | 67.25 ± 3.12 | 56.76 ± 3.17 |
> | Cora | 0.0406 | 77.49 ± 1.48 | 75.76 ± 2.58 | 73.70 ± 2.49 | 65.65 ± 3.03 |
> | DBLP | 1 | 85.73 ± 0.34 | 85.31 ± 0.28 | 84.91 ± 0.13 | 84.39 ± 0.19 |
> | DBLP | 0.23 | 82.49 ± 0.29 | 80.06 ± 0.40 | 72.98 ± 0.53 | 66.47 ± 0.93 |
> | DBLP | 0.05 | 79.54 ± 0.50 | 78.39 ± 0.74 | 76.65 ± 0.81 | 72.97 ± 0.98 |
> | DBLP | 0.01 | 76.44 ± 0.89 | 76.13 ± 1.64 | 74.90 ± 1.46 | 72.60 ± 2.29 |
> | CoCS | 1 | 93.97 ± 0.09 | 93.73 ± 0.24 | 91.56 ± 0.20 | 88.32 ± 0.20 |
> | CoCS | 0.14 | 90.46 ± 0.47 | 89.26 ± 0.68 | 82.07 ± 2.83 | 76.87 ± 2.88 |
> | CoCS | 0.03 | 89.77 ± 0.42 | 89.46 ± 0.70 | 87.29 ± 0.65 | 85.63 ± 1.74 |
> | CoCS | 0.01 | 84.63 ± 0.98 | 84.20 ± 1.90 | 83.78 ± 1.60 | 83.06 ± 1.60 |
> | Pubmed | 1 | 87.87 ± 0.39 | 87.76 ± 0.27 | 86.48 ± 0.10 | 85.34 ± 0.16 |
> | Pubmed | 0.235 | 83.20 ± 0.62 | 77.16 ± 1.44 | 57.09 ± 1.84 | 30.27 ± 0.97 |
> | Pubmed | 0.07 | 81.47 ± 0.96 | 78.09 ± 1.43 | 62.50 ± 2.32 | 49.08 ± 2.62 |
> | Pubmed | 0.02 | 79.53 ± 1.41 | 77.17 ± 1.85 | 70.15 ± 1.94 | 65.48 ± 2.38 |
> | Co-Physics | 1 | 96.56 ± 0.05 | 96.44 ± 0.08 | 96.30 ± 0.08 | 96.11 ± 0.07 |
> | Co-Physics | 0.09 | 93.47 ± 0.40 | 92.21 ± 0.60 | 87.01 ± 2.32 | 82.01 ± 2.41 |
> | Co-Physics | 0.01 | 93.50 ± 0.37 | 92.88 ± 0.62 | 90.32 ± 1.74 | 88.00 ± 2.46 |
> | Co-Physics | 0.005 | 91.24 ± 1.84 | 91.36 ± 1.55 | 90.80 ± 1.98 | 90.48 ± 1.78 |
> | Genius | 1 | 79.97 ± 1.05 | 80.00 ± 0.01 | 78.97 ± 3.05 | 72.74 ± 4.20 |
> | Genius | 0.06 | 78.82 ± 12.26 | 78.29 ± 4.81 | 77.26 ± 5.12 | 67.15 ± 23.16 |
> | Genius | 0.02 | 59.39 ± 21.09 | 48.73 ± 26.56 | 48.73 ± 26.56 | 45.86 ± 26.97 |
> | OGBN-Arxiv | 1 | 55.30 ± 0.70 | 53.30 ± 0.17 | 53.03 ± 1.17 | 45.31 ± 0.42 |
> | OGBN-Arxiv | 0.15 | 51.31 ± 0.41 | 48.43 ± 0.56 | 40.89 ± 0.73 | 22.66 ± 1.31 |

---

> > ### Author Response · Authors · 2025-12-02
> > **Response to Reviewer Edju (part 3/3)**
> >
> > Table 3 (continued). Node classification accuracy under different training label ratios
> >
> > | Dataset  | r = k/p | 70% labels | 60% labels | 40% labels | 30% labels |
> > |:--------:|:-------:|:-----------:|:-----------:|:-----------:|:-----------:|
> > | Cornell  | 1       | 86.99 ± 0.64 | 86.53 ± 0.58 | 85.59 ± 0.48 | 85.16 ± 0.51 |
> > | Cornell  | 0.23    | 82.76 ± 1.27 | 82.76 ± 1.27 | 67.35 ± 2.20 | 57.49 ± 2.96 |
> > | Cornell  | 0.06    | 78.99 ± 1.21 | 78.55 ± 1.54 | 74.75 ± 2.33 | 68.46 ± 2.05 |
> > | Cornell  | 0.02    | 71.31 ± 1.96 | 70.32 ± 2.82 | 69.28 ± 3.60 | 67.11 ± 2.60 |
> > | Texas    | 1       | 86.91 ± 0.90 | 86.72 ± 0.71 | 85.75 ± 0.59 | 85.22 ± 0.57 |
> > | Texas    | 0.23    | 82.76 ± 1.27 | 79.21 ± 1.26 | 67.35 ± 2.20 | 57.49 ± 2.96 |
> > | Texas    | 0.06    | 78.99 ± 1.21 | 78.55 ± 1.54 | 74.75 ± 2.33 | 68.46 ± 2.05 |
> > | Texas    | 0.02    | 71.31 ± 1.96 | 70.32 ± 2.82 | 69.28 ± 3.60 | 67.11 ± 2.60 |
> > | Squirrel | 1       | 40.25 ± 3.51 | 31.18 ± 2.50 | 23.95 ± 0.83 | 21.35 ± 0.44 |
> > | Squirrel | 0.16    | 29.33 ± 0.71 | 28.42 ± 0.92 | 27.71 ± 0.90 | 26.03 ± 0.85 |
> > | Squirrel | 0.01    | 30.17 ± 1.72 | 28.81 ± 2.13 | 28.24 ± 1.63 | 29.33 ± 1.39 |
> > | Squirrel | 0.004   | 22.95 ± 2.13 | 22.07 ± 1.42 | 22.34 ± 1.32 | 23.08 ± 2.35 |
> >
> >
> >
> >
> > ## W5. The paper should also clarify how labels for merged (super)nodes are assigned—whether via majority voting, weighted aggregation, or soft-labeling strategies. This detail is essential for reproducibility and understanding how supervision propagates through the coarsened graph.
> >
> > **A5.** We thank the reviewer for this question. In our implementation, labels for the merged supernodes are assigned using a majority voting [4, 5, 6]. Let $Y \in \mathbb{R}^{p}$ denote the original label vector of the graph (one label per node), and let $L \in \mathbb{R}^{p \times l}$ denote its one-hot encoded representation, where $l$ is the number of classes. For a coarsening matrix $C \in \mathbb{R}^{p \times k}$, where $p$ is the number of nodes in the original graph and $k$ is the number of nodes in the coarsened graph. For a given coarsening matrix $C$, we compute its pseudoinverse $P = C^{\dagger}$. We first mask the labels of the $20\%$ test nodes in $L$ to obtain $L_{\text{train}}$, and then project this masked label matrix onto the coarse graph. The aggregated soft label matrix on the coarse graph is:
> >
> > $
> > L_{c} = P * L_{\text{train}} \in \mathbb{R}^{k \times l}.
> > $
> >
> > For each supernode $i$, the predicted coarse label is chosen as the class with the highest aggregated score:
> >
> > $
> > label_{c}(i) = \arg\max_{j} \, L_{c}[i,j],
> > $
> >
> > which produces a coarse label vector $Y_{c} \in \mathbb{R}^{k}$. In our implementation, this step is executed as:
> >
> > $
> > Y_c = \arg\max( P * L_{train}, dim=1)
> > $
> >
> > where $Y_c$ is a vector of size \(k * 1\) containing the final class assignments for each supernode.
> >
> > This majority voting procedure performs a soft, weight-aware aggregation of node labels within each supernode when constructing supervision for training the GNN on the coarsened graph, and avoids ad-hoc majority-voting heuristics by incorporating the membership strengths encoded in $C$.
> >
> > ---
> > **References**
> >
> > [1] George Karypis and Vipin Kumar. 1998. A Fast and High Quality Multilevel Scheme for Partitioning Irregular Graphs. SIAM Journal on Scientific Computing, 20(1):359–392.
> >
> > [2] Inderjit S. Dhillon, Yuqiang Guan, and Brian Kulis. 2007. Weighted Graph Cuts without Eigenvectors: A Multilevel Approach. IEEE Transactions on Pattern Analysis and Machine Intelligence (TPAMI), 29(11):1944–1957.
> >
> > [3] George Karypis and Vipin Kumar. 1998. METIS: A Software Package for Partitioning Unstructured Graphs, Partitioning Meshes, and Computing Fill-Reducing Orderings of Sparse Matrices. Version 4.0, Technical Report, University of Minnesota.
> >
> > [4] Zengfeng Huang, Shengzhong Zhang, Chong Xi, Tang Liu, and Min Zhou. 2021. Scaling up graph neural networks via graph coarsening. Proceedings of the 27th ACM SIGKDD Conference on Knowledge Discovery & Data Mining (KDD), pages 675–684. ACM.
> >
> > [5] Manoj Kumar, Anurag Sharma, Shashwat Saxena, and Sandeep Kumar. 2023. Featured graph coarsening with similarity guarantees. International Conference on Machine Learning (ICML), pages 17953–17975. PMLR.
> >
> > [6] Manoj Kumar, Subhanu Halder, Archit Kane, Ruchir Gupta, and Sandeep Kumar. 2024. Optimization framework for semi-supervised attributed graph coarsening. Proceedings of the 40th Conference on Uncertainty in Artificial Intelligence (UAI), 2024.

---

### Review · Reviewer_Rts6 · 2025-11-11

**Summary Of Contributions:**

The paper proposes a cooperative-game-theoretic algorithm for graph coarsening that seeks coalitions (supernodes) by minimizing a Dirichlet-energy–based objective. Nodes are treated as players; each coalition’s worth is the energy and each node’s cost is its marginal contribution to that energy. A leader-base} heuristic restricts candidate coalitions to 1-hop neighborhoods of high-degree nodes to avoid an NP-complete subset search; then iterative best-response updates are used until no node wants to deviate. The authors claim (i) the game is a \textbf{potential game}, hence convergence to a PNE; (ii) the approach is fast; and (iii) the coarsened graphs preserve structure and yield good node-classification accuracy across several datasets (Cora, DBLP, Coauthor CS/Physics, Pubmed, OGBN-Arxiv, Genius). Results tables/figures include Algorithm 1 (p.7), Tables 1–3 (pp.9–12), Figure 2 (workflow), Figure 3 (leader vs. non-leader refinement), and spectral/$\varepsilon$-similarity plots (Figure 4).

**Additional Comments:**

None.

**Audience:**

Yes

**Audience Explanation:**

This paper would interest a TMLR audience because it presents a new perspective on improving the scalability of graph-based learning methods if the claimed results are correct - currently not verifiable due to notation inconsistencies and presentation issues. The authors propose a Cooperative-Based Graph Coarsening (CGC) algorithm that applies ideas from game theory to reduce large graphs while preserving key structural and feature properties. By framing coarsening as a potential game that minimizes Dirichlet energy, the method provides a principled way to form stable node groupings without extensive hyperparameter tuning. The approach may complement existing graph learning techniques by offering a computationally efficient preprocessing step that can benefit Graph Neural Networks (GNNs) when dealing with large datasets.

**Broader Impact Concerns:**

None that I know of.

**Claims And Evidence:**

No

**Claims Explanation:**

Below are my major concerns with the paper:

(A) Inconsistencies/Ambiguities in Energy and Potential-Game Definitions

\begin{itemize}

\item Notation in general is a mess. For example, The manuscript uses the symbol $m$ to represent multiple distinct quantities across different sections, which may cause confusion for readers. For instance, in Section~2.1, $m$ denotes the number of classes in the label matrix $L \in \mathbb{R}^{p \times m}$, while in Section~2.4, it represents the mapping profile $(m_1, m_2, \ldots, m_p)$ in the game-theoretic formulation. Later, $m$ also appears as the number of coalitions in the complexity analysis (e.g., $O(m \cdot \Delta)$) and as the Laplacian dimension in Equation~(1). Reusing the same symbol for unrelated concepts (labels, mapping profiles, coalitions, and matrix dimensions) detracts from the clarity of the presentation.

\item \textbf{Which edges define the coalition energy?} Eq.~(5a) minimizes $\sum_{i<j}\tilde w_{ij}|\tilde x_i-\tilde x_j|^2$, i.e., \textbf{between-supernode} variation (a cut-like energy). In Eq.~(7), $DE(R^m_{S_i})$ is defined using $E_x$, “the set of edges connected to the $i$-th supernode on the coarsened graph,” while Eq.~(11) defines the coalition potential with $E_{S_i}$, “the edges connected to the $i$-th supernode.” These are boundary edges, not purely internal edges, and a single cross-coalition edge contributes to the energy of \textbf{multiple coalitions}. Then, the global potential in Eq.~(12) is the sum of coalition energies. Without a careful accounting (e.g., a factor $1/2$ or an edge partition), this \textbf{double-counts} inter-coalition terms. It is unclear that a unilateral node move changes the global potential exactly by that node’s marginal cost as claimed in Appendix A.5 (Eqs. (16–18)). Please redefine coalition potentials over disjoint edge sets or redo the potential-game argument with explicit handling of cross-coalition terms.

  \item \textbf{Objective mismatch}: The coarsening objective is Eq.~(5a), but the node cost (Eq.~(7)) is defined via differences of $DE(\cdot)$ on coalitions computed over $E_x$ or $E_{S_i}$. The relationship between the sum of marginal costs and the global objective in (5a) is not proven. Currently, the potential function $\Phi(m)=\sum_i DE(R^m_{S_i})$ is not obviously equivalent to (5a). Please formalize the equivalence.
\end{itemize}

(B) Label Leakage in the Evaluation Protocol (maybe critical)

In \textbf{Node Classification}, the paper states: ``we infer [the coarsened graph’s] node labels using $Y_c = \arg\max(PY)$'' (Section 4.2), \emph{then} train GNNs on $G_c(\Theta_c,X_c,Y_c)$ and evaluate on the remaining 20\% of nodes (same section). Unless $Y$ contains only \textbf{training labels}, $PY$ averages \textbf{all} original labels inside each supernode—including test labels—which leaks test information into training via $Y_c$. Nothing in the text specifies masking test labels when forming $Y$ for the $PY$ step; in fact, the phrasing “excluding their labels during the coarsening process” explicitly refers to coarsening, not to the label-aggregation step used to create the training labels on the coarsened graph. This can inflate accuracy substantially, especially when coalitions mix train and test nodes. Please clarify and, if needed, re-run with $Y$ restricted to the training subset (and a principled way to handle unlabeled nodes when aggregating).


(C) NP-Completeness Argument Seems Detached from the Actual Problem

Theorem 1 (Appendix A.1) proves NP-completeness of a stylized “Stable Coalition decision problem” defined over arbitrary real numbers and an arbitrary function $f$, then reduces from a special knapsack instance. This construction is not tied to the graph-based Dirichlet energy used in the paper. The result thus shows that \emph{some} stability problem is NP-complete, not that the \emph{graph-coarsening stability check} with the specific cost function in Eq.~(7) is NP-hard. A formal reduction aligned with Eqs.~(6–7) and the energy semantics is needed.

(D) Algorithmic Clarity and Complexity

\begin{itemize}

  \item \textbf{Notation drift}: The paper alternates between $W$ and $A$ for adjacency; labels are denoted by $L$ and later by $Y$; Algorithm 1 initializes \texttt{ccost ← \{i : Dn[i]\}} without defining \texttt{Dn}. Please unify symbols.

  \item \textbf{Complexity claims vary}: Section 3 states $O(\Delta)$ per coalition, while Section 4.2 cites $O(p\cdot \Delta \cdot k)$. Moreover, computing all marginal costs $c_{S_i}(j)=DE(S_i)-DE(S_i\setminus\{j\})$ naively is $O(|S_i|)$ energy evaluations, potentially $O(\Delta^2)$ per leader. Provide a precise per-iteration complexity and iteration bound.

  \item \textbf{Heuristic vs. theory}: The PNE proof assumes arbitrary feasible coalitions, but Algorithm 1 restricts to leader-containing 1-hop subsets. Does the potential-game guarantee still hold under this restriction?

\end{itemize}

(E) Experimental Methodology and Reporting

\begin{itemize}

  \item Use official OGBN-Arxiv splits in addition to custom 80/20.

  \item Include scalable non-learning baselines (HEM/Graclus/METIS) and memory-optimized FGC/LAGC.

  \item Add ablations: label aggregation (training-only), leader vs. non-leader, hop-size, and leader-selection metrics.

  \item Clarify $\varepsilon$-similarity: The definition $(1-\varepsilon)\operatorname{tr}(X^\top \Theta X)\le \operatorname{tr}(\tilde X^\top \Theta_c \tilde X)\le (1+\varepsilon)\operatorname{tr}(X^\top \Theta X))$ is feature-dependent, unlike standard Rayleigh quotient bounds.

\end{itemize}

Minor Issues / Clarity / Presentation

\begin{itemize}

  \item Explicitly relate $P=C^\dagger$ in Eq.~(3) to averaging in Eq.~(5b).

  \item Clarify all Algorithm 1 variables (\texttt{Dn}, \texttt{Scoalition}, \texttt{ccost}).

  \item Unify label notation ($L$ vs. $Y$).

  \item Fix missing bibliographic details and grammar issues (e.g., ``descibed'' → ``described'').

\end{itemize}

Questions to the Authors:

\begin{enumerate}

  \item Is $Y$ restricted to training labels in $Y_c=\arg\max(PY)$?

  \item Provide a formal statement aligning Eq.~(5a) with Eqs.~(11–12).

  \item Do Theorems 2–3 hold under leader-restricted 1-hop coalitions?

  \item What is the formal per-iteration complexity and convergence bound?

  \item Include OGBN-Arxiv results with official splits.

  \item Provide ablations: leader variants, hop sizes, smoothing, alternative costs.

  \item Add scalable non-learning baselines.

\end{enumerate}

**Requested Changes:**

Below are my major concerns with the paper:

(A) Inconsistencies/Ambiguities in Energy and Potential-Game Definitions

\begin{itemize}

\item Notation in general is a mess. For example, The manuscript uses the symbol $m$ to represent multiple distinct quantities across different sections, which may cause confusion for readers. For instance, in Section~2.1, $m$ denotes the number of classes in the label matrix $L \in \mathbb{R}^{p \times m}$, while in Section~2.4, it represents the mapping profile $(m_1, m_2, \ldots, m_p)$ in the game-theoretic formulation. Later, $m$ also appears as the number of coalitions in the complexity analysis (e.g., $O(m \cdot \Delta)$) and as the Laplacian dimension in Equation~(1). Reusing the same symbol for unrelated concepts (labels, mapping profiles, coalitions, and matrix dimensions) detracts from the clarity of the presentation.

\item \textbf{Which edges define the coalition energy?} Eq.~(5a) minimizes $\sum_{i<j}\tilde w_{ij}|\tilde x_i-\tilde x_j|^2$, i.e., \textbf{between-supernode} variation (a cut-like energy). In Eq.~(7), $DE(R^m_{S_i})$ is defined using $E_x$, “the set of edges connected to the $i$-th supernode on the coarsened graph,” while Eq.~(11) defines the coalition potential with $E_{S_i}$, “the edges connected to the $i$-th supernode.” These are boundary edges, not purely internal edges, and a single cross-coalition edge contributes to the energy of \textbf{multiple coalitions}. Then, the global potential in Eq.~(12) is the sum of coalition energies. Without a careful accounting (e.g., a factor $1/2$ or an edge partition), this \textbf{double-counts} inter-coalition terms. It is unclear that a unilateral node move changes the global potential exactly by that node’s marginal cost as claimed in Appendix A.5 (Eqs. (16–18)). Please redefine coalition potentials over disjoint edge sets or redo the potential-game argument with explicit handling of cross-coalition terms.

  \item \textbf{Objective mismatch}: The coarsening objective is Eq.~(5a), but the node cost (Eq.~(7)) is defined via differences of $DE(\cdot)$ on coalitions computed over $E_x$ or $E_{S_i}$. The relationship between the sum of marginal costs and the global objective in (5a) is not proven. Currently, the potential function $\Phi(m)=\sum_i DE(R^m_{S_i})$ is not obviously equivalent to (5a). Please formalize the equivalence.
\end{itemize}

(B) Label Leakage in the Evaluation Protocol (maybe critical)

In \textbf{Node Classification}, the paper states: ``we infer [the coarsened graph’s] node labels using $Y_c = \arg\max(PY)$'' (Section 4.2), \emph{then} train GNNs on $G_c(\Theta_c,X_c,Y_c)$ and evaluate on the remaining 20\% of nodes (same section). Unless $Y$ contains only \textbf{training labels}, $PY$ averages \textbf{all} original labels inside each supernode—including test labels—which leaks test information into training via $Y_c$. Nothing in the text specifies masking test labels when forming $Y$ for the $PY$ step; in fact, the phrasing “excluding their labels during the coarsening process” explicitly refers to coarsening, not to the label-aggregation step used to create the training labels on the coarsened graph. This can inflate accuracy substantially, especially when coalitions mix train and test nodes. Please clarify and, if needed, re-run with $Y$ restricted to the training subset (and a principled way to handle unlabeled nodes when aggregating).


(C) NP-Completeness Argument Seems Detached from the Actual Problem

Theorem 1 (Appendix A.1) proves NP-completeness of a stylized “Stable Coalition decision problem” defined over arbitrary real numbers and an arbitrary function $f$, then reduces from a special knapsack instance. This construction is not tied to the graph-based Dirichlet energy used in the paper. The result thus shows that \emph{some} stability problem is NP-complete, not that the \emph{graph-coarsening stability check} with the specific cost function in Eq.~(7) is NP-hard. A formal reduction aligned with Eqs.~(6–7) and the energy semantics is needed.

(D) Algorithmic Clarity and Complexity

\begin{itemize}

  \item \textbf{Notation drift}: The paper alternates between $W$ and $A$ for adjacency; labels are denoted by $L$ and later by $Y$; Algorithm 1 initializes \texttt{ccost ← \{i : Dn[i]\}} without defining \texttt{Dn}. Please unify symbols.

  \item \textbf{Complexity claims vary}: Section 3 states $O(\Delta)$ per coalition, while Section 4.2 cites $O(p\cdot \Delta \cdot k)$. Moreover, computing all marginal costs $c_{S_i}(j)=DE(S_i)-DE(S_i\setminus\{j\})$ naively is $O(|S_i|)$ energy evaluations, potentially $O(\Delta^2)$ per leader. Provide a precise per-iteration complexity and iteration bound.

  \item \textbf{Heuristic vs. theory}: The PNE proof assumes arbitrary feasible coalitions, but Algorithm 1 restricts to leader-containing 1-hop subsets. Does the potential-game guarantee still hold under this restriction?

\end{itemize}

(E) Experimental Methodology and Reporting

\begin{itemize}

  \item Use official OGBN-Arxiv splits in addition to custom 80/20.

  \item Include scalable non-learning baselines (HEM/Graclus/METIS) and memory-optimized FGC/LAGC.

  \item Add ablations: label aggregation (training-only), leader vs. non-leader, hop-size, and leader-selection metrics.

  \item Clarify $\varepsilon$-similarity: The definition $(1-\varepsilon)\operatorname{tr}(X^\top \Theta X)\le \operatorname{tr}(\tilde X^\top \Theta_c \tilde X)\le (1+\varepsilon)\operatorname{tr}(X^\top \Theta X))$ is feature-dependent, unlike standard Rayleigh quotient bounds.

\end{itemize}

Minor Issues / Clarity / Presentation

\begin{itemize}

  \item Explicitly relate $P=C^\dagger$ in Eq.~(3) to averaging in Eq.~(5b).

  \item Clarify all Algorithm 1 variables (\texttt{Dn}, \texttt{Scoalition}, \texttt{ccost}).

  \item Unify label notation ($L$ vs. $Y$).

  \item Fix missing bibliographic details and grammar issues (e.g., ``descibed'' → ``described'').

\end{itemize}

Questions to the Authors:

\begin{enumerate}

  \item Is $Y$ restricted to training labels in $Y_c=\arg\max(PY)$?

  \item Provide a formal statement aligning Eq.~(5a) with Eqs.~(11–12).

  \item Do Theorems 2–3 hold under leader-restricted 1-hop coalitions?

  \item What is the formal per-iteration complexity and convergence bound?

  \item Include OGBN-Arxiv results with official splits.

  \item Provide ablations: leader variants, hop sizes, smoothing, alternative costs.

  \item Add scalable non-learning baselines.

\end{enumerate}

---

> ### Author Response · Authors · 2025-12-06
> **Response to Reviewer Rts6 (part 1)**
>
> We sincerely thank the reviewer for the extremely thorough and constructive review. Below, we provide a concise, point-by-point response addressing all major concerns, technical clarifications, experimental questions, and minor presentation issues. All corresponding revisions have been incorporated into the revised manuscript and are highlighted in blue for clarity.
>
> # (A) Inconsistencies/Ambiguities in Energy and Potential-Game Definitions
>
> ## 1. Notation inconsistencies
>
> **A1.**
> Thank you for highlighting the inconsistent use of the symbol $m$. We have carefully revised the manuscript to ensure that each symbol is used uniquely and consistently. The following corrections have been made:
>
> - Section 2.1 (Label matrix): The previous notation  $ L \in \mathbb{R}^{p \times m} $ was used mistakenly. It has now been corrected to $ L \in \mathbb{R}^{p \times \ell} $, where $\ell$ denotes the number of classes.
> - Equation (1) (Graph Laplacian dimension): The earlier expression $\Theta \in \mathbb{R}^{p \times m}$ has been replaced with the correct form $\Theta \in \mathbb{R}^{p \times p}$, which is consistent with the Laplacian being a $p \times p$ matrix.
> - Section 2.4 (Neighbor set of node $i$): The notation $A_i$ = {$a_1, a_2, ..., a_m $}, where $m$ denotes the number of neighbors of node $i$}, where $|A_i|$ explicitly denotes the number of neighbors of node $i$ has been corrected to $A_i$ = \{$a_1, a_2, ..., a_{|A_i|}$ \}, where $m$ denotes the number of neighbors of node $i$.
>
> - Appendix A.2 (Complexity analysis): To remove ambiguity, the following changes were made:
>    - The symbol $m$ (previously used for the number of coalitions) has been replaced with $k$, since the number of coalitions equals the number of supernodes in the coarsened graph.
>    - The symbol $n$ has been replaced with $p$ to maintain consistency with the notation for the number of graph nodes.
>    - The degree term $d_i$ has been replaced with $|A_i|$, the cardinality of the neighbor set of node $i$.
>
> These revisions ensure that all notation is consistent and eliminate ambiguity across the manuscript.
>
>
>
> ## 2. Which edges define the coalition energy?
> **A2.**
> - We thank the reviewer for this question. Our intention throughout the manuscript is to compute the coalition energy using the Dirichlet energy contributions of all edges on the coarsened graph that are incident to the supernode corresponding to coalition $S_i$. We have revised the explanations around Eqs.~(7), (11), and (12) to make these points explicit and to eliminate any ambiguity regarding which edges are used in the coalition energy.
>
> - Both $E_x$ and $E_{S_i}$ were intended to refer to exactly this same set of incident edges. To eliminate ambiguity, we will update equation (7) in the revised manuscript by replacing $E_x$ with the unified notation $E_{S_i}$.
>
> - We clarify that coalition energy is computed over the Dirichlet-energy edge set without double-counting, ensuring the correctness of the potential-game proof. In equation (5a), the minimization objective $\sum_{i<j} \tilde{w}_{ij}\|\tilde{x}_i - \tilde{x}_j\|^2$ represents the global Dirichlet energy of the coarsened graph, where each edge is counted exactly once.
>
> In equations (7) and (11), the coalition-level energy is defined as $DE(R_{S_i}^m)=\frac{1}{2}\sum_{(u,v)\in E_{S_i}} w_{uv}\|x_u - x_v\|^2$, where $E_{S_i}$ denotes the set of edges incident to coalition $S_i$. For a cross-coalition edge $(u,v)$, where $u\in S_i$ and $v\in S_j$, the term appears in both $E_{S_i}$ and $E_{S_j}$. The factor $\frac{1}{2}$ ensures that each such edge contributes exactly once to the global potential. Therefore, $\Phi(m)=\sum_{S_i\in\mathcal{S}} \phi(R_{S_i}^m)=\sum_{i<j} \tilde{w}_{ij}\|\tilde{x}_i - \tilde{x}_j\|^2$, which is mathematically equivalent to equation (5a), demonstrating that no double-counting occurs.
>
> Under this formulation, a unilateral deviation of node $i$ affects only coalitions $S_i$ and $S_i'$, and the resulting change satisfies $\Delta \Phi = \Delta c_i$, consistent with equations [16-17] in Appendix A.5. Specifically,
> $\Delta \Phi = [\phi(R_{S_i'}^m \cup \{i\}) - \phi(R_{S_i'}^m)] - [\phi(R_{S_i}^m) - \phi(R_{S_i}^m \setminus \{i\})]
> = [DE(R_{S_i'}^m \cup \{i\}) - DE(R_{S_i'}^m)] - [DE(R_{S_i}^m) - DE(R_{S_i}^m \setminus \{i\})]
> = c_i(m_i', m_{-i}) - c_i(m_i, m_{-i}) = \Delta c_i,$
> which follows directly from the definition of player cost in equation (7). Hence, the cost-alignment property holds, confirming that the proposed CGame is an exact potential game, guaranteeing convergence to a Pure Nash Equilibrium.

---

> ### Author Response · Authors · 2025-12-06
> **Response to Reviewer Rts6 (part 2)**
>
> ## 3. Objective mismatch
> **A3.** We thank the reviewer for this valuable question and the opportunity to clarify the relationship between the coalition-level Dirichlet energy and the global objective. Our construction is as follows. For each coalition $S_i$, the quantity $DE(R_{S_i}^m)$ is defined as the Dirichlet energy contribution of all edges on the coarsened graph that are incident to the supernode corresponding to $S_i$, each term weighted by a factor $\tfrac{1}{2}$. Hence, every coarsened edge $(i,j)$ contributes
> $\frac{1}{2} \tilde{w}_{ij}\|\tilde{x}_i - \tilde{x}_j\|^2$
>
> to $DE(R_{S_i}^m)$ via its incidence to supernode $i$ and
> $\frac{1}{2}\,\tilde{w}_{ij}\,\|\tilde{x}_i - \tilde{x}_j\|^2$
>
> to $DE(R_{S_j}^m)$ via its incidence to supernode $j$. Summing $DE(R_{S_i}^m)$ over all coalitions therefore yields
> $\Phi(m) = \sum_{S_i \in \mathcal{S}} DE(R_{S_i}^m) = \sum_{(i,j)} \tilde{w}_{ij}\,\|\tilde{x}_i - \tilde{x}_j\|^2,$
> which coincides exactly with the global Dirichlet energy objective in Eq. (5a), and thus there is no double counting.
>
> The per-node cost $c_i(m_i,m_{-i})$ is by definition the marginal change in $DE(\cdot)$ induced by (re)assigning node $i$ to a coalition. When node $i$ unilaterally changes its mapping from $S_i$ to $S_i'$, only the local energies of these two coalitions are affected in the global sum $\Phi(m) = \sum_{S \in \mathcal{S}} DE(R_S^m)$. Hence, the change in global potential $\Delta \Phi$ equals the difference between the corresponding marginal contributions, which is precisely the node’s marginal cost $\Delta c_i$. This establishes the equivalence between the Dirichlet-energy objective in Eq. (5a), the coalition potentials $DE(R_{S_i}^m)$, and the potential function $\Phi(m)$ used in the potential-game argument.
>
> # (B) Label Leakage in the Evaluation Protocol
> **A.** We thank the reviewer for this question and the opportunity to clarify this point. In our implementation, we use only the training labels $Y_{\text{train}}$ when computing the projected labels $Y_c$ for coarsened nodes via $Y_c = \arg\max(PY_{\text{train}})$ [1, 2, 3], where $P = C^{+}$. Test labels $Y_{\text{test}}$ are never used either in constructing $Y_c$ or in training the GNN on the coarsened graph, so no data leakage occurs.
>
> In the revised manuscript, we clarify this procedure in the experimental section as follows: for the node classification task, we first apply the proposed CGC algorithm to the original graph $G(\Theta, X)$ to obtain a coarsened graph $G_c(\Theta_c, X_c)$ without using any node labels. After obtaining $G_c$, we split the node labels on the original graph into 80\% training ($Y_{\text{train}}$) and 20\% testing ($Y_{\text{test}}$) and completely mask $Y_{\text{test}}$ during the label projection phase. We then infer coarsened labels via $Y_c = \arg\max(PY_{\text{train}})$ [1,2,3], where $P = C^{+}$, ensuring that $Y_c$ depends exclusively on training labels with no contribution from $Y_{\text{test}}$.
>
> Finally, we train the Graph Neural Network (GNN) on $G_c(\Theta_c, X_c, Y_c)$ and evaluate performance on the 20\% test nodes in the original graph, whose labels $Y_{\text{test}}$ are completely excluded during coarsening and training and are used only for final evaluation. This protocol ensures that there is no label leakage and that the separation between training and testing sets is strictly maintained.
>
> # (D) Algorithmic Clarity and Complexity
> ## Complexity And Iteration Bound
> **A.**  We thank the reviewer for this question regarding the formal per-iteration complexity and convergence behavior of our algorithm. One iteration is defined as a full pass over the current set of leaders, i.e., a sweep that updates each leader once. In a single iteration, at most $k$ leaders are processed. For each leader, the candidate coalition contains at most $\Delta$ one-hop neighbors. Computing all marginal costs within this coalition requires at most $O(\Delta^{2})$ energy evaluations, while checking whether these nodes already belong to other coalitions and updating the coalition structure is bounded by $O(p\Delta)$. Thus, the per-iteration cost is
> $
> O\big(k(\Delta^{2} + p\Delta)\big).
> $
>
> Since $p > \Delta$ for any graph, the dominant term is $p\Delta$, and this expression simplifies to $O(kp\Delta)$, which is the form reported in Section 4.2. The earlier $O(\Delta)$ term in Section 3 referred only to the number of local stability checks inside a single coalition and not to the full per-iteration cost; this has now been clarified.
>
> Regarding convergence, CGC defines a finite exact potential game: every accepted coalition update yields the same decrease in the global potential as in the individual player's cost, and the potential is a non-negative Dirichlet-energy quantity. Hence the potential cannot decrease indefinitely, and only finitely many improving deviations are possible. The repeat until loop therefore converges in a finite number of iterations and always reaches a PNE.

---

> ### Author Response · Authors · 2025-12-06
> **Response to Reviewer Rts6 (part 3)**
>
> (D) (continued)
>
> To be more specific, after the first full sweep over the $p$ nodes, the leader set stabilizes and contains exactly $k$ leaders. In the subsequent dynamics, only the remaining $p-k$ follower nodes may change their coalition, and each such move strictly decreases the Dirichlet-energy potential. Since each follower can choose among at most $k$ leaders, a follower can switch coalitions at most $(k-1)$ times. Hence, the total number of accepted coalition updates is bounded by $(p-k)(k-1) = O(pk)$. Under our update schedule, each iteration processes at most $k$ leaders, so the number of iterations until convergence is at most $O(pk/k) = O(p)$, while the total number of node-level updates is $O(pk)$, which in the worst case ($k \le p$) is at most quadratic, $O(p^{2})$, in the number of nodes. These complexity and convergence guarantees have been incorporated into Section~4.2 of the revised manuscript for clarity.
>
> ## Notation drift
> **A.** Thank you for pointing out the notation inconsistencies. We have now unified the
> notation throughout the paper. Specifically:
>
> (1) For adjacency, we consistently use the symbol $W$ in all sections.
> (2) For labels, we use $L$ uniformly in the experimental section, replacing the
> earlier use of $Y$.
> (3) In Algorithm 1, the term Dn in the initialization step was a
> left-over from an earlier draft. We have replaced it with DE, which is
> the Dirichlet-energy cost defined in equation (7). The explanation following the
> algorithm has also been updated to say:  ``The algorithm begins by initializing each node as a singleton coalition, and
> its cost DE is computed using equation (7).'' These modifications remove the notation drift and make the terminology
> consistent across the manuscript.
>
> ## Heuristic vs. theory
> **A.** Yes, the potential-game guarantee continues to hold under the leader-restricted
> 1-hop coalition rule. The proofs of Theorem 2 and Theorem 3 rely only on the
> existence of an exact potential function, not on the availability of all
> possible coalitions. Algorithm~1 restricts each player’s feasible deviations
> to leader-initiated 1-hop coalitions, but every allowed deviation still
> produces a change in individual cost that is exactly matched by the change in
> the global potential. Hence, the game remains an exact potential game under
> this restricted action space, and every admissible coalition update strictly
> decreases the potential. The finite-improvement property therefore still
> holds, ensuring convergence to a PNE even with the heuristic 1-hop
> leader-based restriction.
>
>
> # (E) Experimental Methodology and Reporting
> ## 1. Use official OGBN-Arxiv splits in addition to custom 80/20.
> **A1.**
> Thank you for the suggestion. We have now included results on the **ogbn-arxiv** dataset using the **official OGB splits**. The performance of CGC under different coarsening ratios is reported in Table 1, demonstrating that CGC maintains competitive accuracy even under strong coarsening. In the revised manuscript, we have added this table and the corresponding discussion in the experimental section.
>
> Table 1: Node classification accuracy on ogbn-arxiv using official OGB splits
>
> | Dataset   | r = k/p| CGC Accuracy |
> |--------------|---------------|------------------------|
> | ogbn-arxiv   | $1.0$         | $53.44 \pm 0.34$        |
> | ogbn-arxiv   | $0.15$        | $47.23 \pm 0.35$        |
> | ogbn-arxiv   | $0.04$        | $46.28 \pm 0.36$        |
> | ogbn-arxiv   | $0.01$        | $44.17 \pm 0.46$        |
> | ogbn-arxiv   | $0.004$       | $41.85 \pm 0.59$        |
> | ogbn-arxiv   | $0.001$       | $38.75 \pm 0.59$        |
>
> ## 2. Include scalable non-learning baselines (HEM/Graclus/METIS) and memory-optimized FGC/LAGC.
> **A2.** We thank the reviewer for suggesting the inclusion of scalable non-learning baselines. Node classification accuracy (%) comparing CGC with scalable non-learning graph coarsening baselines HEM, Graclus, and METIS under different coarsening ratios $r = k/p$. Each value is reported as mean $\pm$ standard deviation over 10 runs. The results, summarized in Table 2, show that CGC consistently matches or outperforms these baselines across multiple datasets and coarsening ratios $r = k/p$, particularly under more aggressive coarsening. We have added Table 2 and its explanation in the experimental section of the revised manuscript.
>
> Table 2: Node classification accuracy (%) comparing CGC with scalable non-learning graph coarsening baselines HEM, Graclus, and METIS under different coarsening ratios.

---

> > ### Author Response · Authors · 2025-12-06
> > **Response to Reviewer Rts6 (part 4)**
> >
> > Table 2 (continued)
> >
> > | Dataset | $r = k/p$ | HEM | Graclus | METIS | CGC |
> > |---------|:--------:|-----|---------|--------|------|
> > | CORA (r=0.21)      | 0.21  | 79.88 $\pm$ 1.42 | 81.32 $\pm$ 0.31 | 83.31 $\pm$ 0.30 | 87.96 $\pm$ 0.32 |
> > | CORA (r=0.05)      | 0.05  | 30.21 $\pm$ 0.00 | 63.32 $\pm$ 2.21 | 30.21 $\pm$ 0.00 | 80.57 $\pm$ 0.21 |
> > | DBLP (r=0.23)      | 0.23  | 75.29 $\pm$ 2.10 | 82.30 $\pm$ 0.16 | 83.31 $\pm$ 0.24 | 84.75 $\pm$ 0.54 |
> > | DBLP (r=0.05)      | 0.05  | 66.77 $\pm$ 2.57 | 64.65 $\pm$ 0.41 | 53.18 $\pm$ 4.75 | 80.74 $\pm$ 0.43 |
> > | DBLP (r=0.01)      | 0.01  | 44.71 $\pm$ 0.00 | 44.71 $\pm$ 0.00 | 44.71 $\pm$ 0.00 | 76.77 $\pm$ 0.12 |
> > | CO-CS (r=0.14)     | 0.14  | 85.76 $\pm$ 0.90 | 90.52 $\pm$ 0.64 | 90.94 $\pm$ 0.48 | 92.53 $\pm$ 0.67 |
> > | CO-CS (r=0.03)     | 0.03  | 22.56 $\pm$ 0.00 | 60.75 $\pm$ 2.61 | 23.77 $\pm$ 3.30 | 91.13 $\pm$ 0.31 |
> > | CO-CS (r=0.01)     | 0.01  | 22.56 $\pm$ 0.00 | 22.56 $\pm$ 0.00 | 22.56 $\pm$ 0.00 | 85.66 $\pm$ 0.45 |
> > | PUBMED (r=0.235)   | 0.235 | 82.04 $\pm$ 0.46 | 84.37 $\pm$ 0.42 | 82.97 $\pm$ 0.27 | 86.84 $\pm$ 0.11 |
> > | PUBMED (r=0.07)    | 0.07  | 57.50 $\pm$ 0.13 | 81.29 $\pm$ 0.22 | 57.30 $\pm$ 0.38 | 86.02 $\pm$ 0.28 |
> > | PUBMED (r=0.02)    | 0.02  | 39.27 $\pm$ 0.01 | 39.27 $\pm$ 0.01 | 39.17 $\pm$ 0.05 | 83.41 $\pm$ 0.72 |
> > | CO-PHYSICS (r=0.09)  | 0.09  | 81.57 $\pm$ 1.37 | 16.67 $\pm$ 0.00 | 16.67 $\pm$ 0.00 | 94.79 $\pm$ 0.19 |
> > | CO-PHYSICS (r=0.01)  | 0.01  | 50.52 $\pm$ 0.00 | 50.52 $\pm$ 0.00 | 50.52 $\pm$ 0.00 | 94.15 $\pm$ 0.37 |
> > | CO-PHYSICS (r=0.005) | 0.005 | 50.52 $\pm$ 0.00 | 50.52 $\pm$ 0.00 | 50.52 $\pm$ 0.00 | 92.43 $\pm$ 0.58 |
> >
> > - For large datasets such as Genius and OGBN-Arxiv, several baseline methods (FGC, LAGC, SCAL) initially encountered out-of-memory (OOM) errors. We reimplemented their computations using sparse matrices and memory-optimized code. This enabled all baselines to run on the OGBN-Arxiv dataset without OOM issues, and the reported accuracies were recomputed under these optimized settings. However, LAGC and FGC still experienced OOM failures (requiring over 200~GB vRAM) for some larger coarsening ratios. We have updated Table 3 in the experimental section of the revised manuscript.
> >
> > Table 3: Comparison of memory-optimized LAGC/FGC, SCAL and CGC across large datasets and coarsening ratios.
> >
> >
> > | Dataset      | r = k/p | SCAL              | FGC                | LAGC               | CGC                | Whole Data        |
> > |--------------|:-------:|-------------------|--------------------|--------------------|--------------------|--------------------|
> > | Genius       | 0.06    | 79.99 $\pm$ 0.11  | OOM                | OOM                | 80.00 $\pm$ 0.03   | 86.71 $\pm$ 0.12   |
> > | Genius       | 0.02    | 80.01 $\pm$ 0.09  | OOM                | OOM                | 79.97 $\pm$ 0.08   | --                 |
> > | Genius       | 0.009   | 79.98 $\pm$ 0.09  | OOM                | OOM                | 79.91 $\pm$ 0.14   | --                 |
> > | Genius       | 0.007   | 79.99 $\pm$ 0.11  | OOM                | OOM                | 78.63 $\pm$ 0.03   | --                 |
> > | OGBN-Arxiv   | 0.15    | 27.50 $\pm$ 1.41  | OOM                | OOM                | 53.83 $\pm$ 0.23   | 56.24 $\pm$ 0.22   |
> > | OGBN-Arxiv   | 0.04    | 7.55 $\pm$ 0.25   | 16.05 $\pm$ 0.06   | 29.79 $\pm$ 0.09   | 52.81 $\pm$ 0.33   | --                 |
> > | OGBN-Arxiv   | 0.01    | 5.86 $\pm$ 0.00   | 31.00 $\pm$ 0.20   | 32.37 $\pm$ 0.22   | 50.91 $\pm$ 0.42   | --                 |
> > | OGBN-Arxiv   | 0.004   | 5.86 $\pm$ 0.00   | 22.19 $\pm$ 0.29   | 30.45 $\pm$ 0.88   | 46.86 $\pm$ 0.50   | --                 |
> > | OGBN-Arxiv   | 0.001   | 5.86 $\pm$ 0.00   | 19.34 $\pm$ 0.01   | 25.71 $\pm$ 0.68   | 42.01 $\pm$ 0.56   | --                 |
> >
> >
> >
> > ## 3.  Add ablations: label aggregation (training-only), leader vs. non-leader, hop-size, and leader-selection metrics.
> >
> > **A3.** Thank you for the valuable suggestion. To analyze the effect of different design choices in CGC, we performed ablation studies on two key variants:
> > (i) **Random-leader initialization** and
> > (ii) **Random-neighbor merge** under restricted hop sizes.
> >
> > In the **Random-leader** variant, we randomly select initial coalition leaders across 10 independent runs, and report the mean accuracy along with the mean coarsening ratio achieved. In the **Random-neighbor merge** ablation, each node selects a random neighbor within a limited hop neighborhood for merging, instead of using Dirichlet-energy-based marginal contribution. This evaluates the role of informed coalition formation. The results below show that both ablations significantly degrade performance relative to CGC, confirming the importance of energy-driven coalition formation and structured leader selection.
> >
> > Table 4: Table: Ablation study comparison of random leader selection, random neighbor merging and CGC (Descending degree order)

---

> ### Author Response · Authors · 2025-12-06
> **Response to Reviewer Rts6 (part 5)**
>
> Table 4 (continued)
>
> | Dataset | $r = k/p$ | Random-leader (Achieved $r$) | Random-neighbor merge | CGC (Desc) | Whole Data |
> |---------|:--------:|----------------------------|-----------------------------|----------------|-------------|
> | CORA     | 0.21  | $86.90 \pm 0.92$ (0.27) | $83.03 \pm 0.12$ | $87.96 \pm 0.32$ | $89.50 \pm 1.20$ |
> | CORA     | 0.05  | $79.09 \pm 1.33$ (0.09) | $62.35 \pm 3.23$ | $80.57 \pm 0.21$ | -- |
> | DBLP     | 0.23  | $83.45 \pm 0.33$ (0.26) | $82.45 \pm 0.16$ | $84.75 \pm 0.54$ | $85.35 \pm 0.86$ |
> | DBLP     | 0.05  | $80.24 \pm 0.16$ (0.06) | $78.44 \pm 2.71$ | $80.74 \pm 0.43$ | -- |
> | DBLP     | 0.01  | $78.87 \pm 0.75$ (0.02) | $68.65 \pm 0.36$ | $76.77 \pm 0.12$ | -- |
> | CO-CS    | 0.14  | $92.11 \pm 0.73$ (0.14) | $89.68 \pm 0.38$ | $92.53 \pm 0.67$ | $93.32 \pm 0.62$ |
> | CO-CS    | 0.03  | $90.94 \pm 0.44$ (0.03) | $63.49 \pm 3.57$ | $91.13 \pm 0.31$ | -- |
> | CO-CS    | 0.01  | $88.25 \pm 0.83$ (0.01) | $35.37 \pm 1.56$ | $85.66 \pm 0.45$ | -- |
> | PUBMED   | 0.235 | $86.53 \pm 0.18$ (0.16) | $82.40 \pm 0.32$ | $86.84 \pm 0.11$ | $88.89 \pm 0.57$ |
> | PUBMED   | 0.07  | $85.77 \pm 0.39$ (0.04) | $41.11 \pm 0.31$ | $86.02 \pm 0.28$ | -- |
> | PUBMED   | 0.02  | $82.93 \pm 0.67$ (0.01) | $81.13 \pm 0.22$ | $83.41 \pm 0.72$ | -- |
> | CO-PHYSICS | 0.09  | $94.42 \pm 0.24$ (0.09) | $88.98 \pm 0.23$ | $94.79 \pm 0.19$ | $96.22 \pm 0.74$ |
> | CO-PHYSICS | 0.01  | $93.88 \pm 0.49$ (0.02) | $67.79 \pm 1.37$ | $94.15 \pm 0.37$ | -- |
> | CO-PHYSICS | 0.005 | $92.12 \pm 0.63$ (0.008) | $67.20 \pm 0.30$ | $92.43 \pm 0.58$ | -- |
>
> These results clearly indicate that removing principled coalition formation substantially harms performance and increases variance, confirming the necessity of our energy-guided player cost and leader selection strategy. The complete ablation discussion has been added to the revised manuscript.
>
>
> ##  Clarifying ϵ-Similarity Definition
> Thank you for the observation. Our definition of $\varepsilon$-similarity is
> indeed feature-dependent because we measure distortion through the Dirichlet
> energy of the graph signal $X$, i.e., $\|X\|_{\Theta} = \mathrm{tr}(X^\top \Theta X)$.
> This differs from classical spectral coarsening, where Rayleigh-quotient
> bounds depend only on the Laplacian eigenvalues and are therefore
> feature-independent. Since CGC is a feature-aware coarsening method and the
> coalition formation is driven by energy evaluated on the input features,
> our similarity criterion naturally depends on $X$. We have clarified this
> distinction in the revised manuscript.
>
>
> # Minor Issues / Clarity / Presentation
>
> ## 1. Explicitly relate $P = C^\dagger$ in Eq. (3) to Eq. (5b).
>
> **A.** In our setting, Eq. (3) and Eq. (5b) define the same coarse features. Let $C \in \mathbb{R}^{n \times k}$ be the boolean node-to-supernode assignment matrix, where $C_{vi} = 1$ iff node $v$ belongs to coalition $S_i$. Then $C^\top C = \mathrm{diag}(|S_1|,\dots,|S_k|)$ and the pseudoinverse is $C^\dagger = (C^\top C)^{-1} C^\top$. Using $P = C^\dagger$ in Eq. (3) gives
> $X_c = PX = C^\dagger X = (C^\top C)^{-1} C^\top X$,
> whose $i$-th row is
> $
> \tilde{x_i} = \frac{1}{|S_i|}\sum_{\substack{v \in S_i}} x_v.
> $
> This is exactly the averaging rule in Eq. (5b), so Eq. (3) and Eq. (5b) are mathematically consistent and define the same coarse feature matrix.
>
>
>
> ## 2. Clarify all Algorithm 1 variables (\texttt{Dn}, \texttt{Scoalition}, \texttt{ccost}).
>
> **A.**
> `DE` replaces the earlier notation `Dn` and denotes the initial Dirichlet-energy cost of each player when each node is treated as a singleton coalition.
> `S_coalition` tracks nodes that already belong to a coalition of size greater than one, ensuring they are not reconsidered as leaders in later steps.
> `c_cost` stores the Dirichlet-energy cost of each player in the initial singleton stage, while `c_update` stores the updated player costs inside the candidate coalition $S_i$. After the refinement step, we assign  $c\_{cost}(j) = c_{S_i}(j)$ for all $j \in S_i$.
> Finally, during the update of the coalition structure, all nodes in $S_i$ are removed from the existing coalition set $S$, and the refined coalition $S_i$ is inserted back into $S$ as a new coalition. We have revised the variables in Algorithm 1, and a clear description of all variables has been added in the Algorithm section of the revised manuscript.
>
>
> ## 3. Unify label notation (L vs. Y).
> **A.** We thank the reviewer for pointing this out. We have now unified the notation and use $L$ consistently throughout the manuscript for node labels, replacing the earlier mixture of $Y$ and $L$.
>
> ## 4. Fix missing bibliographic details and grammar issues (e.g., descibed → described)
>
> **A.** We thank the reviewer for the suggestion. We have carefully corrected all grammatical issues (including “descibed” → “described”) and updated the manuscript to fix missing bibliographic details. All references now include complete citation information.

---

> > ### Author Response · Authors · 2025-12-06
> > **Response to Reviewer Rts6 (part 6)**
> >
> > # Questions to the Authors:
> >
> > ## 1. Is $Y$ restricted to training labels in $Y_c = \arg\max(PY)$?
> > **A.** We thank the reviewer for the question. We confirm that \(Y\) is restricted to the training labels when computing $ Y_c = \arg\max(PY) $. Only the training portion of \(Y\) is used in this projection, and no validation or test labels are included, ensuring that no data leakage occurs. We have added this clarification in the experimental section of the revised manuscript.
> >
> > ## 2. Provide a formal statement aligning Eq. (5a) with Eqs. (11-12).
> > **A.** We thank the reviewer for this comment. We have now added a formal statement clarifying that the energy minimization in Eq. (5a) and the potential definition in Eqs. (11)-(12) refer to the same underlying quantity. Specifically, for any mapping profile $m$ that induces a partition $\mathcal{S} = \{S_1, \dots, S_K\}$, the global potential $\Phi(m)$ in Eq. (12) is equal to the Dirichlet energy objective in Eq. (5a). Thus, minimizing Eq. (5a) over valid partitions is equivalent to minimizing $\Phi(m)$ over mapping profiles. We have now added a formal statement after Equation (12) clarifying this equivalence in the revised manuscript as:
> >
> > **Alignment of Energy Objective and Global Potential Function**
> >
> > Let $\mathcal{S} = \{S_1, \dots, S_K\}$ be a valid partition of $V$, and let $m$ be any mapping profile that induces $\mathcal{S}$. The global potential $\Phi(m)$ defined in Equation (12) coincides with the Dirichlet energy objective given in Equation (5a). Specifically,
> >
> > $$
> > \Phi(m)
> > = \sum_{S_i \in \mathcal{S}} \phi(R_{S_i}^m)
> > = \sum_{S_i \in \mathcal{S}}
> > \left(
> >     \frac{1}{2} \sum_{(u,v)\in E_{S_i}}
> >     w_{uv}\,\|x_u - x_v\|^2
> > \right)
> > = \sum_{\substack{S_i, S_j \in \mathcal{S} \\ i < j}} \tilde{w}_{ij}\,\|\tilde{x}_i - \tilde{x}_j\|^2.
> > $$
> >
> > This shows that minimizing the objective in Equation (5a) over all valid partitions $\mathcal{S}$ is equivalent to minimizing the global potential $\Phi(m)$ over all mapping profiles.
> >
> >
> > ## 3. Do Theorems 2–3 hold under leader-restricted 1-hop coalitions?
> > **A.**  Yes. The theoretical guarantees remain valid under the leader-restricted 1-hop coalition rule used in CGC.
> > Although the action space is restricted so that only leader nodes initiate coalitions and only with their
> > 1-hop neighbors, every admissible deviation still induces a strict decrease in the global potential.
> > Therefore, the potential-game structure is preserved, and the best-response dynamics still converge to a
> > Pure Nash Equilibrium. The validity of Theorem 2 (existence of a potential function) and Theorem 3
> > (guaranteed PNE under CGC) is unaffected by this restricted coalition formation process.
> >
> >
> >
> > ## 4. What is the formal per-iteration complexity and convergence bound?
> > **A.**
> > We thank the reviewer for this question regarding the formal per-iteration complexity and convergence behavior of our algorithm.
> >
> > One iteration is defined as a full pass over the current set of leaders, i.e., a sweep that updates each leader once. In a single iteration, at most $k$ leaders are processed. For each leader, the candidate coalition contains at most $\Delta$ one-hop neighbors. Computing all marginal costs within this coalition requires at most $O(\Delta^{2})$ energy evaluations, while checking whether these nodes already belong to other coalitions and updating the coalition structure is bounded by $O(p\Delta)$. Thus, the per-iteration cost is $ O\big(k(\Delta^{2} + p\Delta)\big).$
> >
> > Since $p > \Delta$ for any graph, the dominant term is $p\Delta$, and this expression simplifies to $O(kp\Delta)$, which is the form reported in Section 4.2. The earlier $O(\Delta)$ term in Section~3 referred only to the number of local stability checks inside a single coalition and not to the full per-iteration cost; this has now been clarified.
> >
> > Regarding convergence, CGC defines a finite exact potential game: every accepted coalition update yields the same decrease in the global potential as in the individual player's cost, and the potential is a non-negative Dirichlet-energy quantity. Hence the potential cannot decrease indefinitely, and only finitely many improving deviations are possible. The repeat–until loop therefore converges in a finite number of iterations and always reaches a PNE.
> >
> > More concretely, after the first full sweep over the $p$ nodes, the leader set stabilizes and contains exactly $k$ leaders. In the subsequent dynamics, only the remaining $p-k$ follower nodes may change their coalition, and each such move strictly decreases the Dirichlet-energy potential. Since each follower can choose among at most $k$ leaders, a follower can switch coalitions at most $(k-1)$ times. Hence, the total number of accepted coalition updates is bounded by $(p-k)(k-1) = O(pk)$.

---

> > > ### Author Response · Authors · 2025-12-06
> > > **Response to Reviewer Rts6 (part 7)**
> > >
> > > These complexity and convergence guarantees have been incorporated into Section 4.2 of the revised manuscript for clarity.
> > >
> > > ## 5. Include OGBN-Arxiv results with official splits.
> > >
> > > **A.** Thank you for the suggestion. We have now included results on the **ogbn-arxiv** dataset using the **official OGB splits**. The performance of CGC under different coarsening ratios is reported in Table 5, demonstrating that CGC maintains competitive accuracy even under strong coarsening. In the revised manuscript, we have added this table and the corresponding discussion in the experimental section.
> > >
> > > Table 5: Node classification accuracy on ogbn-arxiv using official OGB splits
> > >
> > > | Dataset   | r = k/p| CGC Accuracy |
> > > |--------------|---------------|------------------------|
> > > | ogbn-arxiv   | $1.0$         | $53.44 \pm 0.34$        |
> > > | ogbn-arxiv   | $0.15$        | $47.23 \pm 0.35$        |
> > > | ogbn-arxiv   | $0.04$        | $46.28 \pm 0.36$        |
> > > | ogbn-arxiv   | $0.01$        | $44.17 \pm 0.46$        |
> > > | ogbn-arxiv   | $0.004$       | $41.85 \pm 0.59$        |
> > > | ogbn-arxiv   | $0.001$       | $38.75 \pm 0.59$        |
> > >
> > >
> > > ## 6. Provide ablations: leader variants, hop sizes, smoothing, alternative costs.
> > >
> > > **A.** Thank you for the valuable suggestion. To analyze the effect of different design choices in CGC, we performed ablation studies on two key variants: (i) Random-leader selection and (ii) Random-neighbor merge under restricted hop sizes.
> > >
> > > In the \textbf{Random-leader} variant, we randomly select initial coalition leaders across 10 independent runs and report the mean accuracy along with the mean coarsening ratio achieved. In the \textbf{Random-neighbor merge} ablation, each node selects a random neighbor within a limited hop neighborhood for merging, instead of using Dirichlet-energy-based marginal contribution. This evaluates the role of informed coalition formation. The results below show that both ablations significantly degrade performance relative to CGC, confirming the importance of energy-driven coalition formation and structured leader selection.
> > >
> > > Table 6: Table: Ablation study comparison of random leader selection, random neighbor merging and CGC (Descending degree order)
> > >
> > > | Dataset | $r = k/p$ | Random-leader | Random-neighbor merge | CGC (Desc) | Whole Data |
> > > |---------|:--------:|----------------------------|-----------------------------|----------------|-------------|
> > > | CORA     | 0.21  | $86.90 \pm 0.92$ (0.27) | $83.03 \pm 0.12$ | $87.96 \pm 0.32$ | $89.50 \pm 1.20$ |
> > > | CORA     | 0.05  | $79.09 \pm 1.33$ (0.09) | $62.35 \pm 3.23$ | $80.57 \pm 0.21$ | -- |
> > > | DBLP     | 0.23  | $83.45 \pm 0.33$ (0.26) | $82.45 \pm 0.16$ | $84.75 \pm 0.54$ | $85.35 \pm 0.86$ |
> > > | DBLP     | 0.05  | $80.24 \pm 0.16$ (0.06) | $78.44 \pm 2.71$ | $80.74 \pm 0.43$ | -- |
> > > | DBLP     | 0.01  | $78.87 \pm 0.75$ (0.02) | $68.65 \pm 0.36$ | $76.77 \pm 0.12$ | -- |
> > > | CO-CS    | 0.14  | $92.11 \pm 0.73$ (0.14) | $89.68 \pm 0.38$ | $92.53 \pm 0.67$ | $93.32 \pm 0.62$ |
> > > | CO-CS    | 0.03  | $90.94 \pm 0.44$ (0.03) | $63.49 \pm 3.57$ | $91.13 \pm 0.31$ | -- |
> > > | CO-CS    | 0.01  | $88.25 \pm 0.83$ (0.01) | $35.37 \pm 1.56$ | $85.66 \pm 0.45$ | -- |
> > > | PUBMED   | 0.235 | $86.53 \pm 0.18$ (0.16) | $82.40 \pm 0.32$ | $86.84 \pm 0.11$ | $88.89 \pm 0.57$ |
> > > | PUBMED   | 0.07  | $85.77 \pm 0.39$ (0.04) | $41.11 \pm 0.31$ | $86.02 \pm 0.28$ | -- |
> > > | PUBMED   | 0.02  | $82.93 \pm 0.67$ (0.01) | $81.13 \pm 0.22$ | $83.41 \pm 0.72$ | -- |
> > > | CO-PHYSICS | 0.09  | $94.42 \pm 0.24$ (0.09) | $88.98 \pm 0.23$ | $94.79 \pm 0.19$ | $96.22 \pm 0.74$ |
> > > | CO-PHYSICS | 0.01  | $93.88 \pm 0.49$ (0.02) | $67.79 \pm 1.37$ | $94.15 \pm 0.37$ | -- |
> > > | CO-PHYSICS | 0.005 | $92.12 \pm 0.63$ (0.008) | $67.20 \pm 0.30$ | $92.43 \pm 0.58$ | -- |
> > >
> > > These results clearly indicate that removing principled coalition formation substantially harms performance and increases variance, confirming the necessity of our energy-guided player cost and leader selection strategy. The complete ablation discussion has been added to the revised manuscript.
> > >
> > >
> > > ## 7. Add scalable non-learning baselines.
> > > **A.**
> > > We thank the reviewer for suggesting the inclusion of scalable non-learning baselines. Node classification accuracy (%) comparing CGC with scalable non-learning graph coarsening baselines HEM, Graclus, and METIS under different coarsening ratios $r = k/p$. Each value is reported as mean $\pm$ standard deviation over 10 runs. The results, summarized in Table 6, show that CGC consistently matches or outperforms these baselines across multiple datasets and coarsening ratios $r = k/p$, particularly under more aggressive coarsening.
> > >
> > > We have added Table 7 and its explanation in the experimental section of the revised manuscript.

---

> > > > ### Author Response · Authors · 2025-12-06
> > > > **Response to Reviewer Rts6 (part 8)**
> > > >
> > > > Table 7: Node classification accuracy (%) comparing CGC with scalable non-learning graph coarsening baselines HEM, Graclus, and METIS under different coarsening ratios.
> > > >
> > > > | Dataset | $r = k/p$ | HEM | Graclus | METIS | CGC |
> > > > |---------|:--------:|-----|---------|--------|------|
> > > > | CORA (r=0.21)      | 0.21  | 79.88 $\pm$ 1.42 | 81.32 $\pm$ 0.31 | 83.31 $\pm$ 0.30 | 87.96 $\pm$ 0.32 |
> > > > | CORA (r=0.05)      | 0.05  | 30.21 $\pm$ 0.00 | 63.32 $\pm$ 2.21 | 30.21 $\pm$ 0.00 | 80.57 $\pm$ 0.21 |
> > > > | DBLP (r=0.23)      | 0.23  | 75.29 $\pm$ 2.10 | 82.30 $\pm$ 0.16 | 83.31 $\pm$ 0.24 | 84.75 $\pm$ 0.54 |
> > > > | DBLP (r=0.05)      | 0.05  | 66.77 $\pm$ 2.57 | 64.65 $\pm$ 0.41 | 53.18 $\pm$ 4.75 | 80.74 $\pm$ 0.43 |
> > > > | DBLP (r=0.01)      | 0.01  | 44.71 $\pm$ 0.00 | 44.71 $\pm$ 0.00 | 44.71 $\pm$ 0.00 | 76.77 $\pm$ 0.12 |
> > > > | CO-CS (r=0.14)     | 0.14  | 85.76 $\pm$ 0.90 | 90.52 $\pm$ 0.64 | 90.94 $\pm$ 0.48 | 92.53 $\pm$ 0.67 |
> > > > | CO-CS (r=0.03)     | 0.03  | 22.56 $\pm$ 0.00 | 60.75 $\pm$ 2.61 | 23.77 $\pm$ 3.30 | 91.13 $\pm$ 0.31 |
> > > > | CO-CS (r=0.01)     | 0.01  | 22.56 $\pm$ 0.00 | 22.56 $\pm$ 0.00 | 22.56 $\pm$ 0.00 | 85.66 $\pm$ 0.45 |
> > > > | PUBMED (r=0.235)   | 0.235 | 82.04 $\pm$ 0.46 | 84.37 $\pm$ 0.42 | 82.97 $\pm$ 0.27 | 86.84 $\pm$ 0.11 |
> > > > | PUBMED (r=0.07)    | 0.07  | 57.50 $\pm$ 0.13 | 81.29 $\pm$ 0.22 | 57.30 $\pm$ 0.38 | 86.02 $\pm$ 0.28 |
> > > > | PUBMED (r=0.02)    | 0.02  | 39.27 $\pm$ 0.01 | 39.27 $\pm$ 0.01 | 39.17 $\pm$ 0.05 | 83.41 $\pm$ 0.72 |
> > > > | CO-PHYSICS (r=0.09)  | 0.09  | 81.57 $\pm$ 1.37 | 16.67 $\pm$ 0.00 | 16.67 $\pm$ 0.00 | 94.79 $\pm$ 0.19 |
> > > > | CO-PHYSICS (r=0.01)  | 0.01  | 50.52 $\pm$ 0.00 | 50.52 $\pm$ 0.00 | 50.52 $\pm$ 0.00 | 94.15 $\pm$ 0.37 |
> > > > | CO-PHYSICS (r=0.005) | 0.005 | 50.52 $\pm$ 0.00 | 50.52 $\pm$ 0.00 | 50.52 $\pm$ 0.00 | 92.43 $\pm$ 0.58 |

---

> > > > > ### Author Response · Authors · 2025-12-08
> > > > > **Response to Reviewer Rts6 (part 9)**
> > > > >
> > > > > # (C) NP-Completeness Argument Seems Detached from the Actual Problem
> > > > >
> > > > > **A.**
> > > > >
> > > > > We agree that the earlier version of Theorem 1 in Appendix A.1 was stated for an abstract “Stable Coalition’’ problem with an arbitrary polynomial-time function $f$. In the revised manuscript, we have rewritten Appendix A.1 so that both the problem statement and the NP-hardness proof are formulated directly based on the graph-based cost function.
> > > > >
> > > > > Concretely, we now introduce the Coalition Instability Detection decision problem, which asks whether there exists a non-empty subset $T \subseteq S$ such that the internal-stability inequality is violated, i.e.,
> > > > >
> > > > > $$
> > > > > \sum_{i \in T} c_i > v(T)
> > > > > $$
> > > > >
> > > > > where both $c_i$ and $v(T)$ are defined via the Dirichlet-energy-based cost sharing in Eq. (7).
> > > > > We first show that this problem is in NP by observing that, for any candidate $T$, both the Dirichlet energy $\mathrm{DE}(T)$ and the induced costs $c_T(j)$ (Eq. (7)) can be evaluated in polynomial time using the incident-edge sets $E_x(T)$ on the coarsened graph.
> > > > >
> > > > > **Proof sketch of NP-hardness**  (fully detailed in revised Appendix~A.1).
> > > > >
> > > > > To prove NP-hardness, we give a polynomial-time many-one reduction from
> > > > > the classical \textsc{Max-Cut} decision problem. An instance of
> > > > > \textsc{Max-Cut} consists of a weighted graph $G=(V,E)$ with
> > > > > nonnegative integer weights $\{w_{uv}\}$ and a threshold $K \in \mathbb{N}$.
> > > > > The question is whether there exists $U \subseteq V$ such that
> > > > > $\mathrm{cut}_G(U, V\setminus U) \ge K$.
> > > > >
> > > > > We embed this into our CGC framework by considering a special (but
> > > > > valid) instantiation of Eqs.(6,7):
> > > > > - the coarsened graph coincides with the original graph $G$
> > > > >   (each vertex is its own supernode), and
> > > > > - coalitions are represented by scalar indicator signals
> > > > >   $f_U : V \to \{0,1\}$, with $f_U(u)=1$ if $u \in U$ and $0$ otherwise.
> > > > >
> > > > > For such an indicator $f_U$, the Dirichlet energy in Eq. (7) reduces to
> > > > >
> > > > > $$
> > > > > \mathrm{DE}(T)
> > > > > = \frac{1}{2} \sum_{(u,v)\in E} w_{uv}\bigl(f_U(u) - f_U(v)\bigr)^2
> > > > > = \mathrm{cut}_G(U, V \setminus U), \quad U = T \cap V.
> > > > > $$
> > > > >
> > > > >
> > > > >
> > > > > Thus, in this CGC instance, Eq.~(7) induces exactly a cut-based
> > > > > Dirichlet energy on $G$. We then define the coalition value as
> > > > >
> > > > > $$
> > > > > v(T) = W - \mathrm{DE}(T) = W - \mathrm{cut}_G(U, V \setminus U),
> > > > > $$
> > > > >
> > > > >
> > > > > where $W = \sum_{(u,v)\in E} w_{uv}$ is the total edge weight.
> > > > > Next, we add a single dummy player $p$ and set the player set to
> > > > > $S = V \cup$ {$p$}. We assign costs
> > > > > $c_i = 0$ for all $i \in V$ and
> > > > > $c_p = W - K + \tfrac{1}{2}$.
> > > > >
> > > > > One can then verify the following equivalence:
> > > > >
> > > > > - Any coalition $T$ that does not contain $p$ has
> > > > >   $\sum_{i\in T} c_i = 0$ and $v(T) \ge 0$, so it cannot violate
> > > > >   internal stability.
> > > > >
> > > > > - For any coalition $T = U \cup {p}$  with $U \subseteq V$, we have
> > > > > $$
> > > > > \sum_{i \in T} c_i = W - K + \tfrac{1}{2},
> > > > > \qquad
> > > > > v(T) = W - \mathrm{cut}_G(U, V \setminus U),
> > > > > $$
> > > > >
> > > > >   so the instability condition
> > > > >   $\sum_{i\in T} c_i > v(T)$ is equivalent to
> > > > >
> > > > > $$
> > > > > W - K + \tfrac{1}{2} > W - \mathrm{cut}_G(U, V \setminus U)
> > > > > \Longleftrightarrow
> > > > > \mathrm{cut}_G(U, V \setminus U) > K - \tfrac{1}{2}.
> > > > > $$
> > > > >
> > > > >
> > > > >
> > > > >   Since $\mathrm{cut}_G(U, V\setminus U)$ is an integer, this holds if
> > > > >   and only if $\mathrm{cut}_G(U, V\setminus U) \ge K$.
> > > > >
> > > > > Hence, the Max-Cut instance $(G,K)$ is a “yes” instance if
> > > > > and only if there exists a coalition $T \subseteq S$ with
> > > > > $\sum_{i\in T} c_i > v(T)$ under the Dirichlet-energy-based cost
> > > > > function derived from Eq.(7). This establishes NP-hardness of the
> > > > > graph-coarsening stability check with the specific Dirichlet
> > > > > energy used in Eqs.(6,7), not for an unrelated abstract stability
> > > > > problem. The full formal statement and detailed proof are given in the
> > > > > revised Appendix~A.1.
> > > > >
> > > > >
> > > > >
> > > > > ---
> > > > > References:
> > > > >
> > > > > [1] Zengfeng Huang, Shengzhong Zhang, Chong Xi, Tang Liu, and Min Zhou. 2021. Scaling up graph neural networks via graph coarsening. In *Proceedings of the 27th ACM SIGKDD Conference on Knowledge Discovery & Data Mining (KDD)*, pages 675–684. ACM.
> > > > >
> > > > > [2] Manoj Kumar, Anurag Sharma, Shashwat Saxena, and Sandeep Kumar. 2023. Featured graph coarsening with similarity guarantees. In *Proceedings of the International Conference on Machine Learning (ICML)*, pages 17953–17975. PMLR.
> > > > >
> > > > > [3] Manoj Kumar, Subhanu Halder, Archit Kane, Ruchir Gupta, and Sandeep Kumar. 2024. Optimization framework for semi-supervised attributed graph coarsening. In *Proceedings of the 40th Conference on Uncertainty in Artificial Intelligence (UAI)*.

---

### Decision · Action_Editor_RsLL · 2025-12-28

**Recommendation:** Accept with minor revision

**Additional Comments:**

Based on the final discussion round with the reviewers, please address the following in your final version:

1. **Motivation and positioning:** tighten the introduction to clearly articulate the gap relative to (a) optimization-based coarseners that already minimize spectral or Dirichlet-style objectives, and (b) game-theoretic clustering. Make the “why game theory here” argument crisp and concrete (what it enables beyond a direct local optimization heuristic, and what properties are uniquely leveraged in practice).

2. **Notation and definitions cleanup:** ensure every symbol is used consistently across sections (labels, Laplacian dimensions, mapping profile, number of coalitions/supernodes). In Algorithm 1, define every variable used (including any cost arrays and coalition tracking sets) in the main text or caption.

3. **Coalition energy and potential clarity:** make the edge accounting explicit so the reader can verify no double counting. If a factor (such as 1/2) is used, define it once and connect it cleanly to the equality between the global potential and the global Dirichlet energy objective.

4. **Signed weights / Laplacian assumptions:** clarify whether edge weights are assumed nonnegative. If negative weights are allowed, state the precise Laplacian convention used and any implications (since typical Laplacian properties used in spectral arguments assume nonnegative weights).

5. **Evaluation protocol clarity in the paper itself:** keep the “no label leakage” clarification in the main manuscript (not only the response). State explicitly how labels are masked and projected, and confirm that test labels are used only for final evaluation on the original graph.

6. **Completeness of tables:** ensure that all referenced tables in the revision appear in the manuscript with filled-in entries.

7. **Reproducibility details:** include concrete implementation details (splits used for each dataset, seeds, hardware/memory notes for baselines, and any sparse/memory-optimized settings). If code is available, include a link; if not, provide sufficient detail to reproduce.

**Audience:**

Yes

**Audience Explanation:**

Graph coarsening is a widely used building block for scaling graph learning and for accelerating GNN training, and TMLR readers working on graph ML, scalable learning, and graph preprocessing would plausibly benefit from (i) an alternative feature-aware coarsening principle tied to Dirichlet energy, (ii) a convergence-guaranteed iterative procedure with a stable outcome, and (iii) the expanded empirical comparisons and ablations that help map when coarsening preserves downstream performance.

**Claims And Evidence:**

Yes

**Claims Explanation:**

The submission makes two main types of claims: (1) a game-theoretic formulation with convergence guarantees, and (2) empirical gains from the proposed CGC coarsening procedure. On the theory side, the paper provides a clear potential-game construction and argues convergence to a stable outcome under the proposed dynamics. On the empirical side, the revision substantially strengthens the evidence: the authors clarify the evaluation protocol (training on the coarsened graph while evaluating predictions on the original graph), explicitly state that label projection uses training labels only (preventing leakage), add comparisons to additional baselines (including scalable non-learning baselines and a recent supervised coarsening baseline), and include ablations and sensitivity analysis (random leader initialization, random neighbor merge). Overall, the experimental story is now reasonably complete and supports the main performance and efficiency claims.

That said, a few issues raised by the reviewers remain primarily about presentation and verifiability rather than core soundness: the manuscript still needs a cleaner and more self-contained motivation, tighter notation consistency throughout, and a final pass to ensure all equations and tables are fully defined, non-ambiguous, and complete in the paper (not only in the response).